# Efficient Active Learning with Abstention

**Yinglun Zhu**
Department of Computer Sciences
University of Wisconsin-Madison
Madison, WI 53706
yinglun@cs.wisc.edu

**Robert Nowak**
Department of Electrical and Computer Engineering
University of Wisconsin-Madison
Madison, WI 53706
rdnowak@wisc.edu

## Abstract

The goal of active learning is to achieve the same accuracy achievable by passive learning, while using much fewer labels. Exponential savings in terms of label complexity have been proved in very special cases, but fundamental lower bounds show that such improvements are impossible in general. This suggests a need to explore alternative goals for active learning. Learning with abstention is one such alternative. In this setting, the active learning algorithm may abstain from prediction and incur an error that is marginally smaller than random guessing. We develop the first computationally efficient active learning algorithm with abstention. Our algorithm provably achieves $\mathrm{polylog}(\frac{1}{\varepsilon})$ label complexity, without any low noise conditions. Such performance guarantee reduces the label complexity by an exponential factor, relative to passive learning and active learning that is not allowed to abstain. Furthermore, our algorithm is guaranteed to only abstain on hard examples (where the true label distribution is close to a fair coin), a novel property we term *proper abstention* that also leads to a host of other desirable characteristics (e.g., recovering minimax guarantees in the standard setting, and avoiding the undesirable "noise-seeking" behavior often seen in active learning). We also provide novel extensions of our algorithm that achieve *constant* label complexity and deal with model misspecification.

## 1 Introduction

Active learning aims at learning an accurate classifier with a small number of labeled data points (Settles, 2009; Hanneke, 2014). Active learning has become increasingly important in modern application of machine learning, where unlabeled data points are abundant yet the labeling process requires expensive time and effort. Empirical successes of active learning have been observed in many areas (Tong and Koller, 2001; Gal et al., 2017; Sener and Savarese, 2018). In noise-free or certain low-noise cases (i.e., under Massart noise (Massart and Nédélec, 2006)), active learning algorithms with *provable* exponential savings over the passive counterpart have been developed (Balcan et al., 2007; Hanneke, 2007; Dasgupta et al., 2009; Hsu, 2010; Dekel et al., 2012; Hanneke, 2014; Zhang and Chaudhuri, 2014; Krishnamurthy et al., 2019; Katz-Samuels et al., 2021). On the other hand, however, not much can be said in the general case. In fact, Kääriäinen (2006) provides a $\Omega(\frac{1}{\varepsilon^2})$ lower bound by reducing active learning to a simple mean estimation problem: It takes $\Omega(\frac{1}{\varepsilon^2})$ samples to distinguish $\eta(x) = \frac{1}{2} + \varepsilon$ and $\eta(x) = \frac{1}{2} - \varepsilon$. Even with the relatively benign Tsybakov noise (Tsybakov, 2004), Castro and Nowak (2006, 2008) derive a $\Omega(\mathrm{poly}(\frac{1}{\varepsilon}))$ lower bound, again, indicating that exponential speedup over passive learning is not possible in general. These fundamental lower bounds lay out statistical barriers to active learning, and suggests considering a refinement of the label complexity goals in active learning (Kääriäinen, 2006).

Inspecting these lower bounds, one can see that active learning suffers from classifying hard examples that are close to the decision boundary. However, *do we really require a trained classifier to do well*

*on those hard examples?* In high-risk domains such as medical imaging, it makes more sense for the classifier to abstain from making the decision and leave the problem to a human expert. Such idea is formalized under Chow's error (Chow, 1970): Whenever the classifier chooses to abstain, a loss that is barely smaller than random guessing, i.e., $\frac{1}{2} - \gamma$, is incurred. The parameter $\gamma$ should be thought as a small positive quantity, e.g., $\gamma = 0.01$. The inclusion of abstention is not only practically interesting, but also provides a statistical refinement of the label complexity goal of active learning: Achieving exponential improvement under Chow's excess error. When abstention is allowed as an action, Puchkin and Zhivotovskiy (2021) shows, for the first time, that exponential improvement in label complexity can be achieved by active learning in the general setting. However, the approach provided in Puchkin and Zhivotovskiy (2021) can not be efficiently implemented. Their algorithm follows the disagreement-based approach and requires maintaining a version space and checking whether or not an example lies in the region of disagreement. It is not clear how to generally implement these operations besides enumeration (Beygelzimer et al., 2010). Moreover, their algorithm relies on an Empirical Risk Minimization (ERM) oracle, which is known to be NP-Hard even for a simple linear hypothesis class (Guruswami and Raghavendra, 2009).

In this paper, we break the computational barrier and design an efficient active learning algorithm with exponential improvement in label complexity relative to conventional passive learning. The algorithm relies on weighted square loss regression oracle, which can be efficiently implemented in many cases (Krishnamurthy et al., 2017, 2019; Foster et al., 2018, 2020). The algorithm also abstains properly, i.e., abstain only when it is the optimal choice, which allows us to easily translate the guarantees to the *standard* excess error. Along the way, we propose new noise-seeking noise conditions and show that: "uncertainty-based" active learners can be easily trapped, yet our algorithm provably overcome these noise-seeking conditions. As an extension, we also provide the first algorithm that enjoys *constant* label complexity for a *general* set of regression functions.

## 1.1 Problem setting

Let $\mathcal{X}$ denote the input space and $\mathcal{Y}$ denote the label space. We focus on the binary classification problem where $\mathcal{Y} = \{+1, -1\}$. The joint distribution over $\mathcal{X} \times \mathcal{Y}$ is denoted as $\mathcal{D}_{\mathcal{X}\mathcal{Y}}$. We use $\mathcal{D}_{\mathcal{X}}$ to denote the marginal distribution over the input space $\mathcal{X}$, and use $\mathcal{D}_{\mathcal{Y}|x}$ to denote the conditional distribution of $\mathcal{Y}$ with respect to any $x \in \mathcal{X}$. We define $\eta(x) := \mathbb{P}_{y \sim \mathcal{D}_{\mathcal{Y}|x}}(y = +1)$ as the conditional probability of taking a positive label. We consider the standard active learning setup where $(x, y) \sim \mathcal{D}_{\mathcal{X}\mathcal{Y}}$ but $y$ is observed only after a label querying. We consider hypothesis class $\mathcal{H} : \mathcal{X} \to \mathcal{Y}$. For any classifier $h \in \mathcal{H}$, its (standard) error is defined as $\mathrm{err}(h) := \mathbb{P}_{(x,y) \sim \mathcal{D}_{\mathcal{X}\mathcal{Y}}}(h(x) \neq y)$.

**Function approximation.** We focus on the case where the hypothesis class $\mathcal{H}$ is induced from a set of regression functions $\mathcal{F} : \mathcal{X} \to [0, 1]$ that predicts the conditional probability $\eta(x)$. We write $\mathcal{H} = \mathcal{H}_{\mathcal{F}} := \{h_f : f \in \mathcal{F}\}$ where $h_f(x) := \mathrm{sign}(2f(x) - 1)$. The "size" of $\mathcal{F}$ is measured by the well-known complexity measure: the *Pseudo dimension* $\mathrm{Pdim}(\mathcal{F})$ (Pollard, 1984; Haussler, 1989, 1995). We assume $\mathrm{Pdim}(\mathcal{F}) < \infty$ throughout the paper.[1] Following existing works in active learning (Dekel et al., 2012; Krishnamurthy et al., 2017, 2019) and contextual bandits (Agarwal et al., 2012; Foster et al., 2018; Foster and Rakhlin, 2020; Simchi-Levi and Xu, 2020), we make the following *realizability* assumption.

**Assumption 1** (Realizability)**.** *The learner is given a set of regressors $\mathcal{F} : \mathcal{X} \to [0, 1]$ such that there exists a $f^\star \in \mathcal{F}$ characterize the true conditional probability, i.e., $f^\star = \eta$.*

The realizability assumption allows *rich function approximation*, which strictly generalizes the setting with linear function approximation studied in active learning (e.g., in (Dekel et al., 2012)). We relax Assumption 1 in Section 4.2 to deal with model misspecification.

**Regression oracle.** We consider a regression oracle over $\mathcal{F}$, which is extensively studied in the literature in active learning and contextual bandits (Krishnamurthy et al., 2017, 2019; Foster et al., 2018, 2020). Given any set $\mathcal{S}$ of weighted examples $(w, x, y) \in \mathbb{R}_+ \times \mathcal{X} \times \mathcal{Y}$ as input, the regression

---

[1]See Appendix D for formal definition of the Pseudo dimension. Many function classes of practical interests have finite Pseudo dimension: (1) when $\mathcal{F}$ is finite, we have $\mathrm{Pdim}(\mathcal{F}) = O(\log|\mathcal{F}|)$; (2) when $\mathcal{F}$ is a set of linear functions/generalized linear function with non-decreasing link function, we have $\mathcal{F} = O(d)$; (3) when $\mathcal{F}$ is a set of degree-$r$ polynomial in $\mathbb{R}^d$, we have $\mathrm{Pdim}(\mathcal{F}) = O(\binom{d+r}{r})$.

oracle outputs

$$\widehat{f} = \arg\min_{f \in \mathcal{F}} \sum_{(w,x,y) \in \mathcal{S}} w(f(x) - y)^2. \tag{1}$$

The regression oracle solves a convex optimization problem with respect to the regression function, and admits closed-form solutions in many cases, e.g., it is reduced to least squares when $f$ is linear. We view the implementation of the regression oracle as an efficient operation and quantify the computational complexity in terms of the number of calls to the regression oracle.

**Chow's excess error (Chow, 1970).** Let $h^\star := h_{f^\star} \in \mathcal{H}$ denote the Bayes classifier. The *standard excess error* of classifier $h \in \mathcal{H}$ is defined as $\mathrm{err}(h) - \mathrm{err}(h^\star)$. Since achieving exponential improvement (of active over passive learning) with respect to the standard excess error is impossible in general (Kääriäinen, 2006), we introduce Chow's excess error next. We consider classifier of the form $\widehat{h} : \mathcal{X} \to \mathcal{Y} \cup \{\perp\}$ where $\perp$ denotes the action of abstention. For any fixed $0 < \gamma < \frac{1}{2}$, the Chow's error is defined as

$$\mathrm{err}_\gamma(\widehat{h}) := \mathbb{P}_{(x,y) \sim \mathcal{D}_{\mathcal{X}\mathcal{Y}}}(\widehat{h}(x) \neq y, \widehat{h}(x) \neq \perp) + (1/2 - \gamma) \cdot \mathbb{P}_{(x,y) \sim \mathcal{D}_{\mathcal{X}\mathcal{Y}}}(\widehat{h}(x) = \perp). \tag{2}$$

The parameter $\gamma$ can be chosen as a small constant, e.g., $\gamma = 0.01$, to avoid excessive abstention: The price of abstention is only marginally smaller than random guess. The *Chow's excess error* is then defined as $\mathrm{err}_\gamma(\widehat{h}) - \mathrm{err}(h^\star)$ (Puchkin and Zhivotovskiy, 2021). For any fixed accuracy level $\varepsilon > 0$, we aim at constructing a classifier $\widehat{h} : \mathcal{X} \to \mathcal{Y} \cup \{\perp\}$ with $\varepsilon$ Chow's excess error and $\mathrm{polylog}(\frac{1}{\varepsilon})$ label complexity. We also relate Chow's excess error to standard excess error in Section 3.

**Remark 1.** *Competing against the optimal Chow's error, i.e., $\mathrm{err}_\gamma(\widehat{h}) - \inf_{h:\mathcal{X} \to \{+1,-1,\perp\}} \mathrm{err}_\gamma(h)$, will eliminate active learning gains. As in Kääriäinen (2006), it suffices to consider a simple problem with $\mathcal{X} = \{x\}$. In order to achieve $\varepsilon$ excess error against the optimal Chow's classifier, we need to distinguish cases $\eta(x) = \frac{1}{2} - \gamma - 2\varepsilon$ and $\eta(x) = \frac{1}{2} - \gamma + 2\varepsilon$, which inevitably requires $\Omega(\frac{1}{\varepsilon^2})$ samples. We defer a detailed discussion (with pictorial explanations) of Chow's excess error in Appendix B.*

## 1.2 Contributions and paper organization

We provide informal statements of our main results in this section. Our results depend on complexity measures such as *value function* disagreement coefficient $\theta$ and eluder dimension $\mathfrak{e}$ (formally defined in Section 2 and Appendix C). These complexity measures are previously analyzed in contextual bandits (Russo and Van Roy, 2013; Foster et al., 2020) and we import them to the active learning setup. These complexity measures are well-bounded for many function classes of practical interests, e.g., we have $\theta, \mathfrak{e} = \widetilde{O}(d)$ for linear and generalized linear functions on $\mathbb{R}^d$.

Our first main contribution is that we design the first *computationally efficient* active learning algorithm (Algorithm 1) that achieves exponential labeling savings, *without any low noise assumptions*.

**Theorem 1** (Informal). *There exists an algorithm that constructs a classifier $\widehat{h} : \mathcal{X} \to \{+1, -1, \perp\}$ with Chow's excess error at most $\varepsilon$ and label complexity $\widetilde{O}(\frac{\theta \mathrm{Pdim}(\mathcal{F})}{\gamma^2} \cdot \mathrm{polylog}(\frac{1}{\varepsilon}))$, without any low noise assumptions. The algorithm can be efficiently implemented via a regression oracle: It takes $\widetilde{O}(\frac{\theta \mathrm{Pdim}(\mathcal{F})}{\varepsilon \gamma^3})$ oracle calls for general $\mathcal{F}$, and $\widetilde{O}(\frac{\theta \mathrm{Pdim}(\mathcal{F})}{\varepsilon \gamma})$ oracle calls for convex $\mathcal{F}$.*

The formal statements are provided in Section 2. The *statistical* guarantees (i.e., label complexity) in Theorem 1 is similar to the one achieved in Puchkin and Zhivotovskiy (2021), with one critical difference: The label complexity provided in Puchkin and Zhivotovskiy (2021) is in terms of the *classifier-based* disagreement coefficient $\check{\theta}$ (Hanneke, 2014). Even for a set of linear classifier, $\check{\theta}$ is only known to be bounded in special cases, e.g., when $\mathcal{D}_\mathcal{X}$ is uniform over the unit sphere (Hanneke, 2007). On the other hand, we have $\theta \leq d$ for any $\mathcal{D}_\mathcal{X}$ (Foster et al., 2020).

We say that a classifier $\widehat{h} : \mathcal{X} \to \{+1, -1, \perp\}$ enjoys proper abstention if it abstains only if abstention is indeed the optimal choice (based on Eq. (2)). For any classifier that enjoys proper abstention, one can easily relate its *standard* excess error to the Chow's excess error, under commonly studied Massart/Tsybakov noises (Massart and Nédélec, 2006; Tsybakov, 2004). The classifier obtained in Theorem 1 enjoys proper abstention, and achieves the following guarantees (formally stated in Section 3.1).

**Theorem 2** (Informal)**.** *Under Massart/Tsybakov noise, with appropriate adjustments, the classifier learned in Theorem 1 achieves the minimax optimal label complexity under standard excess error.*

We also propose new noise conditions that *strictly* generalize the usual Massart/Tsybakov noises, which we call noise-seeking conditions. At a high-level, the noise-seeking conditions allow abundant data points with $\eta(x)$ equal/close to $\frac{1}{2}$. These points are somewhat "harmless" since it hardly matters what label is predicted at that point (in terms of excess error). These seemingly "harmless" data points can, however, cause troubles for any active learning algorithm that requests the label for any point that is uncertain, i.e., the algorithm cannot decide if $|\eta(x) - \frac{1}{2}|$ is strictly greater than 0. We call such algorithms "uncertainty-based" active learners. These algorithms could wastefully sample in these "harmless" regions, ignoring other regions where erring could be much more harmful. We derive the following proposition (formally stated in Section 3.2) under these noise-seeking conditions.

**Proposition 1** (Informal)**.** *For any labeling budget $B \gtrsim \frac{1}{\gamma^2} \cdot \mathrm{polylog}(\frac{1}{\varepsilon})$, there exists a learning problem such that (1) any uncertainty-based active learner suffers standard excess error $\Omega(B^{-1})$; yet (2) the classifier $\widehat{h}$ learned in Theorem 1 achieves standard excess error at most $\varepsilon$.*

The above result demonstrates the superiority of our algorithm over any "uncertainty-based" active learner. Moreover, we show that, under these strictly harder noise-seeking conditions, our algorithm still achieve guarantees similar to the ones stated in Theorem 2.

Before presenting our next main result, we first consider a simple active learning problem with $\mathcal{X} = \{x\}$. Under Massart noise, we have $|\eta(x) - \frac{1}{2}| \geq \tau_0$ for some constant $\tau_0 > 0$. Thus, it takes no more than $O(\tau_0^{-2} \log \frac{1}{\delta})$ labels to achieve $\varepsilon$ standard excess error, no matter how small $\varepsilon$ is. This example shows that, at least in simple cases, we can expect to achieve a *constant* label complexity for active learning, with no dependence on $\frac{1}{\varepsilon}$ at all. To the best of our knowledge, our next result provides the first generalization of such phenomenon to a *general* set of (finite) regression functions, as long as its eluder dimension $\mathfrak{e}$ is bounded.

**Theorem 3** (Informal)**.** *Under Massart noise with parameter $\tau_0$ and a general (finite) set of regression function $\mathcal{F}$. There exists an algorithm that returns a classifier with standard excess error at most $\varepsilon$ and label complexity $O(\frac{\mathfrak{e} \cdot \log(|\mathcal{F}|/\delta)}{\tau_0^2})$, which is independent of $\frac{1}{\varepsilon}$.*

A similar constant label complexity holds with Chow's excess error, without any low noise assumptions. We also provide discussion on why previous algorithms do not achieve such constant label complexity, even in the case with linear functions. We defer formal statements and discussion to Section 4.1. In Section 4.2, we relax Assumption 1 and propose an algorithm that can deal with model misspecification.

**Paper organization.**    The rest of this paper is organized as follows. We present our main algorithm and its guarantees in Section 2. We further analyze our algorithm under standard excess error in Section 3, and discuss other important properties of the algorithm. Extensions of our algorithm, e.g., achieving *constant* label complexity and dealing with model misspecification, are provided in Section 4. We defer the discussion of additional related work and all proofs to the Appendix due to lack of space.

## 2   Efficient active learning with abstention

We provide our main algorithm (Algorithm 1) in this section. Algorithm 1 is an adaptation of the algorithm developed in Krishnamurthy et al. (2017, 2019), which studies active learning under the standard excess error (and Massart/Tsybakov noises). We additionally take the abstention option into consideration, and *manually construct* classifiers using the active set of (uneliminated) regression functions (which do not belong to the original hypothesis class). These new elements allow us to achieve $\varepsilon$ Chow's excess error with $\mathrm{polylog}(\frac{1}{\varepsilon})$ label complexity, without any low noise assumptions.

---

**Algorithm 1** Efficient Active Learning with Abstention

---

**Input:** Accuracy level $\varepsilon > 0$, abstention parameter $\gamma \in (0, 1/2)$ and confidence level $\delta \in (0, 1)$.

1: Define $T := \widetilde{O}(\frac{\theta \, \mathrm{Pdim}(\mathcal{F})}{\varepsilon \, \gamma})$, $M := \lceil \log_2 T \rceil$ and $C_\delta := O(\mathrm{Pdim}(\mathcal{F}) \cdot \log(T/\delta))$.

2: Define $\tau_m := 2^m$ for $m \geq 1$, $\tau_0 := 0$ and $\beta_m := (M - m + 1) \cdot C_\delta$.

3: **for** epoch $m = 1, 2, \ldots, M$ **do**

4:    Get $\widehat{f}_m := \arg\min_{f \in \mathcal{F}} \sum_{t=1}^{\tau_{m-1}} Q_t (f(x_t) - y_t)^2$.
         `// We use` $Q_t \in \{0, 1\}$ `to indicate whether the label of` $x_t$ `is queried.`

5:    (Implicitly) Construct active set of regression functions $\mathcal{F}_m \subseteq \mathcal{F}$ as

$$\mathcal{F}_m := \left\{ f \in \mathcal{F} : \sum_{t=1}^{\tau_{m-1}} Q_t (f(x_t) - y_t)^2 \leq \sum_{t=1}^{\tau_{m-1}} Q_t (\widehat{f}_m(x_t) - y_t)^2 + \beta_m \right\}.$$

6:    Construct classifier $\widehat{h}_m : \mathcal{X} \to \{+1, -1, \perp\}$ as

$$\widehat{h}_m(x) := \begin{cases} \perp, & \text{if } [\mathsf{lcb}(x; \mathcal{F}_m), \mathsf{ucb}(x; \mathcal{F}_m)] \subseteq \left[ \frac{1}{2} - \gamma, \frac{1}{2} + \gamma \right]; \\ \mathrm{sign}(2\widehat{f}_m(x) - 1), & \text{o.w.} \end{cases}$$

   and construct query function $g_m(x) := \mathbb{1}\left( \frac{1}{2} \in (\mathsf{lcb}(x; \mathcal{F}_m), \mathsf{ucb}(x; \mathcal{F}_m)) \right) \cdot \mathbb{1}(\widehat{h}_m(x) \neq \perp)$.

7:    **if** epoch $m = M$ **then**

8:       **Return** classifier $\widehat{h}_M$.

9:    **for** time $t = \tau_{m-1} + 1, \ldots, \tau_m$ **do**

10:      Observe $x_t \sim \mathcal{D}_{\mathcal{X}}$. Set $Q_t := g_m(x_t)$.

11:      **if** $Q_t = 1$ **then**

12:         Query the label $y_t$ of $x_t$.

---

Algorithm 1 runs in epochs of geometrically increasing lengths. At the beginning of epoch $m \in [M]$, Algorithm 1 first computes the empirical best regression function $\widehat{f}_m$ that achieves the smallest cumulative square loss over previously labeled data points ($\widehat{f}_1$ can be selected arbitrarily); it then (implicitly) constructs an active set of regression functions $\mathcal{F}_m$, where the cumulative square loss of each $f \in \mathcal{F}_m$ is not too much larger than the cumulative square loss of empirical best regression function $\widehat{f}_m$. For any $x \in \mathcal{X}$, based on the active set of regression functions, Algorithm 1 constructs a lower bound $\mathsf{lcb}(x; \mathcal{F}_m) := \inf_{f \in \mathcal{F}_m} f(x)$ and an upper bound $\mathsf{ucb}(x; \mathcal{F}_m) := \sup_{f \in \mathcal{F}_m} f(x)$ for the true conditional probability $\eta(x)$. An empirical classifier $\widehat{h}_m : \mathcal{X} \to \{+1, -1, \perp\}$ and a query function $g_m : \mathcal{X} \to \{0, 1\}$ are then constructed based on these confidence ranges and the abstention parameter $\gamma$. For any time step $t$ within epoch $m$, Algorithm 1 queries the label of the observed data point $x_t$ if and only if $Q_t := g_m(x_t) = 1$. Algorithm 1 returns $\widehat{h}_M$ as the learned classifier.

We now discuss the empirical classifier $\widehat{h}_m$ and the query function $g_m$ in more detail. Consider the event where $f^\star \in \mathcal{F}_m$ for all $m \in [M]$, which can be shown to hold with high probability. The constructed confidence intervals are valid under this event, i.e., $\eta(x) \in [\mathsf{lcb}(x; \mathcal{F}_m), \mathsf{ucb}(x; \mathcal{F}_m)]$. First, let us examine the conditions that determine a label query. The label of $x$ is *not* queried if

- **Case 1:** $\widehat{h}_m(x) = \perp$**.** We have $\eta(x) \in [\mathsf{lcb}(x; \mathcal{F}_m), \mathsf{ucb}(x; \mathcal{F}_m)] \subseteq [\frac{1}{2} - \gamma, \frac{1}{2} + \gamma]$. Abstention leads to the smallest error (Herbei and Wegkamp, 2006), and no query is needed.

- **Case 2:** $\frac{1}{2} \notin (\mathsf{lcb}(x; \mathcal{F}_m), \mathsf{ucb}(x; \mathcal{F}_m))$**.** We have $\mathrm{sign}(2\widehat{f}_m(x) - 1) = \mathrm{sign}(2f^\star(x) - 1)$. Thus, no excess error is incurred and there is no need to query.

The only case when label query *is* issued, and thus when the classifier $\widehat{h}_m$ may suffer from excess error, is when

$$\frac{1}{2} \in (\mathsf{lcb}(x; \mathcal{F}_m), \mathsf{ucb}(x; \mathcal{F}_m)) \quad \text{and} \quad [\mathsf{lcb}(x; \mathcal{F}_m), \mathsf{ucb}(x; \mathcal{F}_m)] \nsubseteq \left[ \frac{1}{2} - \gamma, \frac{1}{2} + \gamma \right] \qquad (3)$$

hold simultaneously. Eq. (3) necessarily leads to the condition $w(x; \mathcal{F}_m) := \mathsf{ucb}(x; \mathcal{F}_m) - \mathsf{lcb}(x; \mathcal{F}_m) > \gamma$. Our theoretical analysis shows that the event must $\mathbb{1}(w(x; \mathcal{F}_m) > \gamma)$ happens in-

frequently, and its frequency is closely related to the so-called *value function disagreement coefficient* (Foster et al., 2020), which we introduce as follows.[2]

**Definition 1** (Value function disagreement coefficient). *For any $f^\star \in \mathcal{F}$ and $\gamma_0, \varepsilon_0 > 0$, the value function disagreement coefficient $\theta_{f^\star}^{\mathrm{val}}(\mathcal{F}, \gamma_0, \varepsilon_0)$ is defined as*

$$\sup_{\mathcal{D}_{\mathcal{X}}} \sup_{\gamma > \gamma_0, \varepsilon > \varepsilon_0} \left\{ \frac{\gamma^2}{\varepsilon^2} \cdot \mathbb{P}_{\mathcal{D}_{\mathcal{X}}} \left( \exists f \in \mathcal{F} : |f(x) - f^\star(x)| > \gamma, \|f - f^\star\|_{\mathcal{D}_{\mathcal{X}}} \le \varepsilon \right) \right\} \vee 1,$$

*where $\|f\|_{\mathcal{D}_{\mathcal{X}}}^2 := \mathbb{E}_{x \sim \mathcal{D}_{\mathcal{X}}}[f^2(x)]$.*

Combining the insights discussed above, we derive the following label complexity guarantee for Algorithm 1 (we use $\theta := \sup_{f^\star \in \mathcal{F}, \iota > 0} \theta_{f^\star}^{\mathrm{val}}(\mathcal{F}, \gamma/2, \iota)$ and discuss its boundedness below). [3]

**Theorem 4.** *With probability at least $1 - 2\delta$, Algorithm 1 returns a classifier with Chow's excess error at most $\varepsilon$ and label complexity $O(\frac{\theta \operatorname{Pdim}(\mathcal{F})}{\gamma^2} \cdot \log^2(\frac{\theta \operatorname{Pdim}(\mathcal{F})}{\varepsilon \gamma}) \cdot \log(\frac{\theta \operatorname{Pdim}(\mathcal{F})}{\varepsilon \gamma \delta}))$.*

Theorem 4 shows that Algorithm 1 achieves exponential label savings (i.e., $\operatorname{polylog}(\frac{1}{\varepsilon})$) without any low noise assumptions. We discuss the result in more detail next.

- **Boundedness of $\theta$.** The value function disagreement coefficient is well-bounded for many function classes of practical interests. For instance, we have $\theta \le d$ for linear functions on $\mathbb{R}^d$ and $\theta \le C_{\mathsf{link}} \cdot d$ for generalized linear functions (where $C_{\mathsf{link}}$ is a quantity related to the link function). Moreover, $\theta$ is *always* upper bounded by complexity measures such as (squared) star number and eluder dimension (Foster et al., 2020). See Appendix C for the detailed definitions/bounds.

- **Comparison to Puchkin and Zhivotovskiy (2021).** The label complexity bound derived in Theorem 4 is similar to the one derived in Puchkin and Zhivotovskiy (2021), with one critical difference: The bound derived in Puchkin and Zhivotovskiy (2021) is in terms of *classifier-based* disagreement coefficient $\check{\theta}$ (Hanneke, 2014). Even in the case with linear classifiers, $\check{\theta}$ is only known to be bounded under additional assumptions, e.g., when $\mathcal{D}_{\mathcal{X}}$ is uniform over the unit sphere.

**Computational efficiency.** We discuss how to efficiently implement Algorithm 1 with the regression oracle defined in Eq. (1). [4] Our implementation relies on subroutines developed in Krishnamurthy et al. (2017); Foster et al. (2018), which allow us to approximate confidence bounds $\mathrm{ucb}(x; \mathcal{F}_m)$ and $\mathrm{lcb}(x; \mathcal{F}_m)$ up to $\alpha$ approximation error with $O(\frac{1}{\alpha^2} \log \frac{1}{\alpha})$ (or $O(\log \frac{1}{\alpha})$ when $\mathcal{F}$ is convex and closed under pointwise convergence) calls to the regression oracle. To achieve the same theoretical guarantees shown in Theorem 4 (up to changes in constant terms), we show that it suffices to (i) control the approximation error at level $O(\frac{\gamma}{\log T})$, (ii) construct the approximated confidence bounds $\widehat{\mathrm{lcb}}(x; \mathcal{F}_m)$ and $\widehat{\mathrm{ucb}}(x; \mathcal{F}_m)$ in a way such that the confidence region is non-increasing with respect to the epoch $m$, i.e., $(\widehat{\mathrm{lcb}}(x; \mathcal{F}_m), \widehat{\mathrm{ucb}}(x; \mathcal{F}_m)) \subseteq (\widehat{\mathrm{lcb}}(x; \mathcal{F}_{m-1}), \widehat{\mathrm{ucb}}(x; \mathcal{F}_{m-1}))$ (this ensures that the sampling region is non-increasing even with *approximated* confidence bounds, which is important to our theoretical analysis), and (iii) use the approximated confidence bounds $\widehat{\mathrm{lcb}}(x; \mathcal{F}_m)$ and $\widehat{\mathrm{ucb}}(x; \mathcal{F}_m)$ to construct the classifier $\widehat{h}_m$ and the query function $g_m$. We provide our guarantees as follows, and leave details to Appendix E (we redefine $\theta := \sup_{f^\star \in \mathcal{F}, \iota > 0} \theta_{f^\star}^{\mathrm{val}}(\mathcal{F}, \gamma/4, \iota)$ in the Theorem 5 to account to approximation error).

**Theorem 5.** *Algorithm 1 can be efficiently implemented via the regression oracle and enjoys the same theoretical guarantees stated in Theorem 4. The number of oracle calls needed is $\widetilde{O}(\frac{\theta \operatorname{Pdim}(\mathcal{F})}{\varepsilon \gamma^3})$*

---

[2]Compared to the original definition studied in contextual bandits (Foster et al., 2020), our definition takes an additional "sup" over all possible marginal distributions $\mathcal{D}_{\mathcal{X}}$ to account for *distributional shifts* incurred by selective querying (which do not occur in contextual bandits). Nevertheless, as we show below, our disagreement coefficient is still well-bounded for many important function classes.

[3]It suffices to take $\theta := \theta_{f^\star}^{\mathrm{val}}(\mathcal{F}, \gamma/2, \iota)$ with $\iota \propto \sqrt{\gamma \varepsilon}$ to derive a slightly different guarantee. See Appendix E.

[4]Recall that the implementation of the regression oracle should be viewed as an efficient operation since it solves a convex optimization problem with respect to the regression function, and it even admits closed-form solutions in many cases, e.g., it is reduced to least squares when $f$ is linear. On the other hand, the ERM oracle used in Puchkin and Zhivotovskiy (2021) is NP-hard even for a set of linear classifiers (Guruswami and Raghavendra, 2009).

*for a general set of regression functions $\mathcal{F}$, and $\widetilde{O}(\frac{\theta \operatorname{Pdim}(\mathcal{F})}{\varepsilon \gamma})$ when $\mathcal{F}$ is convex and closed under pointwise convergence. The per-example inference time of the learned $\widehat{h}_M$ is $\widetilde{O}(\frac{1}{\gamma^2} \log^2(\frac{\theta \operatorname{Pdim}(\mathcal{F})}{\varepsilon}))$ for general $\mathcal{F}$, and $\widetilde{O}(\log \frac{1}{\gamma})$ when $\mathcal{F}$ is convex and closed under pointwise convergence.*

With Theorem 5, we provide the first computationally efficient active learning algorithm that achieves exponential label savings, without any low noise assumptions.

## 3 Guarantees under standard excess error

We provide guarantees for Algorithm 1 under *standard* excess error. In Section 3.1, we show that Algorithm 1 can be used to recover the usual minimax label complexity under Massart/Tsybakov noise; we also provide a new learning paradigm based on Algorithm 1 under limited budget. In Section 3.2, we show that Algorithm 1 provably avoid the undesired *noise-seeking* behavior often seen in active learning.

### 3.1 Recovering minimax optimal label complexity

One way to convert an abstaining classifier $\widehat{h} : \mathcal{X} \to \mathcal{Y} \cup \{\bot\}$ into a standard classifier $\check{h} : \mathcal{X} \to \mathcal{Y}$ is by randomizing the prediction in its abstention region, i.e., if $\widehat{h}(x) = \bot$, then its randomized version $\check{h}(x)$ predicts $+1/-1$ with equal probability (Puchkin and Zhivotovskiy, 2021). With such randomization, the *standard excess error* of $\check{h}$ can be characterized as

$$\operatorname{err}(\check{h}) - \operatorname{err}(h^\star) = \operatorname{err}_\gamma(\widehat{h}) - \operatorname{err}(h^\star) + \gamma \cdot \mathbb{P}_{x \sim \mathcal{D}_\mathcal{X}}(\widehat{h}(x) = \bot). \tag{4}$$

The standard excess error depends on the (random) abstention region of $\widehat{h}$, which is difficult to quantify in general. To give a more practical characterization of the standard excess error, we introduce the concept of proper abstention in the following.

**Definition 2** (Proper abstention). *A classifier $\widehat{h} : \mathcal{X} \to \mathcal{Y} \cup \{\bot\}$ enjoys proper abstention if and only if it abstains in regions where abstention is indeed the optimal choice, i.e., $\{x \in \mathcal{X} : \widehat{h}(x) = \bot\} \subseteq \{x \in \mathcal{X} : \eta(x) \in [\frac{1}{2} - \gamma, \frac{1}{2} + \gamma]\} =: \mathcal{X}_\gamma$.*

**Proposition 2.** *The classifier $\widehat{h}$ returned by Algorithm 1 enjoys proper abstention. With randomization over the abstention region, we have the following upper bound on its standard excess error*

$$\operatorname{err}(\check{h}) - \operatorname{err}(h^\star) \leq \operatorname{err}_\gamma(\widehat{h}) - \operatorname{err}(h^\star) + \gamma \cdot \mathbb{P}_{x \sim \mathcal{D}_\mathcal{X}}(x \in \mathcal{X}_\gamma). \tag{5}$$

The proper abstention property of $\widehat{h}$ returned by Algorithm 1 is achieved via conservation: $\widehat{h}$ will avoid abstention unless it is absolutely sure that abstention is the optimal choice.[5] To characterize the standard excess error of classifier with proper abstention, we only need to upper bound the term $\mathbb{P}_{x \sim \mathcal{D}_\mathcal{X}}(x \in \mathcal{X}_\gamma)$, which does *not* depends on the (random) classifier $\widehat{h}$. Instead, it only depends on the marginal distribution. We next introduce the common Massart/Tsybakov noise conditions.

**Definition 3** (Massart noise, Massart and Nédélec (2006)). *A distribution $\mathcal{D}_{\mathcal{X}\mathcal{Y}}$ satisfies the Massart noise condition with parameter $\tau_0 > 0$ if $\mathbb{P}_{x \sim \mathcal{D}_\mathcal{X}}(|\eta(x) - 1/2| \leq \tau_0) = 0$.*

**Definition 4** (Tsybakov noise, Tsybakov (2004)). *A distribution $\mathcal{D}_{\mathcal{X}\mathcal{Y}}$ satisfies the Tsybakov noise condition with parameter $\beta \geq 0$ and a universal constant $c > 0$ if $\mathbb{P}_{x \sim \mathcal{D}_\mathcal{X}}(|\eta(x) - 1/2| \leq \tau) \leq c\tau^\beta$ for any $\tau > 0$.*

As in Balcan et al. (2007); Hanneke (2014), we assume knowledge of noise parameters (e.g., $\tau_0, \beta$). Together with the active learning lower established in Castro and Nowak (2006, 2008), and focusing on the dependence of $\varepsilon$, our next theorem shows that Algorithm 1 can be used to recover the minimax label complexity in active learning, under the *standard* excess error.

---

[5]On the other hand, however, the algorithm provided in Puchkin and Zhivotovskiy (2021) is very unlikely to have such property. In fact, only a small but *nonzero* upper bound of abstention rate is provided (Proposition 3.6 therein) under the Massart noise with $\gamma \leq \frac{\tau_0}{2}$; yet any classifier that enjoys proper abstention should have exactly zero abstention rate.

**Theorem 6.** *With an appropriate choice of the abstention parameter $\gamma$ in Algorithm 1 and randomization over the abstention region, Algorithm 1 learns a classifier $\check{h}$ at the minimax optimal rates: To achieve $\varepsilon$ standard excess error, it takes $\widetilde{\Theta}(\tau_0^{-2})$ labels under Massart noise and takes $\widetilde{\Theta}(\varepsilon^{-2/(1+\beta)})$ labels under Tsybakov noise.*

**Remark 2.** *In addition to recovering the minimax rates, the proper abstention property is desirable in practice: It guarantees that $\widehat{h}$ will not abstain on easy examples, i.e., it will not mistakenly flag easy examples as "hard-to-classify", thus eliminating unnecessary human labeling efforts.*

Algorithm 1 can also be used to provide new learning paradigms in the limited budget setting, which we introduce below. No prior knowledge of noise parameters are required in this setup.

**New learning paradigm under limited budget.** Given any labeling budget $B > 0$, we can then choose $\gamma \approx B^{-1/2}$ in Algorithm 1 to make sure the label complexity is never greater than $B$ (with high probability). The learned classifier enjoys Chow's excess error (with parameter $\gamma$) at most $\varepsilon$; its standard excess error (with randomization over the abstention region) can be analyzed by relating the $\gamma \cdot \mathbb{P}_{x \sim \mathcal{D}_{\mathcal{X}}}(x \in \mathcal{X}_\gamma)$ term in Eq. (5) to the Massart/Tsybakov noise conditions, as discussed above.

## 3.2 Abstention to avoid noise-seeking

Active learning algorithms sometimes exhibit *noise-seeking* behaviors, i.e., oversampling in regions where $\eta(x)$ is close to the $\frac{1}{2}$ level. Such noise-seeking behavior is known to be a fundamental barrier to achieve low label complexity (under standard excess error), e.g., see Kääriäinen (2006). We show in this section that abstention naturally helps avoiding noise-seeking behaviors and speeding up active learning.

To better illustrate how properly abstaining classifiers avoid noise-seeking behavior, we first propose new noise conditions below, which strictly generalize the usual Massart/Tsybakov noises.

**Definition 5** (Noise-seeking Massart noise). *A distribution $\mathcal{D}_{\mathcal{X}\mathcal{Y}}$ satisfies the noise-seeking Massart noise condition with parameters $0 \leq \zeta_0 < \tau_0 \leq 1/2$ if $\mathbb{P}_{x \sim \mathcal{D}_{\mathcal{X}}}(\zeta_0 < |\eta(x) - 1/2| \leq \tau_0) = 0$.*

**Definition 6** (Noise-seeking Tsybakov noise). *A distribution $\mathcal{D}_{\mathcal{X}\mathcal{Y}}$ satisfies the noise-seeking Tsybakov noise condition with parameters $0 \leq \zeta_0 < 1/2$, $\beta \geq 0$ and a universal constant $c > 0$ if $\mathbb{P}_{x \sim \mathcal{D}_{\mathcal{X}}}(\zeta_0 < |\eta(x) - 1/2| \leq \tau) \leq c\tau^\beta$ for any $\tau > \zeta_0$.*

Compared to the standard Massart/Tsybakov noises, these newly proposed noise-seeking conditions allow arbitrary probability mass of data points whose conditional probability $\eta(x)$ is equal/close to $1/2$. As a result, they can trick standard active learning algorithms into exhibiting the noise-seeking bahaviors (and hence their names). We also mention that the parameter $\zeta_0$ should be considered as an *extremely small quantity* (e.g., $\zeta_0 \ll \varepsilon$), with the extreme case corresponding to $\zeta_0 = 0$ (which still allow arbitrary probability for region $\{x \in \mathcal{X} : \eta(x) = 1/2\}$).

Ideally, any active learning algorithm should not be heavily affected by these noise conditions since it hardly matters (in terms of excess error) what label is predicted over region $\{x \in \mathcal{X} : |\eta(x) - 1/2| \leq \zeta_0\}$. However, these seemingly benign noise-seeking conditions can cause troubles for any "uncertainty-based" active learner, i.e., any active learning algorithm that requests the label for any point that is uncertain (see Definition 10 in Appendix F for formal definition). In particular, under limited budget, we derive the following result.

**Proposition 3.** *Fix $\varepsilon, \delta, \gamma > 0$. For any labeling budget $B \gtrsim \frac{1}{\gamma^2} \cdot \log^2(\frac{1}{\varepsilon\gamma}) \cdot \log(\frac{1}{\varepsilon\gamma\delta})$, there exists a learning problem (with a set of linear regression functions) satisfying Definition 5/Definition 6 such that (1) any "uncertainty-based" active learner suffers expected standard excess error $\Omega(B^{-1})$; yet (2) with probability at least $1 - \delta$, Algorithm 1 returns a classifier with standard excess error at most $\varepsilon$.*

The above result demonstrates the superiority of our Algorithm 1 over any "uncertainty-based" active learner. Moreover, we show that Algorithm 1 achieves similar guarantees as in Theorem 6 under the strictly harder noise-seeking conditions. Specifically, we have the following guarantees.

**Theorem 7.** *With an appropriate choice of the abstention parameter $\gamma$ in Algorithm 1 and randomization over the abstention region, Algorithm 1 learns a classifier $\check{h}$ with $\varepsilon + \zeta_0$ standard excess error after querying $\widetilde{\Theta}(\tau_0^{-2})$ labels under Definition 5 or querying $\widetilde{\Theta}(\varepsilon^{-2/(1+\beta)})$ labels under Definition 6.*

The special case of the noise-seeking condition with $\zeta_0 = 0$ is recently studied in (Kpotufe et al., 2021), where the authors conclude that no active learners can outperform the passive counterparts in the *nonparametric* regime. Theorem 7 shows that, in the *parametric* setting (with function approximation), Algorithm 1 provably overcomes these noise-seeking conditions.

## 4 Extensions

We provide two adaptations of our main algorithm (Algorithm 1) that can (1) achieve constant label complexity for a general set of regression functions (Section 4.1); and (2) adapt to model misspecification (Section 4.2). These two adaptations can also be efficiently implemented via regression oracle and enjoy similar guarantees stated in Theorem 5. We defer computational analysis to Appendix G and Appendix H.

### 4.1 Constant label complexity

We start by considering a simple problem instance with $\mathcal{X} = \{x\}$, where active learning is reduced to mean estimation of $\eta(x)$. Consider the Massart noise case where $\eta(x) \notin [\frac{1}{2} - \tau_0, \frac{1}{2} + \tau_0]$. No matter how small the desired accuracy level $\varepsilon > 0$ is, the learner should not spend more than $O(\frac{\log(1/\delta)}{\tau_0^2})$ labels to correctly classify $x$ with probability at least $1 - \delta$, which ensures 0 excess error. In the general setting, but with Chow's excess error, a similar result follows: It takes at most $O(\frac{\log(1/\delta)}{\gamma^2})$ samples to verify if $\eta(x)$ is contained in $[\frac{1}{2} - \gamma, \frac{1}{2} + \gamma]$ or not. Taking the optimal action within $\{+1, -1, \perp\}$ (based on Eq. (2)) then leads to 0 Chow's excess error. This reasoning shows that, at least in simple cases, one should be able to achieve *constant* label complexity no matter how small $\varepsilon$ is. One natural question to ask is as follows.

*Is it possible to achieve constant label complexity in the general case of active learning?*

We provide the first affirmative answer to the above question with a *general* set of regression function $\mathcal{F}$ (finite), and under *general* action space $\mathcal{X}$ and marginal distribution $\mathcal{D}_{\mathcal{X}}$. The positive result is achieved by Algorithm 2 (deferred to Appendix G.2), which differs from Algorithm 1 in two aspects: (1) we drop the epoch scheduling, and (2) apply a tighter elimination step derived from an optimal stopping theorem. Another change comes from the analysis of the algorithm: Instead of analyzing with respect to the disagreement coefficient, we work with the *eluder dimension* $\mathfrak{e} := \sup_{f^\star \in \mathcal{F}} \mathfrak{e}_{f^\star}(\mathcal{F}, \gamma/2)$.[6] To do that, we analyze active learning from the perspective of *regret minimization with selective querying* (Dekel et al., 2012), which allows us to incorporate techniques developed in the field of contextual bandits (Russo and Van Roy, 2013; Foster et al., 2020). We defer a detailed discussion to Appendix G.1 and provide the following guarantees.

**Theorem 8.** *With probability at least $1 - 2\delta$, Algorithm 2 returns a classifier with expected Chow's excess error at most $\varepsilon$ and label complexity $O(\frac{\mathfrak{e} \cdot \log(|\mathcal{F}|/\delta)}{\gamma^2})$, which is independent of $\frac{1}{\varepsilon}$.*

Based on discussion in Section 3, we can immediately translate the above results into *standard* excess error guarantees under the Massart noise (with $\gamma$ replaced by $\tau_0$). We next discuss why existing algorithms/analyses do not guarantee constant label complexity, even in the linear case.

1. **Epoch scheduling.** Many algorithms proceed in epochs and aim at *halving* the excess error after each epoch (Balcan et al., 2007; Zhang and Chaudhuri, 2014; Puchkin and Zhivotovskiy, 2021). One inevitably needs $\log \frac{1}{\varepsilon}$ epochs to achieve $\varepsilon$ excess error.

2. **Relating to disagreement coefficient.** The algorithm presented in Krishnamurthy et al. (2019) does not use epoch scheduling. However, their label complexity are analyzed with disagreement coefficient, which incurs a $\sum_{t=1}^{1/\varepsilon} \frac{1}{t} = O(\log \frac{1}{\varepsilon})$ term in the label complexity.

**Remark 3.** *Algorithm 2 also provides guarantees when $x$ is selected by an adaptive adversary (instead of i.i.d. sampled $x \sim \mathcal{D}_{\mathcal{X}}$). In that case, we simultaneously upper bound the regret and the label complexity (see Theorem 10 in Appendix G.2). Our results can be viewed as a generalization of the results developed in the linear case (Dekel et al., 2012).*

---

[6]We formally define eluder dimension in Appendix C. As examples, we have $\mathfrak{e} = O(d \cdot \log \frac{1}{\gamma})$ for linear functions in $\mathbb{R}^d$, and $\mathfrak{e} = O(C_{\text{link}} \cdot d \log \frac{1}{\gamma})$ for generalized linear functions (where $C_{\text{link}}$ is a quantity related to the link function).

## 4.2 Dealing with model misspecification

Our main results are developed under realizability (Assumption 1), which assumes that there exists a $f^\star \in \mathcal{F}$ such that $f^\star = \eta$. In this section, we relax that assumption and allow model misspecification. We assume the learner is given a set of regression function $\mathcal{F} : \mathcal{X} \to [0, 1]$ that may only *approximates* the conditional probability $\eta$. More specifically, we make the following assumption.

**Assumption 2** (Model misspecification). *There exists a $\overline{f} \in \mathcal{F}$ such that $\overline{f}$ approximate $\eta$ up to $\kappa > 0$ accuracy, i.e., $\sup_{x \in \mathcal{X}} |\overline{f}(x) - \eta(x)| \le \kappa$.*

We use a variation of Algorithm 1 to adapt to model misspecification (Algorithm 3, deferred to Appendix H.1). Compared to Algorithm 1, the main change in Algorithm 3 is to apply a more conservative step in determining the active set $\mathcal{F}_m$ at each epoch: We maintain a larger active set of regression function to ensure that $\overline{f}$ is not eliminated throughout all epochs. Our algorithm proceeds *without* knowing the misspecification level $\kappa$. However, the excess error bound presented next holds under the condition that $\kappa \le \varepsilon$ (i.e., it requires that the misspecification is no larger than the desired accuracy). Abbreviate $\overline{\theta} := \sup_{\iota > 0} \theta_{\overline{f}}^{\mathrm{val}}(\mathcal{F}, \gamma/2, \iota)$, we achieve the following guarantees.

**Theorem 9.** *Suppose $\kappa \le \varepsilon$. With probability at least $1 - 2\delta$, Algorithm 3 returns a classifier with Chow's excess error $O(\varepsilon \cdot \overline{\theta} \cdot \log(\frac{\mathrm{Pdim}(\mathcal{F})}{\varepsilon \gamma \delta}))$ and label complexity $O(\frac{\overline{\theta} \, \mathrm{Pdim}(\mathcal{F})}{\gamma^2} \cdot \log^2(\frac{\mathrm{Pdim}(\mathcal{F})}{\varepsilon \gamma}) \cdot \log(\frac{\mathrm{Pdim}(\mathcal{F})}{\varepsilon \gamma \delta}))$.*

We only provide guarantee when $\kappa \le \varepsilon$, since the learned classifier suffers from an additive $\kappa$ term in the excess error (see Appendix H.2 for more discussion). On the other hand, the (inefficient) algorithm provided in Puchkin and Zhivotovskiy (2021) works without any assumption on the approximation error. An interesting future direction is to study the relation between computational efficiency and learning with *general* approximation error.

## Acknowledgments and Disclosure of Funding

We thank the anonymous reviewers for their helpful comments. This work is partially supported by NSF grant 1934612 and AFOSR grant FA9550-18-1-0166.

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
