# A    Additional related work

Learning with Chow's excess error is closely related to learning under Massart noise (Massart and Nédélec, 2006), which assumes that no data point has conditional expectation close to the decision boundary, i.e., $\mathbb{P}(|\eta(x) - 1/2| \leq \tau_0) = 0$ for some constant $\tau_0 > 0$. Learning with Massart noise is commonly studied in active learning (Balcan et al., 2007; Hanneke, 2014; Zhang and Chaudhuri, 2014; Krishnamurthy et al., 2019), where $\widetilde{O}(\tau_0^{-2})$ type of guarantees are achieved. Instead of making explicit assumptions on the underlying distribution, learning with Chow's excess error empowers the learner with the ability to abstain: There is no need to make predictions on hard data points that are close to the decision boundary, i.e., $\{x : |\eta(x) - 1/2| \leq \gamma\}$. Learning with Chow's excess error thus works on more general settings and still enjoys the $\widetilde{O}(\gamma^{-2})$ type of guarantee as learning under Massart noise (Puchkin and Zhivotovskiy, 2021).[7] We show in Section 3 that statistical guarantees achieved under Chow's excess error can be directly translated to guarantees under (usual and more challenging versions of) Massart/Tsybakov noise (Massart and Nédélec, 2006; Tsybakov, 2004).

Active learning at aim competing the best in-class classifier with few labels. A long line of work directly works with the set of classifiers (Balcan et al., 2007; Hanneke, 2007, 2014; Huang et al., 2015; Puchkin and Zhivotovskiy, 2021), where the algorithms are developed with (in general) hard-to-implement ERM oracles (Guruswami and Raghavendra, 2009) and the the guarantees dependence on the so-called disagreement coefficient (Hanneke, 2014). More recently, learning with function approximation have been studied inactive learning and contextual bandits (Dekel et al., 2012; Agarwal et al., 2012; Foster et al., 2018; Krishnamurthy et al., 2019). The function approximation scheme permits efficient regression oracles, which solve convex optimization problems with respect to regression functions (Krishnamurthy et al., 2017, 2019; Foster et al., 2018). It can also be analyzed with the scale-sensitive version of disagreement coefficient, which is usually tighter than the original one (Foster et al., 2020; Russo and Van Roy, 2013). Our algorithms are inspired Krishnamurthy et al. (2019), where the authors study active learning under the standard excess error. The main deviation from Krishnamurthy et al. (2019) is that we need to *manually* construct a classifier $\widehat{h}$ with an abstention option and $\widehat{h} \notin \mathcal{H}$, which leads to differences in the analysis of excess error and label complexity. We borrow techniques developed in contextual bandits Russo and Van Roy (2013); Foster et al. (2020) to analyze our algorithm.

Although one can also apply our algorithms in the nonparametric regime with proper pre-processing schemes such discretizations, our algorithm primarily works in the parametric setting with finite pseudo dimension (Haussler, 1995) and finite (value function) disagreement coefficient (Foster et al., 2020). Active learning has also been studied in the nonparametric regime (Castro and Nowak, 2008; Koltchinskii, 2010; Minsker, 2012; Locatelli et al., 2017). Notably, Shekhar et al. (2021) studies Chow's excess error with margin-type of assumptions. Their setting is different to ours and $\text{poly}(\frac{1}{\varepsilon})$ label complexities are achieved. If abundant amounts of data points are allowed to be exactly at the decision boundary, i.e., $\eta(x) = \frac{1}{2}$, Kpotufe et al. (2021) recently shows that, in the nonparametric regime, no active learner can outperform the passive counterpart.

# B    Why Chow's excess error helps?

For illustration purpose, we focus on the simple case where $\mathcal{X} = \{x\}$ in this section. The active learning problem is then reduced to mean estimation of the conditional probability $\eta(x) \in [0, 1]$.

**Learning with standard excess error.**    Fix any $\varepsilon > 0$. With respect to the conditional probability $\eta(x)$, we define the positive region $\mathcal{S}_{+,\varepsilon} := \left[\frac{1-\varepsilon}{2}, 1\right]$ and the negative region $\mathcal{S}_{-,\varepsilon} := [0, \frac{1+\varepsilon}{2}]$. These regions are interpreted as follows: If $\eta(x) \in \mathcal{S}_{+,\varepsilon}$ (resp. $\eta(x) \in \mathcal{S}_{-,\varepsilon}$), then labeling $x$ as positive (resp. negative) incurs no more than $\varepsilon$ excess error. Under standard excess error, we define the flexible region as $\mathcal{S}_{\text{flexible},\varepsilon}^{\text{standard}} := \mathcal{S}_{+,\varepsilon} \cap \mathcal{S}_{-,\varepsilon} = [\frac{1-\varepsilon}{2}, \frac{1+\varepsilon}{2}]$ (the grey area in the top plot in Fig. 1). Two important implications of the flexible region are as follows: (1) if $\eta(x) \in \mathcal{S}_{\text{flexible},\varepsilon}^{\text{standard}}$, labeling $x$ as either positive or negative would lead to excess error at most $\varepsilon$; and (2) if $\eta(x) \notin \mathcal{S}_{\text{flexible},\varepsilon}^{\text{standard}}$, then a classifier must correctly label $x$ as either positive or negative to guarantee $\varepsilon$ excess error. Since the

---

[7]However, passive learning with abstention only achieves error rate $\frac{1}{n\gamma}$ with $n$ samples (Bousquet and Zhivotovskiy, 2021).

flexible region is of length $\varepsilon$ under standard excess error, distinguishing two points at the edge of the flexible region, e.g., $\eta(x) = \frac{1}{2} - \varepsilon$ and $\eta(x) = \frac{1}{2} + \varepsilon$, leads to label complexity lower bound $\Omega(\frac{1}{\varepsilon^2})$.

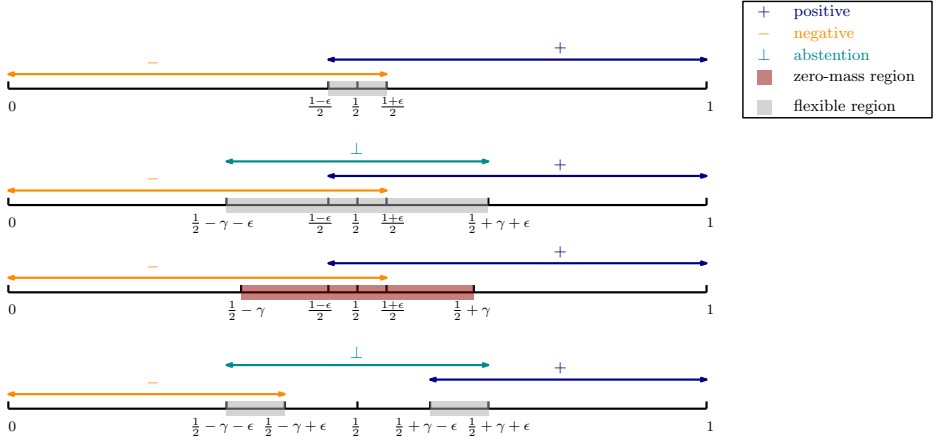

Figure 1: Illustration of regions $\mathcal{S}_{+,\varepsilon}$, $\mathcal{S}_{-,\varepsilon}$ and $\mathcal{S}_{\perp,\varepsilon}$. (1) top plot: learning with standard excess error $\mathrm{err}(\widehat{h}) - \mathrm{err}(h^\star)$; (2) second plot: learning with standard excess error $\mathrm{err}(\widehat{h}) - \mathrm{err}(h^\star)$ and Massart noise with parameter $\gamma$; (3) third plot: learning with Chow's excess error $\mathrm{err}_\gamma(\widehat{h}) - \mathrm{err}(h^\star)$; and (4) bottom plot: learning against the optimal Chow's excess error, i.e., $\mathrm{err}_\gamma(\widehat{h}) - \inf_{h:\mathcal{X}\to\{+1,-1,\perp\}} \mathrm{err}_\gamma(h)$.

**Learning with Chow's excess error.** Now we turn our attention to the Chow's error. We consider $\mathcal{S}_{+,\varepsilon}$ and $\mathcal{S}_{-,\varepsilon}$ as before, and define a third abstention region $\mathcal{S}_{\perp,\varepsilon} := [\frac{1}{2} - \gamma - \varepsilon, \frac{1}{2} + \gamma + \varepsilon]$ If $\eta(x) \in \mathcal{S}_{\perp,\varepsilon}$, then abstaining on $x$ leads to Chow's excess error at most $\varepsilon$. We define the positive flexible region as $\mathcal{S}_{\text{flexible},+,\varepsilon}^{\text{Chow}} := \mathcal{S}_{\perp,\varepsilon} \cap \mathcal{S}_{\perp,\varepsilon} = [\frac{1-\varepsilon}{2}, \frac{1}{2} + \gamma + \varepsilon]$, which is of length $\gamma + \frac{3\varepsilon}{2}$. The negative flexible region is defined similarly (see the second plot in Fig. 1). Since both positive and negative flexible regions are longer than $\gamma$, it takes at most $\widetilde{O}(\frac{1}{\gamma^2})$ samples to identify a labeling action with at most $\varepsilon$ Chow's excess error. Without loss of generality, we assume $\eta(x) < \frac{1}{2}$. It takes $\widetilde{O}(\frac{1}{\gamma^2})$ samples to construct a confidence interval $[\mathsf{lcb}(x), \mathsf{ucb}(x)]$ (of $\eta(x)$) of length at most $\frac{\gamma}{2}$. If $\eta(x) < \frac{1-\gamma}{2}$, we observe $\mathsf{ucb}(x) \leq \frac{1}{2}$ and know that labeling $x$ negative ensures at most $\varepsilon$ excess error. If $\eta(x) \in [\frac{1-\gamma}{2}, \frac{1}{2}]$, we observe $[\mathsf{lcb}(x), \mathsf{ucb}(x)] \subseteq [\frac{1}{2} - \gamma, \frac{1}{2} + \gamma] \subseteq \mathcal{S}_{\perp,\varepsilon}$. Thus, choosing to abstain on $x$ again ensures at most $\varepsilon$ excess error. To summarize, learning with Chow's excess error in the general case resembles the behavior of learning under Massart noise: Hard examples that are close to the decision boundary are not selected for labeling (see the third plot in Fig. 1), and learning is possible with $\widetilde{O}(\frac{1}{\gamma^2})$ labels.

**Why not compete against the optimal Chow's error.** As shown in the last plot in Fig. 1, the flexible regions become narrow (of length $O(\varepsilon)$) again when competing against the optimal Chow's error. Abstention becomes the only action that guarantees at most $\varepsilon$ excess error over region $(\frac{1}{2} - \gamma - \varepsilon, \frac{1}{2} + \gamma + \varepsilon)$. One then needs to distinguish cases $\eta(x) = \frac{1}{2} + \gamma - 2\varepsilon$ and $\eta(x) = \frac{1}{2} + \gamma + 2\varepsilon$, which requires $\Omega(\frac{1}{\varepsilon^2})$ samples. It is also unreasonable to compete against the optimal Chow's error: When $\eta(x) = \frac{1}{2} + \gamma - 2\varepsilon$, $\Omega(\frac{1}{\varepsilon^2})$ samples are required to decide whether take action $+1$ or action $\perp$; however, with only $\widetilde{O}(\frac{1}{\gamma^2})$ samples, one can already guarantee $\eta(x) > \frac{1}{2}$ (and take the positive label in the usual case).

# C  Disagreement coefficient, star number and eluder dimension

We provide formal definitions/guarantees of value function disagreement coefficient, eluder dimension and star number in this section. These results are developed in Foster et al. (2020); Russo and Van Roy (2013). Since our guarantees are developed in terms of these complexity measures, any future developments on these complexity measures (e.g., with respect to richer function classes) directly lead to broader applications of our algorithms.

We first state known upper bound on value function disagreement coefficient with respect to nice sets of regression functions.

**Proposition 4** (Foster et al. (2020)). *For any $f^\star \in \mathcal{F}$ and $\gamma, \varepsilon > 0$, let $\theta^{\mathrm{val}}_{f^\star}(\mathcal{F}, \gamma, \varepsilon)$ be the value function disagreement coefficient defined in Definition 1. Let $\phi : \mathcal{X} \to \mathbb{R}^d$ be a fixed feature mapping and $\mathcal{W} \subseteq \mathbb{R}^d$ be a fixed set. The following upper bounds hold true.*

- *Suppose $\mathcal{F} := \{x \mapsto \langle \phi(x), w \rangle : w \in \mathcal{W}\}$ is a set of linear functions. We then have $\sup_{f \in \mathcal{F}, \gamma > 0, \varepsilon > 0} \theta^{\mathrm{val}}_f(\mathcal{F}, \gamma, \varepsilon) \le d$.*

- *Suppose $\mathcal{F} := \{x \mapsto \sigma(\langle \phi(x), w \rangle) : w \in \mathcal{W}\}$ is a set of generalized linear functions with any fixed link function $\sigma : \mathbb{R} \to \mathbb{R}$ such that $0 < c_l < \sigma' \le c_u$. We then have $\sup_{f \in \mathcal{F}, \gamma > 0, \varepsilon > 0} \theta^{\mathrm{val}}_f(\mathcal{F}, \gamma, \varepsilon) \le \frac{c_u}{c_l}^2 \cdot d$.*

We next provide the formal definition of value function eluder dimension and star number (Foster et al., 2020; Russo and Van Roy, 2013).

**Definition 7** (Value function eluder dimension). *For any $f^\star \in \mathcal{F}$ and $\gamma > 0$, let $\check{\mathfrak{e}}_{f^\star}(\mathcal{F}, \gamma)$ be the length of the longest sequence of data points $x^1, \ldots, x^m$ such that for all $i$, there exists $f^i \in \mathcal{F}$ such that*

$$|f^i(x^i) - f^\star(x^i)| > \gamma, \quad \text{and} \quad \sum_{j < i} (f^i(x^j) - f^\star(x^j))^2 \le \gamma^2.$$

*The value function eluder dimension is defined as $\mathfrak{e}_{f^\star}(\mathcal{F}, \gamma_0) := \sup_{\gamma \ge \gamma_0} \check{\mathfrak{e}}_{f^\star}(\mathcal{F}, \gamma)$.*

**Definition 8** (Value function star number). *For any $f^\star \in \mathcal{F}$ and $\gamma > 0$, let $\check{\mathfrak{s}}_{f^\star}(\mathcal{F}, \gamma)$ be the length of the longest sequence of data points $x^1, \ldots, x^m$ such that for all $i$, there exists $f^i \in \mathcal{F}$ such that*

$$|f^i(x^i) - f^\star(x^i)| > \gamma, \quad \text{and} \quad \sum_{j \ne i} (f^i(x^j) - f^\star(x^j))^2 \le \gamma^2.$$

*The value function eluder dimension is defined as $\mathfrak{s}_{f^\star}(\mathcal{F}, \gamma_0) := \sup_{\gamma \ge \gamma_0} \check{\mathfrak{s}}_{f^\star}(\mathcal{F}, \gamma)$.*

Since the second constrain in the definition of star number is more stringent than the counterpart in the definition of eluder dimension, one immediately have that $\mathfrak{s}_{f^\star}(\mathcal{F}, \gamma) \le \mathfrak{e}_{f^\star}(\mathcal{F}, \gamma)$. We provide known upper bounds for eluder dimension next.

**Proposition 5** (Russo and Van Roy (2013)). *Let $\phi : \mathcal{X} \to \mathbb{R}^d$ be a fixed feature mapping and $\mathcal{W} \subseteq \mathbb{R}^d$ be a fixed set. Suppose $\sup_{x \in \mathcal{X}} \|\phi(x)\|_2 \le 1$ and $\sup_{w \in \mathcal{W}} \|w\|_2 \le 1$. The following upper bounds hold true.*

- *Suppose $\mathcal{F} := \{x \mapsto \langle \phi(x), w \rangle : w \in \mathcal{W}\}$ is a set of linear functions. We then have $\sup_{f^\star \in \mathcal{F}} \mathfrak{e}_{f^\star}(\mathcal{F}, \gamma) = O(d \log \frac{1}{\gamma})$.*

- *Suppose $\mathcal{F} := \{x \mapsto \sigma(\langle \phi(x), w \rangle) : w \in \mathcal{W}\}$ is a set of generalized linear functions with any fixed link function $\sigma : \mathbb{R} \to \mathbb{R}$ such that $0 < c_l < \sigma' \le c_u$. We then have $\sup_{f^\star \in \mathcal{F}} \mathfrak{e}_{f^\star}(\mathcal{F}, \gamma) = O\big(\big(\frac{c_u}{c_l}\big)^2 d \log\big(\frac{c_u}{\gamma}\big)\big)$.*

The next result shows that the disagreement coefficient (with our Definition 1) can be always upper bounded by (squared) star number and eluder dimension.

**Proposition 6** (Foster et al. (2020)). *Suppose $\mathcal{F}$ is a uniform Glivenko-Cantelli class.*

*For any $f^\star : \mathcal{X} \to [0, 1]$ and $\gamma, \varepsilon > 0$, we have $\theta^{\mathrm{val}}_{f^\star}(\mathcal{F}, \gamma, \varepsilon) \le 4(\mathfrak{s}_{f^\star}(\mathcal{F}, \gamma))^2$, and $\theta^{\mathrm{val}}_{f^\star}(\mathcal{F}, \gamma, \varepsilon) \le 4\, \mathfrak{e}_{f^\star}(\mathcal{F}, \gamma)$.*

The requirement that $\mathcal{F}$ is a uniform Glivenko-Cantelli class is rather weak: It is satisfied as long as $\mathcal{F}$ has finite Pseudo dimension (Anthony, 2002).

In our analysis, we sometimes work with sub probability measure (due to selective sampling). Our next result shows that defining the disagreement coefficient over all (sub) probability measures will not affect its value. More specifically, denote $\widetilde{\theta}_{f^\star}^{\mathrm{val}}(\mathcal{F}, \gamma, \varepsilon)$ be the disagreement coefficient defined in Definition 1, but with $\sup$ taking over all probability and sub probability measures. We then have the following equivalence.

**Proposition 7.** *Fix any $\gamma_0, \varepsilon_0 \geq 0$. We have $\widetilde{\theta}_{f^\star}^{\mathrm{val}}(\mathcal{F}, \gamma_0, \varepsilon_0) = \theta_{f^\star}^{\mathrm{val}}(\mathcal{F}, \gamma_0, \varepsilon_0)$.*

*Proof.* We clearly have $\widetilde{\theta}_{f^\star}^{\mathrm{val}}(\mathcal{F}, \gamma_0, \varepsilon_0) \geq \theta_{f^\star}^{\mathrm{val}}(\mathcal{F}, \gamma_0, \varepsilon_0)$ by additionally considering sub probability measures. We next show the opposite direction.

Fix any sub probability measure $\widetilde{\mathcal{D}}_\mathcal{X}$ that is non-zero (otherwise we have $\mathbb{P}_{x \sim \widetilde{\mathcal{D}}_\mathcal{X}}(\cdot) = 0$). Suppose $\mathbb{E}_{x \sim \widetilde{\mathcal{D}}_\mathcal{X}}[1] = \kappa < 1$. We can now consider its normalized probability measure $\overline{\mathcal{D}}_\mathcal{X}$ such that $\overline{\mathcal{D}}_\mathcal{X}(\omega) = \frac{\widetilde{\mathcal{D}}_\mathcal{X}(\omega)}{\kappa}$ (for any $\omega$ in the sigma algebra). Now fix any $\gamma > \gamma_0$ and $\varepsilon > \varepsilon_0$. We have

$$\frac{\gamma^2}{\varepsilon^2} \cdot \mathbb{P}_{\widetilde{\mathcal{D}}_\mathcal{X}}\left(\exists f \in \mathcal{F} : |f(x) - f^\star(x)| > \gamma, \|f - f^\star\|_{\widetilde{\mathcal{D}}_\mathcal{X}}^2 \leq \varepsilon^2\right)$$

$$= \frac{\gamma^2}{\varepsilon^2/\kappa} \cdot \mathbb{P}_{\overline{\mathcal{D}}_\mathcal{X}}\left(\exists f \in \mathcal{F} : |f(x) - f^\star(x)| > \gamma, \|f - f^\star\|_{\overline{\mathcal{D}}_\mathcal{X}}^2 \leq \varepsilon^2/\kappa\right)$$

$$= \frac{\gamma^2}{\bar{\varepsilon}^2} \cdot \mathbb{P}_{\overline{\mathcal{D}}_\mathcal{X}}\left(\exists f \in \mathcal{F} : |f(x) - f^\star(x)| > \gamma, \|f - f^\star\|_{\overline{\mathcal{D}}_\mathcal{X}}^2 \leq \bar{\varepsilon}^2\right)$$

$$\leq \theta_{f^\star}^{\mathrm{val}}(\mathcal{F}, \gamma_0, \varepsilon_0),$$

where we denote $\bar{\varepsilon} := \frac{\varepsilon}{\sqrt{\kappa}} > \varepsilon$, and the last follows from the fact that $\overline{\mathcal{D}}_\mathcal{X}$ is a probability measure. We then have $\widetilde{\theta}_{f^\star}^{\mathrm{val}}(\mathcal{F}, \gamma_0, \varepsilon_0) \leq \theta_{f^\star}^{\mathrm{val}}(\mathcal{F}, \gamma_0, \varepsilon_0)$, and thus the desired result. $\qquad\square$

# D   Concentration results

The Freedman's inequality is quite common in the field of active learning and contextual bandits, e.g., (Freedman, 1975; Agarwal et al., 2014; Krishnamurthy et al., 2019; Foster et al., 2020). We thus state the result without proof.

**Lemma 1** (Freedman's inequality)**.** *Let $(Z_t)_{t \leq T}$ be a real-valued martingale difference sequence adapted to a filtration $\mathfrak{F}_t$, and let $\mathbb{E}_t[\cdot] := \mathbb{E}[\cdot \mid \mathfrak{F}_{t-1}]$. If $|Z_t| \leq B$ almost surely, then for any $\eta \in (0, 1/B)$ it holds with probability at least $1 - \delta$,*

$$\sum_{t=1}^{T} Z_t \leq \eta \sum_{t=1}^{T} \mathbb{E}_t[Z_t^2] + \frac{\log \delta^{-1}}{\eta}.$$

**Lemma 2.** *Let $(Z_t)_{t \leq T}$ to be real-valued sequence of random variables adapted to a filtration $\mathfrak{F}_t$. If $|Z_t| \leq B$ almost surely, then with probability at least $1 - \delta$,*

$$\sum_{t=1}^{T} Z_t \leq \frac{3}{2} \sum_{t=1}^{T} \mathbb{E}_t[Z_t] + 4B \log(2\delta^{-1}),$$

*and*

$$\sum_{t=1}^{T} \mathbb{E}_t[Z_t] \leq 2 \sum_{t=1}^{T} Z_t + 8B \log(2\delta^{-1}).$$

*Proof.* This is a direct consequence of Lemma 1. $\qquad\square$

We define/recall some notations first. Fix any epoch $m \in [M]$ and any time step $t$ within epoch $m$. For any $f \in \mathcal{F}$, we denote $M_t(f) := Q_t((f(x_t) - y_t)^2 - (f^\star(x_t) - y_t)^2)$, and

$\widehat{R}_m(f) := \sum_{t=1}^{\tau_{m-1}} Q_t(f(x_t) - y_t)^2$. Recall that we have $Q_t = g_m(x_t)$. We define filtration $\mathfrak{F}_t := \sigma((x_1, y_1), \ldots, (x_t, y_t))$,[8] and denote $\mathbb{E}_t[\cdot] := \mathbb{E}[\cdot \mid \mathfrak{F}_{t-1}]$.

We first provide a simple concentration result with respect to a finite $\mathcal{F}$.

**Lemma 3.** *Suppose $\mathcal{F}$ is finite. Fix any $\delta \in (0, 1)$. For any $\tau, \tau' \in [T]$ such that $\tau < \tau'$, with probability at least $1 - \delta$, we have*

$$\sum_{t=\tau}^{\tau'} M_t(f) \leq \sum_{t=\tau}^{\tau'} \frac{3}{2} \mathbb{E}_t[M_t(f)] + C_\delta(\mathcal{F}),$$

*and*

$$\sum_{t=\tau}^{\tau'} \mathbb{E}_t[M_t(f)] \leq 2 \sum_{t=\tau}^{\tau'} M_t(f) + C_\delta(\mathcal{F}),$$

*where $C_\delta(\mathcal{F}) = 8 \log\left(\frac{|\mathcal{F}| \cdot T^2}{\delta}\right)$.*

*Proof.* We first notice that $M_t(f)$ adapts to filtration $\mathfrak{F}_t$, and satisfies $|M_t(f)| \leq 1$. The results follow by taking Lemma 2 together with a union bound over $f \in \mathcal{F}$ and $\tau, \tau' \in [T]$. $\square$

Although one can not directly apply a union bound as in Lemma 3 in the case when the set of regression function $\mathcal{F}$ is infinite (but has finite Pseudo dimension by assumption), it turns out that similar guarantees as in Lemma 3 can be derived. We first recall the formal definition of the Pseudo dimension of $\mathcal{F}$.

**Definition 9** (Pseudo Dimension, Pollard (1984); Haussler (1989, 1995))**.** *Consider a set of real-valued function $\mathcal{F} : \mathcal{X} \to \mathbb{R}$. The pseudo-dimension $\mathrm{Pdim}(\mathcal{F})$ of $\mathcal{F}$ is defined as the VC dimension of the set of threshold functions $\{(x, \zeta) \mapsto \mathbb{1}(f(x) > \zeta) : f \in \mathcal{F}\}$.*

**Lemma 4** (Krishnamurthy et al. (2019))**.** *Suppose $\mathrm{Pdim}(\mathcal{F}) < \infty$. Fix any $\delta \in (0, 1)$. For any $\tau, \tau' \in [T]$ such that $\tau < \tau'$, with probability at least $1 - \delta$, we have*

$$\sum_{t=\tau}^{\tau'} M_t(f) \leq \sum_{t=\tau}^{\tau'} \frac{3}{2} \mathbb{E}_t[M_t(f)] + C_\delta(\mathcal{F}),$$

*and*

$$\sum_{t=\tau}^{\tau'} \mathbb{E}_t[M_t(f)] \leq 2 \sum_{t=\tau}^{\tau'} M_t(f) + C_\delta(\mathcal{F}),$$

*where $C_\delta(\mathcal{F}) = C \cdot \left(\mathrm{Pdim}(\mathcal{F}) \cdot \log T + \log\left(\frac{\mathrm{Pdim}(\mathcal{F}) \cdot T}{\delta}\right)\right) \leq C' \cdot \left(\mathrm{Pdim}(\mathcal{F}) \cdot \log\left(\frac{T}{\delta}\right)\right)$, where $C, C' > 0$ are universal constants.*

We will be primarily using Lemma 4 in the following. However, one can replace Lemma 4 with Lemma 3 to derive results with respect to a finite set of regressions $\mathcal{F}$.

## E   Proofs of results in Section 2

We give the proof of Theorem 4 and Theorem 5. Supporting lemmas used in the proofs are deferred to Appendix E.1.

Fix any classifier $\widehat{h} : \mathcal{X} \to \{+1, -1, \bot\}$. For any $x \in \mathcal{X}$, we introduce the notion

$\mathsf{excess}_\gamma(\widehat{h}; x) :=$

$\mathbb{P}_{y|x}\left(y \neq \mathrm{sign}(\widehat{h}(x))\right) \cdot \mathbb{1}\left(\widehat{h}(x) \neq \bot\right) + \left(1/2 - \gamma\right) \cdot \mathbb{1}\left(\widehat{h}(x) = \bot\right) - \mathbb{P}_{y|x}\left(y \neq \mathrm{sign}(h^\star(x))\right)$

$= \mathbb{1}\left(\widehat{h}(x) \neq \bot\right) \cdot \left(\mathbb{P}_{y|x}\left(y \neq \mathrm{sign}(\widehat{h}(x))\right) - \mathbb{P}_{y|x}\left(y \neq \mathrm{sign}(h^\star(x))\right)\right)$

$\quad + \mathbb{1}\left(\widehat{h}(x) = \bot\right) \cdot \left((1/2 - \gamma) - \mathbb{P}_{y|x}\left(y \neq \mathrm{sign}(h^\star(x))\right)\right)$         (6)

---

[8]$y_t$ is not observed (and thus not included in the filtration) when $Q_t = 0$. Note that $Q_t$ is measurable with respect to $\sigma((\mathfrak{F}_{t-1}, x_t))$.

to represent the excess error of $\widehat{h}$ at point $x \in \mathcal{X}$. Excess error of classifier $\widehat{h}$ can be then written as $\text{excess}_\gamma(\widehat{h}) := \text{err}_\gamma(\widehat{h}) - \text{err}(h^\star) = \mathbb{E}_{x \sim \mathcal{D}_\mathcal{X}}[\text{excess}_\gamma(\widehat{h}; x)]$.

**Theorem 4.** *With probability at least $1 - 2\delta$, Algorithm 1 returns a classifier with Chow's excess error at most $\varepsilon$ and label complexity $O(\frac{\theta \, \text{Pdim}(\mathcal{F})}{\gamma^2} \cdot \log^2(\frac{\theta \, \text{Pdim}(\mathcal{F})}{\varepsilon \, \gamma}) \cdot \log(\frac{\theta \, \text{Pdim}(\mathcal{F})}{\varepsilon \, \gamma \, \delta}))$.*

*Proof.* We analyze under the good event $\mathcal{E}$ defined in Lemma 4, which holds with probability at least $1 - \delta$. Note that all supporting lemmas stated in Appendix E.1 hold true under this event.

We analyze the Chow's excess error of $\widehat{h}_m$, which is measurable with respect to $\mathfrak{F}_{\tau_{m-1}}$. For any $x \in \mathcal{X}$, if $g_m(x) = 0$, Lemma 9 implies that $\text{excess}_\gamma(\widehat{h}_m; x) \leq 0$. If $g_m(x) = 1$, we know that $\widehat{h}_m(x) \neq \perp$ and $\frac{1}{2} \in (\text{lcb}(x; \mathcal{F}_m), \text{ucb}(x; \mathcal{F}_m))$. Note that $\widehat{h}_m(x) \neq h^\star(x)$ only if $\text{sign}(2f^\star(x) - 1) \cdot \text{sign}(2\widehat{f}_m(x) - 1) \leq 0$. Since $f^\star, \widehat{f}_m \in \mathcal{F}_m$ by Lemma 5. The error incurred in this case can be upper bounded by $2|f^\star(x) - 1/2| \leq 2w(x; \mathcal{F}_m)$, which results in $\text{excess}_\gamma(\widehat{h}_m; x) \leq 2w(x; \mathcal{F}_m)$. Combining these two cases together, we have

$$\text{excess}_\gamma(\widehat{h}_m) \leq 2\mathbb{E}_{x \sim \mathcal{D}_\mathcal{X}}[\mathbb{1}(g_m(x) = 1) \cdot w(x; \mathcal{F}_m)].$$

Take $m = M$ and apply Lemma 8, with notation $\rho_m := 2\beta_m + C_\delta$, leads to the following guarantee.

$$\text{excess}_\gamma(\widehat{h}_M) \leq \frac{8\rho_M}{\tau_{M-1}\gamma} \cdot \theta_{f^\star}^{\text{val}}\left(\mathcal{F}, \gamma/2, \sqrt{\rho_M/2\tau_{M-1}}\right)$$
$$= O\left(\frac{\text{Pdim}(\mathcal{F}) \cdot \log(T/\delta)}{T\gamma} \cdot \theta_{f^\star}^{\text{val}}\left(\mathcal{F}, \gamma/2, \sqrt{C_\delta/T}\right)\right),$$

where we use the fact that $\frac{T}{2} \leq \tau_{M-1} \leq T$ and definitions of $\beta_m$ and $C_\delta$. Simply considering $\theta := \sup_{f^\star \in \mathcal{F}, \iota > 0} \theta_{f^\star}^{\text{val}}(\mathcal{F}, \gamma/2, \iota)$ as an upper bound of $\theta_{f^\star}^{\text{val}}(\mathcal{F}, \gamma/2, \sqrt{C_\delta/T})$ and taking

$$T = O\left(\frac{\theta \, \text{Pdim}(\mathcal{F})}{\varepsilon \, \gamma} \cdot \log\left(\frac{\theta \, \text{Pdim}(\mathcal{F})}{\varepsilon \, \gamma \, \delta}\right)\right)$$

ensures that $\text{excess}_\gamma(\widehat{h}_M) \leq \varepsilon$.

We now analyze the label complexity (note that the sampling process of Algorithm 1 stops at time $t = \tau_{M-1}$). Note that $\mathbb{E}[\mathbb{1}(Q_t = 1) \mid \mathfrak{F}_{t-1}] = \mathbb{E}_{x \sim \mathcal{D}_\mathcal{X}}[\mathbb{1}(g_m(x) = 1)]$ for any epoch $m \geq 2$ and time step $t$ within epoch $m$. Combine Lemma 2 with Lemma 7 leads to

$$\sum_{t=1}^{\tau_{M-1}} \mathbb{1}(Q_t = 1) \leq \frac{3}{2}\sum_{t=1}^{\tau_{M-1}} \mathbb{E}[\mathbb{1}(Q_t = 1) \mid \mathfrak{F}_{t-1}] + 4\log\delta^{-1}$$

$$\leq 3 + \frac{3}{2}\sum_{m=2}^{M-1} \frac{(\tau_m - \tau_{m-1}) \cdot 4\rho_m}{\tau_{m-1}\gamma^2} \cdot \theta_{f^\star}^{\text{val}}\left(\mathcal{F}, \gamma/2, \sqrt{\rho_m/2\tau_{m-1}}\right) + 4\log\delta^{-1}$$

$$\leq 3 + 6\sum_{m=2}^{M-1} \frac{\rho_m}{\gamma^2} \cdot \theta_{f^\star}^{\text{val}}\left(\mathcal{F}, \gamma/2, \sqrt{\rho_m/2\tau_{m-1}}\right) + 4\log\delta^{-1}$$

$$\leq 3 + 4\log\delta^{-1} + \frac{18\log T \cdot M \cdot C_\delta}{\gamma^2} \cdot \theta_{f^\star}^{\text{val}}\left(\mathcal{F}, \gamma/2, \sqrt{C_\delta/T}\right)$$

$$= O\left(\frac{\theta \, \text{Pdim}(\mathcal{F})}{\gamma^2} \cdot \left(\log\left(\frac{\theta \, \text{Pdim}(\mathcal{F})}{\varepsilon \, \gamma}\right)\right)^2 \cdot \log\left(\frac{\theta \, \text{Pdim}(\mathcal{F})}{\varepsilon \, \gamma \, \delta}\right)\right),$$

with probability at least $1 - 2\delta$ (due to an additional application of Lemma 2); where we plug the above choice of $T$ and upper bound other terms as before. $\square$

**A slightly different guarantee for Algorithm 1.** The stated Algorithm 1 takes $\theta := \sup_{f^\star \in \mathcal{F}, \iota > 0} \theta_{f^\star}^{\text{val}}(\mathcal{F}, \gamma/2, \iota)$ as an input (the value of $\theta$ can be upper bounded for many function class $\mathcal{F}$, as discussed in Appendix C). However, we don't necessarily need to take $\theta$ as an input to the algorithm. Indeed, we can simply run a modified version of Algorithm 1 with $T = \frac{\text{Pdim}(\mathcal{F})}{\varepsilon \, \gamma}$.

Following similar analyses in proof of Theorem 4, set $\iota := \sqrt{C_\delta/T} \propto \sqrt{\gamma \varepsilon}$, the modified version achieves excess error

$$\mathsf{excess}_\gamma(\widehat{h}_M) = O\left(\varepsilon \cdot \theta_{f^\star}^{\mathrm{val}}(\mathcal{F}, \gamma/2, \iota) \cdot \log\left(\frac{\mathrm{Pdim}(\mathcal{F})}{\varepsilon\,\delta\,\gamma}\right)\right)$$

with label complexity

$$O\left(\frac{\theta_{f^\star}^{\mathrm{val}}(\mathcal{F}, \gamma/2, \iota) \cdot \mathrm{Pdim}(\mathcal{F})}{\gamma^2} \cdot \left(\log\left(\frac{\mathrm{Pdim}(\mathcal{F})}{\varepsilon\,\gamma}\right)\right)^2 \cdot \log\left(\frac{\mathrm{Pdim}(\mathcal{F})}{\varepsilon\,\gamma\,\delta}\right)\right).$$

We now discuss the efficient implementation of Algorithm 1 and its computational complexity. We first state some known results in computing the confidence intervals with respect to a set of regression functions $\mathcal{F}$.

**Proposition 8** (Krishnamurthy et al. (2017); Foster et al. (2018, 2020))**.** *Consider the setting studied in Algorithm 1. Fix any epoch $m \in [M]$ and denote $\mathcal{B}_m := \{(x_t, Q_t, y_t)\}_{t=1}^{\mathcal{T}_{m-1}}$. Fix any $\alpha > 0$. For any data point $x \in \mathcal{X}$, there exists algorithms $\mathbf{Alg}_{\mathsf{lcb}}$ and $\mathbf{Alg}_{\mathsf{ucb}}$ that certify*

$$\mathsf{lcb}(x; \mathcal{F}_m) - \alpha \le \mathbf{Alg}_{\mathsf{lcb}}(x; \mathcal{B}_m, \beta_m, \alpha) \le \mathsf{lcb}(x; \mathcal{F}_m) \quad and$$
$$\mathsf{ucb}(x; \mathcal{F}_m) \le \mathbf{Alg}_{\mathsf{ucb}}(x; \mathcal{B}_m, \beta_m, \alpha) \le \mathsf{ucb}(x; \mathcal{F}_m) + \alpha.$$

*The algorithms take $O(\frac{1}{\alpha^2} \log \frac{1}{\alpha})$ calls of the regression oracle for general $\mathcal{F}$ and take $O(\log \frac{1}{\alpha})$ calls of the regression oracle if $\mathcal{F}$ is convex and closed under pointwise convergence.*

*Proof.* See Algorithm 2 in Krishnamurthy et al. (2017) for the general case; and Algorithm 3 in Foster et al. (2018) for the case when $\mathcal{F}$ is convex and closed under pointwise convergence. □

We next discuss the computational efficiency of Algorithm 1. Recall that we redefine $\theta := \sup_{f^\star \in \mathcal{F}, \iota > 0} \theta_{f^\star}^{\mathrm{val}}(\mathcal{F}, \gamma/4, \iota)$ in the Theorem 5 to account to approximation error.

**Theorem 5.** *Algorithm 1 can be efficiently implemented via the regression oracle and enjoys the same theoretical guarantees stated in Theorem 4. The number of oracle calls needed is $\widetilde{O}(\frac{\theta\,\mathrm{Pdim}(\mathcal{F})}{\varepsilon\,\gamma^3})$ for a general set of regression functions $\mathcal{F}$, and $\widetilde{O}(\frac{\theta\,\mathrm{Pdim}(\mathcal{F})}{\varepsilon\,\gamma})$ when $\mathcal{F}$ is convex and closed under pointwise convergence. The per-example inference time of the learned $\widehat{h}_M$ is $\widetilde{O}(\frac{1}{\gamma^2} \log^2(\frac{\theta\,\mathrm{Pdim}(\mathcal{F})}{\varepsilon}))$ for general $\mathcal{F}$, and $\widetilde{O}(\log \frac{1}{\gamma})$ when $\mathcal{F}$ is convex and closed under pointwise convergence.*

*Proof.* Fix any epoch $m \in [M]$. Denote $\bar{\alpha} := \frac{\gamma}{4M}$ and $\alpha_m := \frac{(M-m)\gamma}{4M}$. With any observed $x \in \mathcal{X}$, we construct the approximated confidence intervals $\widehat{\mathsf{lcb}}(x; \mathcal{F}_m)$ and $\widehat{\mathsf{ucb}}(x; \mathcal{F}_m)$ as follows.

$$\widehat{\mathsf{lcb}}(x; \mathcal{F}_m) := \mathbf{Alg}_{\mathsf{lcb}}(x; \mathcal{B}_m, \beta_m, \bar{\alpha}) - \alpha_m \quad \text{and} \quad \widehat{\mathsf{ucb}}(x; \mathcal{F}_m) := \mathbf{Alg}_{\mathsf{ucb}}(x; \mathcal{B}_m, \beta_m, \bar{\alpha}) + \alpha_m.$$

For efficient implementation of Algorithm 1, we replace $\mathsf{lcb}(x; \mathcal{F}_m)$ and $\mathsf{ucb}(x; \mathcal{F}_m)$ with $\widehat{\mathsf{lcb}}(x; \mathcal{F}_m)$ and $\widehat{\mathsf{ucb}}(x; \mathcal{F}_m)$ in the construction of $\widehat{h}_m$ and $g_m$.

Based on Proposition 8, we know that

$$\mathsf{lcb}(x; \mathcal{F}_m) - \alpha_m - \bar{\alpha} \le \widehat{\mathsf{lcb}}(x; \mathcal{F}_m) \le \mathsf{lcb}(x; \mathcal{F}_m) - \alpha_m \quad \text{and}$$
$$\mathsf{ucb}(x; \mathcal{F}_m) + \alpha_m \le \widehat{\mathsf{ucb}}(x; \mathcal{F}_m) \le \mathsf{ucb}(x; \mathcal{F}_m) + \alpha_m + \bar{\alpha}.$$

Since $\alpha_m + \bar{\alpha} \le \frac{\gamma}{4}$ for any $m \in [M]$, the guarantee in Lemma 6 can be modified as $g_m(x) = 1 \implies w(x; \mathcal{F}_m) \ge \frac{\gamma}{2}$.

Fix any $m \ge 2$. Since $\mathcal{F}_m \subseteq \mathcal{F}_{m-1}$ by Lemma 5, we have

$$\widehat{\mathsf{lcb}}(x; \mathcal{F}_m) \ge \mathsf{lcb}(x; \mathcal{F}_m) - \alpha_m - \bar{\alpha} \ge \mathsf{lcb}(x; \mathcal{F}_{m-1}) - \alpha_{m-1} \ge \widehat{\mathsf{lcb}}(x; \mathcal{F}_{m-1}) \quad \text{and}$$
$$\widehat{\mathsf{ucb}}(x; \mathcal{F}_m) \le \mathsf{ucb}(x; \mathcal{F}_m) + \alpha_m + \bar{\alpha} \le \mathsf{ucb}(x; \mathcal{F}_{m-1}) + \alpha_{m-1} \le \widehat{\mathsf{ucb}}(x; \mathcal{F}_{m-1}).$$

These ensure $\mathbb{1}(g_m(x) = 1) \le \mathbb{1}(g_{m-1}(x) = 1)$. Thus, the guarantees stated in Lemma 7 and Lemma 8 still hold (with $\frac{\gamma}{2}$ replaced by $\frac{\gamma}{4}$ due to modification of Lemma 6). The guarantee stated in

[Lemma 9](#) also hold since $\widehat{\mathsf{lcb}}(x; \mathcal{F}_m) \le \mathsf{lcb}(x; \mathcal{F}_m)$ and $\widehat{\mathsf{ucb}}(x; \mathcal{F}_m) \ge \mathsf{ucb}(x; \mathcal{F}_m)$ by construction. As a result, the guarantees stated in [Theorem 4](#) hold true with changes only in constant terms.

We now discuss the computational complexity of the efficient implementation. At the beginning of each epoch $m$. We use one oracle call to compute $\widehat{f}_m = \arg\min_{f \in \mathcal{F}} \sum_{t=1}^{\tau_{m-1}} Q_t(f(x_t) - y_t)^2$. The main computational cost comes from computing $\widehat{\mathsf{lcb}}$ and $\widehat{\mathsf{ucb}}$ at each time step. We take $\alpha = \bar{\alpha} := \frac{\gamma}{4M}$ into [Proposition 8](#), which leads to $O(\frac{(\log T)^2}{\gamma^2} \cdot \log(\frac{\log T}{\gamma}))$ calls of the regression oracle for general $\mathcal{F}$ and $O(\log(\frac{\log T}{\gamma}))$ calls of the regression oracle for any convex $\mathcal{F}$ that is closed under pointwise convergence. This also serves as the per-example inference time for $\widehat{h}_M$. The total computational cost of [Algorithm 1](#) is then derived by multiplying the per-round cost by $T$ and plugging $T = \widetilde{O}(\frac{\theta \operatorname{Pdim}(\mathcal{F})}{\varepsilon \gamma})$ into the bound (for any parameter, we only keep $\mathrm{poly}$ factors in the total computational cost and keep $\mathrm{poly}$ or $\mathrm{polylog}$ dependence in the per-example computational cost). $\qquad\square$

### E.1 Supporting lemmas

We use $\mathcal{E}$ to denote the good event considered in [Lemma 4](#), and analyze under this event in this section. We abbreviate $C_\delta := C_\delta(\mathcal{F})$ in the following analysis.

**Lemma 5.** *The followings hold true:*

1. $f^\star \in \mathcal{F}_m$ for any $m \in [M]$.

2. $\sum_{t=1}^{\tau_{m-1}} \mathbb{E}_t[M_t(f)] \le 2\beta_m + C_\delta$ for any $f \in \mathcal{F}_m$.

3. $\mathcal{F}_{m+1} \subseteq \mathcal{F}_m$ for any $m \in [M-1]$.

*Proof.* 1. Fix any epoch $m \in [M]$ and time step $t$ within epoch $m$. Since $\mathbb{E}[y_t] = f^\star(x_t)$, we have $\mathbb{E}_t[M_t(f)] = \mathbb{E}[Q_t(f(x) - f^\star(x))^2] = \mathbb{E}[g_m(x)(f(x) - f^\star(x))^2] \ge 0$ for any $f \in \mathcal{F}$. By [Lemma 4](#), we then have $\widehat{R}_m(f^\star) \le \widehat{R}_m(f) + C_\delta/2 \le \widehat{R}_m(f) + \beta_m$ for any $f \in \mathcal{F}$. The elimination rule in [Algorithm 2](#) then implies that $f^\star \in \mathcal{F}_m$ for any $m \in [M]$.

2. Fix any $f \in \mathcal{F}_m$. With [Lemma 4](#), we have

$$
\begin{aligned}
\sum_{t=1}^{\tau_{m-1}} \mathbb{E}_t[M_t(f)] &\le 2 \sum_{t=1}^{\tau_{m-1}} M_t(f) + C_\delta \\
&= 2\widehat{R}_m(f) - 2\widehat{R}_m(f^\star) + C_\delta \\
&\le 2\widehat{R}_m(f) - 2\widehat{R}_m(\widehat{f}_m) + C_\delta \\
&\le 2\beta_m + C_\delta,
\end{aligned}
$$

where the third line comes from the fact that $\widehat{f}_m$ is the minimizer of $\widehat{R}_m(\cdot)$; and the last line comes from the fact that $f \in \mathcal{F}_m$.

3. Fix any $f \in \mathcal{F}_{m+1}$. We have

$$
\begin{aligned}
\widehat{R}_m(f) - \widehat{R}_m(\widehat{f}_m) &\le \widehat{R}_m(f) - \widehat{R}_m(f^\star) + \frac{C_\delta}{2} \\
&= \widehat{R}_{m+1}(f) - \widehat{R}_{m+1}(f^\star) - \sum_{t=\tau_{m-1}+1}^{\tau_m} M_t(f) + \frac{C_\delta}{2} \\
&\le \widehat{R}_{m+1}(f) - \widehat{R}_{m+1}(\widehat{f}_{m+1}) - \sum_{t=\tau_{m-1}+1}^{\tau_m} \mathbb{E}_t[M_t(f)]/2 + C_\delta \\
&\le \beta_{m+1} + C_\delta \\
&= \beta_m,
\end{aligned}
$$

where the first line comes from [Lemma 4](#); the third line comes from the fact that $\widehat{f}_{m+1}$ is the minimizer with respect to $\widehat{R}_{m+1}$ and [Lemma 4](#); the last line comes from the definition of $\beta_m$.

$\qquad\square$

**Lemma 6.** *For any $m \in [M]$, we have $g_m(x) = 1 \implies w(x; \mathcal{F}_m) > \gamma$.*

*Proof.* We only need to show that $\mathsf{ucb}(x; \mathcal{F}_m) - \mathsf{lcb}(x; \mathcal{F}_m) \leq \gamma \implies g_m(x) = 0$. Suppose otherwise $g_m(x) = 1$, which implies that both

$$\frac{1}{2} \in (\mathsf{lcb}(x; \mathcal{F}_m), \mathsf{ucb}(x; \mathcal{F}_m)) \quad \text{and} \quad [\mathsf{lcb}(x; \mathcal{F}_m), \mathsf{ucb}(x; \mathcal{F}_m)] \not\subseteq \left[\frac{1}{2} - \gamma, \frac{1}{2} + \gamma\right]. \quad (7)$$

If $\frac{1}{2} \in (\mathsf{lcb}(x; \mathcal{F}_m), \mathsf{ucb}(x; \mathcal{F}_m))$ and $\mathsf{ucb}(x; \mathcal{F}_m) - \mathsf{lcb}(x; \mathcal{F}_m) \leq \gamma$, we must have $\mathsf{lcb}(x; \mathcal{F}_m) \geq \frac{1}{2} - \gamma$ and $\mathsf{ucb}(x; \mathcal{F}_m) \leq \frac{1}{2} + \gamma$, which contradicts with Eq. (7). $\square$

We introduce more notations. Fix any $m \in [M]$. We use $n_m := \tau_m - \tau_{m-1}$ to denote the length of epoch $m$, and use abbreviation $\rho_m := 2\beta_m + C_\delta$. Denote $(\mathcal{X}, \Sigma, \mathcal{D}_\mathcal{X})$ as the (marginal) probability space, and denote $\overline{\mathcal{X}}_m := \{x \in \mathcal{X} : g_m(x) = 1\} \in \Sigma$ be the region where query *is* requested within epoch $m$. Since we have $\mathcal{F}_{m+1} \subseteq \mathcal{F}_m$ by Lemma 5, we clearly have $\overline{\mathcal{X}}_{m+1} \subseteq \overline{\mathcal{X}}_m$. We now define a sub probability measure $\bar{\mu}_m := (\mathcal{D}_\mathcal{X})_{|\overline{\mathcal{X}}_m}$ such that $\bar{\mu}_m(\omega) = \mathcal{D}_\mathcal{X}(\omega \cap \overline{\mathcal{X}}_m)$ for any $\omega \in \Sigma$. Fix any time step $t$ within epoch $m$ and any $\overline{m} \leq m$. Consider any measurable function $F$ (that is $\mathcal{D}_\mathcal{X}$ integrable), we have

$$
\begin{aligned}
\mathbb{E}_{x \sim \mathcal{D}_\mathcal{X}}[\mathbb{1}(g_m(x) = 1) \cdot F(x)] &= \int_{x \in \overline{\mathcal{X}}_m} F(x) \, d\mathcal{D}_\mathcal{X}(x) \\
&\leq \int_{x \in \overline{\mathcal{X}}_{\overline{m}}} F(x) \, d\mathcal{D}_\mathcal{X}(x) \\
&= \int_{x \in \mathcal{X}} F(x) \, d\bar{\mu}_{\overline{m}}(x) \\
&=: \mathbb{E}_{x \sim \bar{\mu}_{\overline{m}}}[F(x)],
\end{aligned}
\quad (8)
$$

where, by a slightly abuse of notations, we use $\mathbb{E}_{x \sim \mu}[\cdot]$ to denote the integration with any sub probability measure $\mu$. In particular, Eq. (8) holds with equality when $\overline{m} = m$.

**Lemma 7.** *Fix any epoch $m \geq 2$. We have*

$$\mathbb{E}_{x \sim \mathcal{D}_\mathcal{X}}[\mathbb{1}(g_m(x) = 1)] \leq \frac{4\rho_m}{\tau_{m-1}\gamma^2} \cdot \theta_{f^\star}^{\mathrm{val}}\left(\mathcal{F}, \gamma/2, \sqrt{\rho_m/2\tau_{m-1}}\right).$$

*Proof.* We know that $\mathbb{1}(g_m(x) = 1) = \mathbb{1}(g_m(x) = 1) \cdot \mathbb{1}(w(x; \mathcal{F}_m) > \gamma)$ from Lemma 6. Thus, for any $\overline{m} \leq m$, we have

$$
\begin{aligned}
\mathbb{E}_{x \sim \mathcal{D}_\mathcal{X}}[\mathbb{1}(g_m(x) = 1)] &= \mathbb{E}_{x \sim \mathcal{D}_\mathcal{X}}[\mathbb{1}(g_m(x) = 1) \cdot \mathbb{1}(w(x; \mathcal{F}_m) > \gamma)] \\
&\leq \mathbb{E}_{x \sim \bar{\mu}_{\overline{m}}}[\mathbb{1}(w(x; \mathcal{F}_m) > \gamma)] \\
&\leq \mathbb{E}_{x \sim \bar{\mu}_{\overline{m}}}\left(\mathbb{1}\big(\exists f \in \mathcal{F}_m, |f(x) - f^\star(x)| > \gamma/2\big)\right),
\end{aligned}
\quad (9)
$$

where the second line uses Eq. (8) and the last line comes from the facts that $f^\star \in \mathcal{F}_m$ and $w(x; \mathcal{F}_m) > \gamma \implies \exists f \in \mathcal{F}_m, |f(x) - f^\star(x)| > \gamma/2$.

For any time step $t$, let $m(t)$ denote the epoch where $t$ belongs to. From Lemma 5, we know that, $\forall f \in \mathcal{F}_m$,

$$
\begin{aligned}
\rho_m &\geq \sum_{t=1}^{\tau_{m-1}} \mathbb{E}_t\left[Q_t\big(f(x_t) - f^\star(x_t)\big)^2\right] \\
&= \sum_{t=1}^{\tau_{m-1}} \mathbb{E}_{x \sim \mathcal{D}_\mathcal{X}}\left[\mathbb{1}(g_{m(t)}(x) = 1) \cdot \big(f(x) - f^\star(x)\big)^2\right] \\
&= \sum_{\overline{m}=1}^{m-1} n_{\overline{m}} \cdot \mathbb{E}_{x \sim \bar{\mu}_{\overline{m}}}\left[(f(x) - f^\star(x))^2\right] \\
&= \tau_{m-1} \mathbb{E}_{x \sim \bar{\nu}_m}\left[(f(x) - f^\star(x))^2\right],
\end{aligned}
\quad (10)
$$

where we use $Q_t = g_{m(t)}(x_t) = \mathbb{1}(g_{m(t)}(x) = 1)$ and Eq. (8) on the second line, and define a new sub probability measure

$$\bar{\nu}_m := \frac{1}{\tau_{m-1}} \sum_{\overline{m}=1}^{m-1} n_{\overline{m}} \cdot \bar{\mu}_{\overline{m}}$$

on the third line.

Plugging Eq. (10) into Eq. (9) leads to the bound

$$\mathbb{E}_{x \sim \mathcal{D}_{\mathcal{X}}}[\mathbb{1}(g_m(x) = 1)]$$
$$\leq \mathbb{E}_{x \sim \bar{\nu}_m}\left[\mathbb{1}\left(\exists f \in \mathcal{F}, |f(x) - f^\star(x)| > \gamma/2, \mathbb{E}_{x \sim \bar{\nu}_m}\left[(f(x) - f^\star(x))^2\right] \leq \frac{\rho_m}{\tau_{m-1}}\right)\right],$$

where we use the definition of $\bar{\nu}_m$ again (note that Eq. (9) works with any $\overline{m} \leq m$). Combining the above result with the discussion around Proposition 7 and Definition 1, we then have

$$\mathbb{E}_{x \sim \mathcal{D}_{\mathcal{X}}}[\mathbb{1}(g_m(x) = 1)] \leq \frac{4\rho_m}{\tau_{m-1}\gamma^2} \cdot \theta_{f^\star}^{\text{val}}\left(\mathcal{F}, \gamma/2, \sqrt{\rho_m/2\tau_{m-1}}\right).$$

$\square$

**Lemma 8.** *Fix any epoch $m \geq 2$. We have*

$$\mathbb{E}_{x \sim \mathcal{D}_{\mathcal{X}}}[\mathbb{1}(g_m(x) = 1) \cdot w(x; \mathcal{F}_m)] \leq \frac{4\rho_m}{\tau_{m-1}\gamma} \cdot \theta_{f^\star}^{\text{val}}\left(\mathcal{F}, \gamma/2, \sqrt{\rho_m/2\tau_{m-1}}\right).$$

*Proof.* Similar to the proof of Lemma 7, we have

$$\mathbb{E}_{x \sim \mathcal{D}_{\mathcal{X}}}[\mathbb{1}(g_m(x) = 1) \cdot w(x; \mathcal{F}_m)] = \mathbb{E}_{x \sim \mathcal{D}_{\mathcal{X}}}[\mathbb{1}(g_m(x) = 1) \cdot \mathbb{1}(w(x; \mathcal{F}_m) > \gamma) \cdot w(x; \mathcal{F}_m)]$$
$$\leq \mathbb{E}_{x \sim \bar{\mu}_{\overline{m}}}[\mathbb{1}(w(x; \mathcal{F}_m) > \gamma) \cdot w(x; \mathcal{F}_m)]$$

for any $\overline{m} \leq m$. With $\bar{\nu}_m = \frac{1}{\tau_{m-1}} \sum_{\overline{m}=1}^{m-1} n_{\overline{m}} \cdot \bar{\mu}_{\overline{m}}$, we then have

$$\mathbb{E}_{x \sim \mathcal{D}_{\mathcal{X}}}[\mathbb{1}(g_m(x) = 1) \cdot w(x; \mathcal{F}_m)]$$
$$\leq \mathbb{E}_{x \sim \bar{\nu}_m}[\mathbb{1}(w(x; \mathcal{F}_m) > \gamma) \cdot w(x; \mathcal{F}_m)]$$
$$\leq \mathbb{E}_{x \sim \bar{\nu}_m}\left[\mathbb{1}(\exists f \in \mathcal{F}_m, |f(x) - f^\star(x)| > \gamma/2) \cdot \left(\sup_{f,f' \in \mathcal{F}_m} |f(x) - f'(x)|\right)\right]$$
$$\leq 2\mathbb{E}_{x \sim \bar{\nu}_m}\left[\mathbb{1}(\exists f \in \mathcal{F}_m, |f(x) - f^\star(x)| > \gamma/2) \cdot \left(\sup_{f \in \mathcal{F}_m} |f(x) - f^\star(x)|\right)\right]$$
$$\leq 2 \int_{\gamma/2}^{1} \mathbb{E}_{x \sim \bar{\nu}_m}\left[\mathbb{1}\left(\sup_{f \in \mathcal{F}_m} |f(x) - f^\star(x)| \geq \omega\right)\right] d\omega$$
$$\leq 2 \int_{\gamma/2}^{1} \frac{1}{\omega^2} d\omega \cdot \left(\frac{\rho_m}{\tau_{m-1}} \cdot \theta_{f^\star}^{\text{val}}\left(\mathcal{F}, \gamma/2, \sqrt{\rho_m/2\tau_{m-1}}\right)\right)$$
$$\leq \frac{4\rho_m}{\tau_{m-1}\gamma} \cdot \theta_{f^\star}^{\text{val}}\left(\mathcal{F}, \gamma/2, \sqrt{\rho_m/2\tau_{m-1}}\right),$$

where we use similar steps as in the proof of Lemma 7. $\square$

**Lemma 9.** *Fix any $m \in [M]$. We have $\text{excess}_\gamma(\widehat{h}_m; x) \leq 0$ if $g_m(x) = 0$.*

*Proof.* Recall that

$$\text{excess}_\gamma(\widehat{h}; x) = \mathbb{1}(\widehat{h}(x) \neq \bot) \cdot (\mathbb{P}_y(y \neq \text{sign}(\widehat{h}(x))) - \mathbb{P}_y(y \neq \text{sign}(h^\star(x))))$$
$$+ \mathbb{1}(\widehat{h}(x) = \bot) \cdot ((1/2 - \gamma) - \mathbb{P}_y(y \neq \text{sign}(h^\star(x)))).$$

We now analyze the event $\{g_m(x) = 0\}$ in two cases.

**Case 1: $\widehat{h}_m(x) = \bot$.**

Since $\eta(x) = f^\star(x) \in [\mathsf{lcb}(x; \mathcal{F}_m), \mathsf{ucb}(x; \mathcal{F}_m)]$, we know that $\eta(x) \in [\frac{1}{2} - \gamma, \frac{1}{2} + \gamma]$ and thus $\mathbb{P}_y\big(y \neq \mathrm{sign}(h^\star(x))\big) \geq \frac{1}{2} - \gamma$. As a result, we have $\mathsf{excess}_\gamma(\widehat{h}_m; x) \leq 0$.

**Case 2:** $\widehat{h}_m(x) \neq \perp$ **but** $\frac{1}{2} \notin (\mathsf{lcb}(x; \mathcal{F}_m), \mathsf{ucb}(x; \mathcal{F}_m))$.

In this case, we know that $\mathrm{sign}(\widehat{h}_m(x)) = \mathrm{sign}(h^\star(x))$ whenever $\eta(x) \in [\mathsf{lcb}(x; \mathcal{F}_m), \mathsf{ucb}(x; \mathcal{F}_m)]$. As a result, we have $\mathsf{excess}_\gamma(\widehat{h}_m; x) \leq 0$ as well. $\qquad\square$

## F  Proofs of results in Section 3

**Proposition 2.** *The classifier $\widehat{h}$ returned by Algorithm 1 enjoys proper abstention. With randomization over the abstention region, we have the following upper bound on its standard excess error*

$$\mathrm{err}(\check{h}) - \mathrm{err}(h^\star) \leq \mathrm{err}_\gamma(\widehat{h}) - \mathrm{err}(h^\star) + \gamma \cdot \mathbb{P}_{x \sim \mathcal{D}_\mathcal{X}}(x \in \mathcal{X}_\gamma). \tag{5}$$

*Proof.* The proper abstention property of $\widehat{h}$ returned by Algorithm 1 is achieved via conservation: $\widehat{h}$ will avoid abstention unless it is absolutely sure that abstention is the optimal choice. The proper abstention property implies that $\mathbb{P}_{x \sim \mathcal{D}_\mathcal{X}}(\widehat{h}(x) = \perp) \leq \mathbb{P}_{x \sim \mathcal{D}_\mathcal{X}}(x \in \mathcal{X}_\gamma)$. The desired result follows by combining this inequality with Eq. (4). $\qquad\square$

**Theorem 6.** *With an appropriate choice of the abstention parameter $\gamma$ in Algorithm 1 and randomization over the abstention region, Algorithm 1 learns a classifier $\check{h}$ at the minimax optimal rates: To achieve $\varepsilon$ standard excess error, it takes $\widetilde{\Theta}(\tau_0^{-2})$ labels under Massart noise and takes $\widetilde{\Theta}(\varepsilon^{-2/(1+\beta)})$ labels under Tsybakov noise.*

*Proof.* The results follow by taking the corresponding $\gamma$ in Algorithm 1 and then apply Proposition 2. In the case with Massart noise, we have $\mathbb{P}_{x \sim \mathcal{D}_\mathcal{X}}(x \in \mathcal{X}_\gamma) = 0$ when $\gamma = \tau_0$; and the corresponding label complexity scales as $\widetilde{O}(\tau_0^{-2})$. In the case with Tsybakov noise, we have $\mathbb{P}_{x \sim \mathcal{D}_\mathcal{X}}(x \in \mathcal{X}_\gamma) = \frac{\varepsilon}{2}$ when $\gamma = (\frac{\varepsilon}{2c})^{1/(1+\beta)}$. Applying Algorithm 1 to achieve $\frac{\varepsilon}{2}$ Chow's excess error thus leads to $\frac{\varepsilon}{2} + \frac{\varepsilon}{2} = \varepsilon$ standard excess error. The corresponding label complexity scales as $\widetilde{O}(\varepsilon^{-2/(1+\beta)})$. $\qquad\square$

**Theorem 7.** *With an appropriate choice of the abstention parameter $\gamma$ in Algorithm 1 and randomization over the abstention region, Algorithm 1 learns a classifier $\check{h}$ with $\varepsilon + \zeta_0$ standard excess error after querying $\widetilde{\Theta}(\tau_0^{-2})$ labels under Definition 5 or querying $\widetilde{\Theta}(\varepsilon^{-2/(1+\beta)})$ labels under Definition 6.*

*Proof.* For any abstention parameter $\gamma > 0$, we denote $\mathcal{X}_{\zeta_0, \gamma} := \{x \in \mathcal{X} : \eta(x) \in [\frac{1}{2} - \gamma, \frac{1}{2} + \gamma], |\eta(x) - 1/2| > \zeta_0\}$ as the intersection of the region controlled by noise-seeking conditions and the (possible) abstention region. Let $\widehat{h}$ be the classifier returned by Algorithm 1 and $\check{h}$ be its randomized version (over the abstention region). We denote $\mathcal{S} := \{x \in \mathcal{X} : \widehat{h}(x) = \perp\}$ be the abstention region of $\widehat{h}$. Since $\widehat{h}$ abstains properly, we have $\mathcal{S} \subseteq \{x \in \mathcal{X} : |\eta(x) - 1/2| \leq \gamma\} =: \mathcal{X}_\gamma$. We write $\mathcal{S}_0 := \mathcal{S} \cap \mathcal{X}_{\zeta_0, \gamma}$, $\mathcal{S}_1 := \mathcal{S} \setminus \mathcal{S}_0$ and $\mathcal{S}_2 := \mathcal{X} \setminus \mathcal{S}$. For any $h : \mathcal{X} \to \mathcal{Y}$, we define the notation $\mathsf{excess}(h; x) := (\mathbb{P}_{y|x}\big(y \neq \mathrm{sign}(h(x))\big) - \mathbb{P}_{y|x}\big(y \neq \mathrm{sign}(h^\star(x))\big))$, and have $\mathsf{excess}(h) = \mathbb{E}_{x \sim \mathcal{D}_\mathcal{X}}[\mathsf{excess}(h; x)]$. We then have

$\mathsf{excess}(\check{h})$

$= \mathbb{E}_{x \sim \mathcal{D}_\mathcal{X}}\big[\mathsf{excess}(\check{h}; x) \cdot \mathbb{1}(x \in \mathcal{S}_0)\big] + \mathbb{E}_{x \sim \mathcal{D}_\mathcal{X}}\big[\mathsf{excess}(\check{h}; x) \cdot \mathbb{1}(x \in \mathcal{S}_1)\big] + \mathbb{E}_{x \sim \mathcal{D}_\mathcal{X}}\big[\mathsf{excess}(\check{h}; x) \cdot \mathbb{1}(x \in \mathcal{S}_2)\big]$

$\leq \gamma \cdot \mathbb{E}_{x \sim \mathcal{D}_\mathcal{X}}[\mathbb{1}(x \in \mathcal{S}_0)] + \zeta_0 \cdot \mathbb{E}_{x \sim \mathcal{D}_\mathcal{X}}[\mathbb{1}(x \in \mathcal{S}_1)] + \mathbb{E}_{x \sim \mathcal{D}_\mathcal{X}}[\mathsf{excess}_\gamma(\widehat{h}; x) \cdot \mathbb{1}(x \in \mathcal{S}_2)]$

$\leq \gamma \cdot \mathbb{E}_{x \sim \mathcal{D}_\mathcal{X}}[\mathbb{1}(x \in \mathcal{X}_{\zeta_0, \gamma})] + \zeta_0 + \varepsilon/2,$

where the bound on the second term comes from the fact that $\mathcal{S} \subseteq \mathcal{X}_\gamma$ and the bound on the third term comes from the same analysis that appears in the proof of Theorem 4 (with $\varepsilon/2$ accuracy). One can then tune $\gamma$ in ways discussed in the proof of Theorem 6 to bound the first term by $\varepsilon/2$, i.e., $\gamma \cdot \mathbb{E}_{x \sim \mathcal{D}_\mathcal{X}}[\mathbb{1}(x \in \mathcal{X}_{\zeta_0, \gamma})] \leq \varepsilon/2$, with similar label complexity. $\qquad\square$

**Proposition 3.** *Fix $\varepsilon, \delta, \gamma > 0$. For any labeling budget $B \gtrsim \frac{1}{\gamma^2} \cdot \log^2(\frac{1}{\varepsilon\gamma}) \cdot \log(\frac{1}{\varepsilon\gamma\delta})$, there exists a learning problem (with a set of linear regression functions) satisfying Definition 5/Definition 6 such that (1) any "uncertainty-based" active learner suffers expected standard excess error $\Omega(B^{-1})$; yet (2) with probability at least $1 - \delta$, Algorithm 1 returns a classifier with standard excess error at most $\varepsilon$.*

Before proving Proposition 3, we first construct a simple problem with linear regression function and give the formal definition of "uncertainty-based" active learner.

**Example 1.** *We consider the case where $\mathcal{X} = [0,1]$ and $\mathcal{D}_{\mathcal{X}} = \mathrm{unif}(\mathcal{X})$. We consider feature embedding $\phi : \mathcal{X} \to \mathbb{R}^2$, i.e., $\phi(x) = [\phi_1(x), \phi_2(x)]^\top$. We take $\phi_1(x) := 1$ for any $x \in \mathcal{X}$, and define $\phi_2(x)$ as*

$$\phi_2(x) := \begin{cases} 0, & x \in \mathcal{X}_{\mathrm{hard}}, \\ 1, & x \in \mathcal{X}_{\mathrm{easy}}, \end{cases}$$

*where $\mathcal{X}_{\mathrm{easy}} \subseteq \mathcal{X}$ is any subset such that $\mathcal{D}_{\mathcal{X}}(\mathcal{X}_{\mathrm{easy}}) = p$, for some constant $p \in (0,1)$, and $\mathcal{X}_{\mathrm{hard}} = \mathcal{X} \setminus \mathcal{X}_{\mathrm{easy}}$. We consider a set of linear regression function $\mathcal{F} := \{f_\theta : f_\theta(x) = \langle \phi(x), \theta \rangle, \|\theta\|_2 \le 1\}$. We set $f^\star = f_{\theta^\star}$, where $\theta^\star = [\theta_1^\star, \theta_2^\star]^\top$ is selected such that $\theta_1^\star = \frac{1}{2}$ and $\theta_2^\star = \mathrm{unif}(\{\pm\frac{1}{2}\})$.*

**Definition 10.** *We say an algorithm is a "uncertainty-based" active learner if, for any $x \in \mathcal{X}$, the learner*

- *constructs an open confidence interval $(\mathsf{lcb}(x), \mathsf{ucb}(x))$ such that $\eta(x) \in (\mathsf{lcb}(x), \mathsf{ucb}(x))$;[9]*

- *queries the label of $x \in \mathcal{X}$ if $\frac{1}{2} \in (\mathsf{lcb}(x), \mathsf{ucb}(x))$.*

*Proof.* With any given labeling budget $B$, we consider the problem instance described in Example 1 with $p = B^{-1}/2$. We can easily see that this problem instance satisfy Definition 5 and Definition 6.

We first consider any "uncertainty-based" active learner. Let $Z$ denote the number of data points lie in $\mathcal{X}_{\mathrm{easy}}$ among the first $B$ random draw of examples. We see that $Z \sim \mathcal{B}(B, B^{-1}/2)$ follows a binomial distribution with $B$ trials and $B^{-1}/2$ success rate. By Markov inequality, we have

$$\mathbb{P}\left(Z \ge \frac{3}{2}\mathbb{E}[Z]\right) = \mathbb{P}\left(Z \ge \frac{3}{4}\right) \le \frac{2}{3}.$$

That being said, with probability at least $1/3$, there will be $Z = 0$ data point that randomly drawn from the easy region $\mathcal{X}_{\mathrm{easy}}$. We denote that event as $\mathcal{E}$. Since $\eta(x) = f^\star(x) = \frac{1}{2}$ for any $x \in \mathcal{X}_{\mathrm{hard}}$, any "uncertainty-based" active learner will query the label of any data point $x \in \mathcal{X}_{\mathrm{hard}}$. As a result, under event $\mathcal{E}$, the active learner will use up all the labeling budget in the first $B$ rounds and observe zero label for any data point $x \in \mathcal{X}_{\mathrm{easy}}$. Since the easy region $\mathcal{X}_{\mathrm{easy}}$ has measure $B^{-1}/2$ and $\theta_2^\star = \mathrm{unif}(\{\pm\frac{1}{2}\})$, any classification rule over the easy region would results in expected excess error lower bounded by $B^{-1}/4$. To summarize, with probability at least $\frac{1}{3}$, any "uncertainty-based" active learner without abstention suffers expected excess error $\Omega(B^{-1})$.

We now consider the classifier returned by Algorithm 1.[10] For the linear function considered in Example 1, we have $\mathrm{Pdim}(\mathcal{F}) \le 2$ (Haussler, 1989) and $\theta_{f^\star}^{\mathrm{val}}(\mathcal{F}, \gamma/2, \varepsilon) \le 2$ for any $\varepsilon \ge 0$ (see Appendix C). Thus, by setting $T = O(\frac{1}{\varepsilon\gamma} \cdot \log(\frac{1}{\varepsilon\gamma\delta}))$, with probability at least $1 - \delta$, Algorithm 1 return a classifier $\widehat{h}$ with Chow's excess error at most $\varepsilon$ and label complexity $O(\frac{1}{\gamma^2} \cdot \log^2(\frac{1}{\varepsilon\gamma}) \cdot \log(\frac{1}{\varepsilon\gamma\delta})) = \mathrm{poly}(\frac{1}{\gamma}, \log(\frac{1}{\varepsilon\gamma\delta}))$. Since $\widehat{h}$ enjoys proper abstention, it never abstains for $x \in \mathcal{X}_{\mathrm{easy}}$. Note that we have $\eta(x) = \frac{1}{2}$ for any $x \in \mathcal{X}_{\mathrm{hard}}$. By randomizing the decision of $\widehat{h}$ over the abstention region, we obtain a classifier with standard excess error at most $\varepsilon$. $\square$

---

[9]By restricting to learners that construct an open confidence interval containing $\eta(x)$, we do not consider the corner cases when $\mathsf{lcb}(x) = \frac{1}{2}$ or $\mathsf{ucb}(x) = \frac{1}{2}$ and the confidence interval close.

[10]The version that works with an infinite set of regression functions using concentration results presented in Lemma 4. Or, one can first discretie the set of regression function and then use the version presented in Algorithm 1.

# G   Omitted details for Section 4.1

We introduce a new perspective for designing and analyzing active learning algorithms in Appendix G.1 (with new notations introduced). Based on this new perspective, we present our algorithm and its theoretical guarantees in Appendix G.2. Supporting lemmas are deferred to Appendix G.3.

## G.1   The perspective: Regret minimization with selective sampling

We view active learning as a decision making problem: at each round, the learner selects an action, suffers a loss (that may not be observable), and decides to query the label or not. At a high level, the learner aims at *simultaneously* minimizing the regret and the number of queries; and will randomly return a classifier/decision rule at the end of the learning process.

The perspective is inspired by the seminal results derived in Dekel et al. (2012), where the authors study active learning with linear regression functions and focus on standard excess error guarantees. With this regret minimization perspective, we can also take advantage of fruitful results developed in the field of contextual bandits (Russo and Van Roy, 2013; Foster et al., 2020).

**Decision making for regret minimization.**   To formulate the regret minimization problem, we consider the action set $\mathcal{A} = \{+1, -1, \perp\}$, where the action $+1$ (resp. $-1$) represents labeling any data point $x \in \mathcal{X}$ as positive (resp. negative); and the action $\perp$ represents abstention. At each round $t \in [T]$, the learner observes a data point $x_t \in \mathcal{X}$ (which can be chosen by an adaptive adversary), takes an action $a_t \in \mathcal{A}$, and then suffers a loss, which is defined as

$$\ell_t(a_t) = \mathbb{1}(\text{sign}(y_t) \neq a_t, a_t \neq \perp) + \left(\frac{1}{2} - \gamma\right) \cdot \mathbb{1}(a_t = \perp).$$

We use $a_t^\star := \text{sign}(2f^\star(x_t) - 1) = \text{sign}(2\eta(x_t) - 1)$ to denote the action taken by the Bayes optimal classifier $h^\star \in \mathcal{H}$. Denote filtration $\mathfrak{F}_t := \sigma((x_i, y_i)_{i=1}^t)$. We define the (conditionally) expected regret at time step $t \in [T]$ as

$$\mathbf{Regret}_t = \mathbb{E}[\ell_t(a_t) - \ell_t(a_t^\star) \mid \mathfrak{F}_{t-1}].$$

The (conditionally) expected cumulative regret across $T$ rounds is defined as

$$\mathbf{Regret}(T) = \sum_{t=1}^T \mathbf{Regret}_t,$$

which is the target that the learner aims at minimizing.

**Selective querying for label efficiency.**   Besides choosing an action $a_t \in \mathcal{A}$ at each time step, our algorithm also determines *whether or not* to query the label $y_t$ with respect to $x_t$. Note that such selective querying protocol makes our problem different from contextual bandits (Russo and Van Roy, 2013; Foster et al., 2020): The loss $\ell_t(a_t)$ of an chosen $a_t$ may not be even observed.

We use $Q_t$ to indicate the query status at round $t$, i.e.,

$$Q_t = \mathbb{1}(\text{label } y_t \text{ of } x_t \text{ is queried}).$$

The learner also aims at minimizing the total number of queries across $T$ rounds, i.e., $\sum_{t=1}^T Q_t$.

**Connection to active learning.**   We consider the following learner for the above mentioned decision making problem. At each round, the learner constructs a classifier $\widehat{h}_t : \mathcal{X} \to \{+1, -1, \perp\}$ and a query function $g_t : \mathcal{X} \to \{0, 1\}$; the learner then takes action $a_t = \widehat{h}_t(x_t)$ and decides the query status as $Q_t = g_t(x_t)$.

Conditioned on $\mathfrak{F}_{t-1}$, taking expectation over $\ell_t(a_t)$ leads to the following equivalence.

$$
\begin{aligned}
\mathbb{E}[\ell_t(a_t) \mid \mathfrak{F}_{t-1}] &= \mathbb{E}\left[ \mathbb{1}(\mathrm{sign}(y_t) \neq a_t, a_t \neq \perp) + \left(\frac{1}{2} - \gamma\right) \cdot \mathbb{1}(a_t = \perp) \mid \mathfrak{F}_{t-1}\right] \\
&= \mathbb{E}\left[ \mathbb{1}\left(\mathrm{sign}(y_t) \neq \widehat{h}(x_t), \widehat{h}(x_t) \neq \perp\right) + \left(\frac{1}{2} - \gamma\right) \cdot \mathbb{1}\left(\widehat{h}(x_t) = \perp\right) \mid \mathfrak{F}_{t-1}\right] \\
&= \mathbb{P}_{(x,y)\sim\mathcal{D}_{\mathcal{X}\mathcal{Y}}}\left(\mathrm{sign}(y) \neq \widehat{h}(x), \widehat{h}(x) \neq \perp\right) + \left(\frac{1}{2} - \gamma\right) \cdot \mathbb{P}(\widehat{h}(x) = \perp) \\
&= \mathrm{err}_\gamma(\widehat{h}_t).
\end{aligned}
$$

This shows that the (conditionally) expected instantaneous loss precisely captures the Chow's error of classifier $\widehat{h}_t$. Similarly, we have

$$
\mathbb{E}[\ell_t(a_t^\star) \mid \mathfrak{F}_{t-1}] = \mathbb{P}_{(x,y)\sim\mathcal{D}_{\mathcal{X}\mathcal{Y}}}(\mathbb{1}(y \neq \mathrm{sign}(2\eta(x) - 1))) = \mathrm{err}(h^\star).
$$

Combining the above two results, we notice that the (conditionally) expected instantaneous *regret* exactly captures the Chow's excess error of classifier $\widehat{h}_t$, i.e.,

$$
\mathbf{Regret}_t = \mathrm{err}_\gamma(\widehat{h}_t) - \mathrm{err}(h^\star).
$$

Let $\widehat{h} \sim \mathrm{unif}(\{\widehat{h}_t\}_{t=1}^T)$ be a classifier randomly selected from all the constructed classifiers. Taking expectation with respect to this random selection procedure, we then have

$$
\mathbb{E}_{\widehat{h}\sim\mathrm{unif}(\{\widehat{h}_t\}_{t=1}^T)}[\mathrm{err}_\gamma(\widehat{h}) - \mathrm{err}(h^\star)] = \sum_{t=1}^{T}(\mathrm{err}_\gamma(\widehat{h}_t) - \mathrm{err}(h^\star))/T = \mathbf{Regret}(T)/T. \quad (11)
$$

That being said, the expected Chow's excess error of $\widehat{h}$ can be sublinear in $T$. If the total number of queries is logarithmic in $T$, this immediately implies learning a classifier with exponential savings in label complexity.

### G.2 Algorithm and main results

We present an algorithm that achieves constant label complexity next (Algorithm 2). Compared to Algorithm 1, Algorithm 2 drops the epoch scheduling, uses a sharper elimination rule for the active set (note that $\beta$ doesn't depend on $T$, due to applying optimal stopping theorem in Lemma 10), and is analyzed with respect to eluder dimension (Definition 7) instead of disagreement coefficient. As a result, we shave all three sources of $\log\frac{1}{\varepsilon}$, and achieve constant label complexity for general $\mathcal{F}$ (as long as it's finite and has finite eluder dimension). We abbreviate $\mathfrak{e} := \sup_{f^\star \in \mathcal{F}} \mathfrak{e}_{f^\star}(\mathcal{F}, \gamma/2)$.

---

**Algorithm 2** Efficient Active Learning with Abstention (Constant Label Complexity)

---

**Input:** Time horizon $T \in \mathbb{N}$, abstention parameter $\gamma \in (0, 1/2)$ and confidence level $\delta \in (0, 1)$.

1: Initialize $\widehat{\mathcal{H}} := \emptyset$. Set $T := O(\frac{\mathfrak{e}}{\varepsilon \gamma} \cdot \log(\frac{|\mathcal{F}|}{\delta}))$ and $\beta := \frac{1}{2} \log\left(\frac{|\mathcal{F}|}{\delta}\right)$.

2: **for** $t = 1, 2, \ldots, T$ **do**

3:     Get $\widehat{f}_t := \arg\min_{f \in \mathcal{F}} \sum_{i < t} Q_i (f(x_i) - y_i)^2$.
         `// We use` $Q_t \in \{0, 1\}$ `to indicate whether the label of` $x_t$ `is queried.`

4:     (Implicitly) Construct active set of regression function $\mathcal{F}_t \subseteq \mathcal{F}$ as

$$\mathcal{F}_t := \left\{ f \in \mathcal{F} : \sum_{i=1}^{t-1} Q_i (f(x_i) - y_i)^2 \leq \sum_{i=1}^{t-1} Q_i (\widehat{f}_t(x_i) - y_i)^2 + \beta \right\}.$$

5:     Construct classifier $\widehat{h}_t : \mathcal{X} \to \{+1, -1, \bot\}$ as

$$\widehat{h}_t(x) := \begin{cases} \bot, & \text{if } [\mathsf{lcb}(x; \mathcal{F}_t), \mathsf{ucb}(x; \mathcal{F}_t)] \subseteq \left[\frac{1}{2} - \gamma, \frac{1}{2} + \gamma\right]; \\ \mathrm{sign}(2\widehat{f}_t(x) - 1), & \text{o.w.} \end{cases}$$

Update $\widehat{\mathcal{H}} = \widehat{\mathcal{H}} \cup \{\widehat{h}_t\}$. Construct query function $g_m : \mathcal{X} \to \{0, 1\}$ as

$$g_t(x) := \mathbb{1}\left(\frac{1}{2} \in (\mathsf{lcb}(x; \mathcal{F}_t), \mathsf{ucb}(x; \mathcal{F}_t))\right) \cdot \mathbb{1}(\widehat{h}_t(x) \neq \bot).$$

6:     Observe $x_t \sim \mathcal{D}_\mathcal{X}$. Take action $a_t := \widehat{h}_t(x_t)$. Set $Q_t := g_t(x_t)$.

7:     **if** $Q_t = 1$ **then**

8:         Query the label $y_t$ of $x_t$.

9: **Return** $\widehat{h} := \mathrm{unif}(\widehat{\mathcal{H}})$.

---

Before proving Theorem 8. We define some notations that are specialized to Appendix G.

We define filtrations $\mathfrak{F}_{t-1} := \sigma(x_1, y_1, \ldots, x_{t-1}, y_{t-1})$ and $\overline{\mathfrak{F}}_{t-1} := \sigma(x_1, y_1, \ldots, x_t)$. Note that we additionally include the data point $x_t$ in the filtration $\overline{\mathfrak{F}}_{t-1}$ at time step $t - 1$. We denote $\mathbb{E}_t[\cdot] := \mathbb{E}[\cdot \mid \overline{\mathfrak{F}}_{t-1}]$. For any $t \in [T]$, we denote $M_t(f) := Q_t((f(x_t) - y_t)^2 - (f^\star(x_t) - y_t)^2)$. We have $\sum_{i=1}^{\tau} \mathbb{E}_t[M_t(f)] = \sum_{t=1}^{\tau} Q_t(f(x_t) - f^\star(x_t))^2$. For any given data point $x_t \in \mathcal{X}$, we use abbreviations

$$\mathsf{ucb}_t := \mathsf{ucb}(x_t; \mathcal{F}_t) = \sup_{f \in \mathcal{F}_t} f(x_t) \quad \text{and} \quad \mathsf{lcb}_t := \mathsf{lcb}(x_t; \mathcal{F}_t) = \inf_{f \in \mathcal{F}_t} f(x_t)$$

to denote the upper and lower confidence bounds of $\eta(x_t) = f^\star(x_t)$. We also denote

$$w_t := \mathsf{ucb}_t - \mathsf{lcb}_t = \sup_{f, f' \in \mathcal{F}_t} |f(x_t) - f'(x_t)|$$

as the width of confidence interval.

**Theorem 8.** *With probability at least $1 - 2\delta$, Algorithm 2 returns a classifier with expected Chow's excess error at most $\varepsilon$ and label complexity $O(\frac{\mathfrak{e} \cdot \log(|\mathcal{F}|/\delta)}{\gamma^2})$, which is independent of $\frac{1}{\varepsilon}$.*

*Proof.* We first analyze the label complexity of Algorithm 2. Note that Algorithm 2 constructs $\widehat{h}_t$ and $g_t$ in forms similar to the ones constructed in Algorithm 1, and Lemma 6 holds for Algorithm 2 as well. Based on Lemma 6, we have $Q_t = g_t(x_t) = 1 \implies w_t > \gamma$. Thus, taking $\zeta = \gamma$ in Lemma 13 leads to

$$\sum_{t=1}^{T} \mathbb{1}(Q_t = 1) < \frac{17 \log(2|\mathcal{F}|/\delta)}{2\gamma^2} \cdot \mathfrak{e}_{f^\star}(\mathcal{F}, \gamma/2),$$

*with probability one.* The label complexity of Algorithm 2 is then upper bounded by a constant as long as $\mathfrak{e}_{f^\star}(\mathcal{F}, \gamma/2)$ is upper bounded by a constant (which has no dependence on $T$ or $\frac{1}{\varepsilon}$).

We next analyze the excess error of $\widehat{h}$. We consider the good event $\mathcal{E}$ defined in Lemma 12, which holds true with probability at least $1 - \delta$. Under event $\mathcal{E}$, Lemma 15 shows that

$$\sum_{t=1}^{T} \mathbb{E}[\ell_t(a_t) - \ell_t(a_t^\star) \mid \overline{\mathfrak{F}}_{t-1}] \leq \frac{17\sqrt{2}\beta}{\gamma} \cdot \mathfrak{e}_{f^\star}(\mathcal{F}, \gamma/2).$$

Since

$$\mathbb{E}\Big[\mathbb{E}[\ell_t(a_t) - \ell_t(a_t^\star) \mid \overline{\mathfrak{F}}_{t-1}] \mid \mathfrak{F}_{t-1}\Big] = \mathbb{E}[\ell_t(a_t) - \ell_t(a_t^\star) \mid \mathfrak{F}_{t-1}],$$

and $|\mathbb{E}[\ell_t(a_t) - \ell_t(a_t^\star) \mid \overline{\mathfrak{F}}_{t-1}]| \leq 1$ by construction, applying Lemma 2 with respect to $\mathbb{E}[\ell_t(a_t) - \ell_t(a_t^\star) \mid \overline{\mathfrak{F}}_{t-1}]$ further leads to

$$\mathbf{Regret}(T) = \sum_{t=1}^{T} \mathbb{E}[\ell_t(a_t) - \ell_t(a_t^\star) \mid \mathfrak{F}_{t-1}] \leq \frac{34\sqrt{2}\beta}{\gamma} \cdot \mathfrak{e}_{f^\star}(\mathcal{F}, \gamma/2) + 8\log(2\delta^{-1}),$$

with probability at least $1 - 2\delta$ (due to the additional application of Lemma 2). Since $\widehat{h} \sim \mathrm{unif}(\widehat{\mathcal{H}})$, based on Eq. (11), we thus know that

$$\mathbb{E}_{\widehat{h} \sim \mathrm{unif}(\widehat{\mathcal{H}})}[\mathrm{err}_\gamma(\widehat{h}) - \mathrm{err}(h^\star)] = \sum_{t=1}^{T} \big(\mathrm{err}_\gamma(\widehat{h}_t) - \mathrm{err}(h^\star)\big)/T$$

$$\leq \left(\frac{34\sqrt{2}\beta}{\gamma} \cdot \mathfrak{e}_{f^\star}(\mathcal{F}, \gamma/2) + 8\log\big(2\delta^{-1}\big)\right)/T$$

Since $T := O(\frac{\mathfrak{e}}{\varepsilon\gamma} \cdot \log(\frac{|\mathcal{F}|}{\delta}))$, we then know that the expected Chow's excess error is at most $\varepsilon$. $\qquad\square$

**Theorem 10.** *Consider the setting where the data points $\{x_t\}_{t=1}^{T}$ are chosen by an adaptive adversary with $y_t \sim \mathcal{D}_{\mathcal{Y}|x_t}$. With probability at least $1 - \delta$, Algorithm 2 simultaneously guarantees*

$$\sum_{t=1}^{T} \mathbb{E}[\ell_t(a_t) - \ell_t(a_t^\star) \mid \overline{\mathfrak{F}}_{t-1}] \leq \frac{34\sqrt{2}\beta}{\gamma} \cdot \mathfrak{e}_{f^\star}(\mathcal{F}, \gamma/2),$$

*and*

$$\sum_{t=1}^{T} \mathbb{1}(Q_t = 1) < \frac{17\log(2|\mathcal{F}|/\delta)}{2\gamma^2} \cdot \mathfrak{e}_{f^\star}(\mathcal{F}, \gamma/2).$$

*Proof.* The proof follows the same analysis as in the first part of the proof of Theorem 8 (simply stopped at the step with conditioning on $\overline{\mathfrak{F}}_{t-1}$). $\qquad\square$

We redefine $\mathfrak{e} := \sup_{f^\star \in \mathcal{F}} \mathfrak{e}_{f^\star}(\mathcal{F}, \gamma/4)$ in the following Theorem 11 to account for the induced approximation error in efficient implementation.

**Theorem 11.** *Algorithm 2 can be efficiently implemented via the regression oracle and enjoys the same theoretical guarantees stated in Theorem 8 or Theorem 10. The number of oracle calls needed is $O(\frac{\mathfrak{e}}{\varepsilon\gamma^3} \cdot \log(\frac{|\mathcal{F}|}{\delta}) \cdot \log(\frac{1}{\gamma}))$ for a general set of regression functions $\mathcal{F}$, and $O(\frac{\mathfrak{e}}{\varepsilon\gamma} \cdot \log(\frac{|\mathcal{F}|}{\delta}) \cdot \log(\frac{1}{\gamma}))$ when $\mathcal{F}$ is convex and closed under pointwise convergence. The per-example inference time of the learned $\widehat{h}_M$ is $O(\frac{1}{\gamma^2}\log\frac{1}{\gamma})$ for general $\mathcal{F}$, and $O(\log\frac{1}{\gamma})$ when $\mathcal{F}$ is convex and closed under pointwise convergence.*

*Proof.* Denote $\mathcal{B}_t := \{(x_i, Q_i, y_i)\}_{i=1}^{\tau_{t-1}}$ At any time step $t \in [T]$ of Algorithm 2, we construct classifier $\widehat{h}_t$ and query function $g_t$ with approximated confidence bounds, i.e.,

$$\widehat{\mathsf{lcb}}(x; \mathcal{F}_t) := \mathbf{Alg}_{\mathsf{lcb}}(x; \mathcal{B}_t, \beta_t, \alpha) \quad \text{and} \quad \widehat{\mathsf{ucb}}(x; \mathcal{F}_t) := \mathbf{Alg}_{\mathsf{ucb}}(x; \mathcal{B}_t, \beta_t, \alpha),$$

where $\mathbf{Alg}_{\mathsf{lcb}}$ and $\mathbf{Alg}_{\mathsf{ucb}}$ are subroutines discussed in Proposition 8 and $\alpha := \frac{\gamma}{4}$.

Since the theoretical analysis of Theorem 8 and Theorem 10 do not require an non-increasing (with respect to time step $t$) sampling region, i.e., $\{x \in \mathcal{X} : g_t(x) = 1\}$, we only need to approximate the confidence intervals at $\frac{\gamma}{4}$ level. This slightly save the computational complexity compared to Theorem 5, which approximates the confidence interval at $\frac{\gamma}{4\lceil \log T \rceil}$ level. The rest of the analysis of computational complexity follows similar steps in the proof of Theorem 5. $\qquad\square$

## G.3 Supporting lemmas

Consider a sequence of random variables $(Z_t)_{t \in \mathbb{N}}$ adapted to filtration $\bar{\bar{\mathfrak{F}}}_t$. We assume that $\mathbb{E}[\exp(\lambda Z_t)] < \infty$ for any $\lambda$ and $\mu_t := \mathbb{E}[Z_t \mid \bar{\bar{\mathfrak{F}}}_{t-1}]$. We also denote

$$\psi_t(\lambda) := \log \mathbb{E}[\exp(\lambda \cdot (Z_t - \mu_t)) \mid \bar{\bar{\mathfrak{F}}}_{t-1}].$$

**Lemma 10** (Russo and Van Roy (2013))**.** *With notations defined above. For any $\lambda \geq 0$ and $\delta > 0$, we have*

$$\mathbb{P}\left( \forall \tau \in \mathbb{N}, \sum_{t=1}^{\tau} \lambda Z_t \leq \sum_{t=1}^{\tau} (\lambda \mu_t + \psi_t(\lambda)) + \log\left(\frac{1}{\delta}\right) \right) \geq 1 - \delta. \tag{12}$$

**Lemma 11.** *Fix any $\delta \in (0,1)$. For any $\tau \in [T]$, with probability at least $1 - \delta$, we have*

$$\sum_{t=1}^{\tau} M_t(f) \leq \sum_{t=1}^{\tau} \frac{3}{2} \mathbb{E}_t[M_t(f)] + C_\delta,$$

*and*

$$\sum_{t=1}^{\tau} \mathbb{E}_t[M_t(f)] \leq 2 \sum_{t=1}^{\tau} M_t(f) + C_\delta,$$

*where $C_\delta := \log\left(\frac{2|\mathcal{F}|}{\delta}\right)$.*

*Proof.* Fix any $f \in \mathcal{F}$. We take $Z_t = M_t(f)$ in Lemma 10. We can rewrite

$$Z_t = Q_t\big((f(x_t) - f^\star(x_t))^2 + 2(f(x_t) - f^\star(x_t))\varepsilon_t\big),$$

where we use the notation $\varepsilon_t := f^\star(x_t) - y_t$. Since $\mathbb{E}_t[\varepsilon_t] = 0$ and $\mathbb{E}_t[\exp(\lambda \varepsilon_t) \mid \bar{\bar{\mathfrak{F}}}_{t-1}] \leq \exp(\frac{\lambda^2}{8})$ a.s. by assumption, we have

$$\mu_t = \mathbb{E}_t[Z_t] = Q_t(f(x_t) - f^\star(x_t))^2,$$

and

$$\begin{aligned} \psi_t(\lambda) &= \log \mathbb{E}[\exp(\lambda \cdot (Z_t - \mu_t)) \mid \bar{\bar{\mathfrak{F}}}_{t-1}] \\ &= \log \mathbb{E}_t[\exp(2\lambda Q_t(f(x_t) - f^\star(x_t) \cdot \varepsilon_t))] \\ &\leq \frac{\lambda^2 (Q_t(f(x_t) - f^\star(x_t))^2}{2} \\ &= \frac{\lambda^2 \mu_t}{2}, \end{aligned}$$

where the last line comes from the fact that $Q_t \in \{0, 1\}$. We can similarly upper bound $\mathbb{E}[\exp(\lambda Z_t)] = \mathbb{E}[\mathbb{E}_t[\exp(\lambda Z_t)]] \leq \exp(\lambda + \frac{\lambda^2}{2})$ by noticing the range fact that $\mu_t \leq 1$.

Plugging the above results into Lemma 10 with $\lambda = 1$ leads to

$$\sum_{t=1}^{\tau} M_t(f) \leq \sum_{t=1}^{\tau} \frac{3}{2} \mathbb{E}_t[M_t(f)] + \log \delta^{-1}.$$

Following the same procedures above with $Z_t = -M_t(f)$ and $\lambda = 1$ leads to

$$\sum_{t=1}^{\tau} \frac{3}{2} \mathbb{E}_t[M_t(f)] \leq 2 \sum_{t=1}^{\tau} M_t(f) + \log \delta^{-1}.$$

The final guarantees comes from taking a union abound over $f \in \mathcal{F}$ and splitting the probability for both directions. $\square$

We use $\mathcal{E}$ to denote the good event considered in Lemma 11, we use it through out the rest of this section.

**Lemma 12.** *With probability at least $1 - \delta$, the followings hold true:*

1. *$f^\star \in \mathcal{F}_t$ for any $t \in [T]$.*

2. *$\sum_{t=1}^{\tau-1} \mathbb{E}_t[M_t(f)] \leq 2C_\delta$ for any $f \in \mathcal{F}_\tau$.*

*Proof.* The first statement immediately follows from [Lemma 11](#) (the second inequality) and the fact that $\beta := C_\delta/2$ in [Algorithm 2](#).

For any $f \in \mathcal{F}_\tau$, we have

$$
\begin{aligned}
\sum_{t=1}^{\tau-1} \mathbb{E}_t[M_t(f)] &\leq 2 \sum_{t=1}^{\tau-1} Q_t\big((f(x_t) - y_t)^2 - (f^\star(x_t) - y_t)^2\big) + C_\delta \\
&\leq 2 \sum_{t=1}^{\tau-1} Q_t\Big((f(x_t) - y_t)^2 - (\widehat{f}_\tau(x_t) - y_t)^2\Big) + C_\delta \\
&\leq 2C_\delta,
\end{aligned}
\tag{13}
$$

where the first line comes from [Lemma 11](#), the second line comes from the fact that $\widehat{f}_\tau$ is the minimize among $\mathcal{F}_\tau$, and the third line comes from the fact that $f \in \mathcal{F}_\tau$ and $2\beta = C_\delta$. $\square$

**Lemma 13.** *For any $\zeta > 0$, with probability $1$, we have*

$$
\sum_{t=1}^{T} \mathbb{1}(Q_t = 1) \cdot \mathbb{1}(w_t > \zeta) < \left(\frac{16\beta}{\zeta^2} + 1\right) \cdot \mathfrak{e}_{f^\star}(\mathcal{F}, \zeta/2).
$$

**Remark 4.** *Similar upper bound has been established in the contextual bandit settings for $\sum_{t=1}^{T} \mathbb{1}(w_t > \zeta)$ ([Russo and Van Roy, 2013](#); [Foster et al., 2020](#)). Our results is established with an additional $\mathbb{1}(Q_t = 1)$ term due to selective querying in active learning.*

*Proof.* We give some definitions first. We say that $x$ is $\zeta$-independent of a sequence $x_1, \ldots, x_\tau$ if there exists a $f \in \mathcal{F}$ such that $|f(x) - f^\star(x)| > \zeta$ and $\sum_{i \leq \tau}(f(x_i) - f^\star(x_i))^2 \leq \zeta^2$. We say that $x$ is $\zeta$-dependent of $x_1, \ldots, x_\tau$ if we have $|f(x) - f^\star(x)| \leq \zeta$ for all $f \in \mathcal{F}$ such that $\sum_{i \leq \tau}(f(x_i) - f^\star(x_i))^2 \leq \zeta^2$. The eluder dimension $\check{\mathfrak{e}}_{f^\star}(\mathcal{F}, \zeta)$ can be equivalently defined as the length of the longest sequence $x_1, \ldots, x_\tau$ such that each $x_i$ is $\zeta$-independent of all its predecessors.

For any $t \in [T]$, and we denote $\mathcal{S}_t = \{x_i : Q_i = g_i(x_i) = 1, i \in [t]\}$ as the *queried* data points up to time step $t$. We assume that $|\mathcal{S}_t| = \tau$ and denote $\mathcal{S}_t = (x_{g(1)}, \ldots, x_{g(\tau)})$, where $g(i)$ represents the time step where the $i$-th *queried* data point is queried.

**Claim 1.** For any $j \in [\tau]$, $x_{g(j)}$ is $\frac{\zeta}{2}$-dependent on at most $\frac{16\beta}{\zeta^2}$ disjoint subsequences of $x_{g(1)}, \ldots, x_{g(j-1)}$.

For any $x_{g(j)} \in \mathcal{S}_t$, recall that

$$
w_{g(j)} = \mathsf{ucb}_{g(j)} - \mathsf{lcb}_{g(j)} = \max_{f, f' \in \mathcal{F}_{g(j)}} |f(x_t) - f'(x_t)|.
$$

If $m_{g(j)} > \zeta$, there must exists a $f \in \mathcal{F}_{g(j)}$ such that $\big|f(x_{g(j)}) - f^\star(x_{g(j)})\big| > \frac{\zeta}{2}$. Focus on this specific $f \in \mathcal{F}_{g(j)} \subseteq \mathcal{F}$. If $x_{g(j)}$ is $\frac{\zeta}{2}$-dependent on a subsequence $x_{g(i_1)}, \ldots, x_{g(i_m)}$ (of $x_{g(1)}, \ldots, x_{g(j-1)}$), we must have

$$
\sum_{k \leq m}(f(x_{g(i_k)}) - f^\star(x_{g(i_k)}))^2 > \frac{\zeta^2}{4}.
$$

Suppose $x_{g(j)}$ is $\frac{\zeta}{2}$-dependent on $K$ *disjoint* subsequences of $x_{g(1)}, \ldots, x_{g(j-1)}$, according to [Lemma 12](#), we must have

$$
K \cdot \frac{\zeta^2}{4} < \sum_{i < j}(f(x_{g(i)}) - f^\star(x_{g(i)}))^2 = \sum_{k < g(j)} Q_k(f(x_k) - f^\star(x_k))^2 \leq 4\beta,
$$

which implies that $K < \frac{16\beta}{\zeta^2}$.

**Claim 2.** Denote $d := \check{\mathfrak{e}}_{f^\star}(\mathcal{F}, \zeta/2) \geq 1$ and $K = \lfloor \frac{\tau-1}{d} \rfloor$. There must exists a $j \in [\tau]$ such that $x_{g(j)}$ is $\frac{\zeta}{2}$-dependent on at least $K$ disjoint subsequences of $x_{g(1)}, \ldots, x_{g(j-1)}$.

We initialize $K$ subsequences $\mathcal{C}_i = \{x_{g(i)}\}$. If $x_{g(K+1)}$ is $\frac{\zeta}{2}$-dependent on each $\mathcal{C}_i$, we are done. If not, select a subsequence $\mathcal{C}_i$ such that $x_{g(K+1)}$ is $\frac{\zeta}{2}$-independent of and add $x_{g(K+1)}$ into this subsequence. Repeat this procedure with $j > K + 1$ until $x_{g(j)}$ is $\frac{\zeta}{2}$-dependent of all $\mathcal{C}_i$ or $j = \tau$. In the later case, we have $\sum_{i \leq K} |\mathcal{C}_i| = \tau - 1 \geq Kd$. Since $|\mathcal{C}_i| \leq d$ by definition, we must have $|\mathcal{C}_i| = d$ for all $i \in [K]$. As a result, $x_{g(\tau)}$ must be $\frac{\zeta}{2}$-dependent of all $\mathcal{C}_i$.

It's easy to check that $\lfloor \frac{\tau-1}{d} \rfloor \geq \frac{\tau}{d} - 1$. Combining Claim 1 and 2, we have

$$\frac{\tau}{d} - 1 \leq \left\lfloor \frac{\tau - 1}{d} \right\rfloor \leq K < \frac{16\beta}{\zeta^2}.$$

Rearranging leads to the desired result. $\qquad\square$

The following Lemma 14 is a restatement of Lemma 9 in the regret minimization setting.

**Lemma 14.** *If $Q_t = 0$, we have $\mathbb{E}\big[\ell_t(a_t) - \ell_t(a_t^\star) \mid \bar{\mathfrak{F}}_{t-1}\big] \leq 0$.*

*Proof.* Recall we have $a_t = \widehat{h}_t(x_t)$. We then have

$\mathbb{E}\big[\ell_t(a_t) - \ell_t(a_t^\star) \mid \bar{\mathfrak{F}}_{t-1}\big]$
$= \mathbb{P}_{y_t|x_t}\big(y_t \neq \operatorname{sign}(\widehat{h}_t(x_t))\big) \cdot \mathbb{1}\big(\widehat{h}_t(x_t) \neq \bot\big) + (1/2 - \gamma) \cdot \mathbb{1}\big(\widehat{h}_t(x_t) = \bot\big) - \mathbb{P}_{y_t|x_t}\big(y_t \neq \operatorname{sign}(h^\star(x_t))\big)$
$= \mathbb{1}\big(\widehat{h}_t(x_t) \neq \bot\big) \cdot \big(\mathbb{P}_{y_t|x_t}\big(y_t \neq \operatorname{sign}(\widehat{h}_t(x_t))\big) - \mathbb{P}_{y_t|x_t}\big(y_t \neq \operatorname{sign}(h^\star(x_t))\big)\big)$
$\quad + \mathbb{1}\big(\widehat{h}_t(x_t) = \bot\big) \cdot \big((1/2 - \gamma) - \mathbb{P}_{y_t|x_t}\big(y_t \neq \operatorname{sign}(h^\star(x_t))\big)\big).$

We now analyze the event $\{Q_t = 0\}$ in two cases.

**Case 1: $\widehat{h}_t(x_t) = \bot$.**

Since $\eta(x_t) = f^\star(x_t) \in [\mathsf{lcb}_t, \mathsf{ucb}_t]$, we further know that $\eta(x_t) \in [\frac{1}{2} - \gamma, \frac{1}{2} + \gamma]$ and thus $\mathbb{P}_{y_t|x_t}\big(y_t \neq \operatorname{sign}(h^\star(x_t))\big) \geq \frac{1}{2} - \gamma$. As a result, we have $\mathbb{E}\big[\ell_t(a_t) - \ell_t(a_t^\star) \mid \bar{\mathfrak{F}}_{t-1}\big] \leq 0$.

**Case 2: $\widehat{h}_t(x_t) \neq \bot$ but $\frac{1}{2} \notin (\mathsf{lcb}_t, \mathsf{ucb}_t)$.**

In this case, we know that $\operatorname{sign}(\widehat{h}_t(x_t)) = \operatorname{sign}(h^\star(x_t))$ whenever $\eta(x_t) \in [\mathsf{lcb}_t, \mathsf{ucb}_t]$. As a result, we have $\mathbb{E}\big[\ell_t(a_t) - \ell_t(a_t^\star) \mid \bar{\mathfrak{F}}_{t-1}\big] = 0$. $\qquad\square$

**Lemma 15.** *Assume $\mu(x_t) \in [\mathsf{lcb}_t, \mathsf{ucb}_t]$ and $f^\star$ is not eliminated across all $t \in [T]$. We have*

$$\sum_{t=1}^T \mathbb{E}[\ell_t(a_t) - \ell_t(a_t^\star) \mid \bar{\mathfrak{F}}_{t-1}] \leq \frac{17\sqrt{2}\beta}{\gamma} \cdot \mathfrak{e}_{f^\star}(\mathcal{F}, \gamma/2). \tag{14}$$

*Proof.* Lemma 14 shows that non-positive conditional regret is incurred at whenever $Q_t = 0$, we then have

$$\sum_{t=1}^T \mathbb{E}[\ell_t(a_t) - \ell_t(a_t^\star) \mid \bar{\mathfrak{F}}_{t-1}] \leq \sum_{t=1}^T \mathbb{1}(Q_t = 1)\mathbb{E}\big[\ell_t(a_t) - \ell_t(a_t^\star) \mid \bar{\mathfrak{F}}_{t-1}\big]$$

$$= \sum_{t=1}^T \mathbb{1}(Q_t = 1) \cdot \mathbb{1}(w_t > \gamma) \cdot |2f^\star(x_t) - 1|$$

$$\leq \sum_{t=1}^T \mathbb{1}(Q_t = 1) \cdot \mathbb{1}(w_t > \gamma) \cdot 2w_t,$$

where we use Lemma 6 and Lemma 14 on the second line; and the last line comes from the fact that $|f^\star - \frac{1}{2}| \leq w_t$ whenever $f^\star$ is not eliminated and a query is issued. We can directly apply $w_t \leq 1$ and Lemma 13 to bound the above terms by $\widetilde{O}(\frac{\mathfrak{e}_{f^\star}(\mathcal{F},\gamma/2)}{\gamma^2})$, which has slightly worse dependence on $\gamma$. Following Foster et al. (2020), we take a slightly tighter analysis below.

Let $\mathcal{S}_T := \{x_i : Q_i = 1, i \in [T]\}$ denote the set of queried data points. Suppose $|\mathcal{S}_T| = \tau$. Let $i_1, \ldots, i_\tau$ be a reordering of indices within $\mathcal{S}_T$ such that $w_{i_1}(x_{i_1}) \geq w_{i_2}(x_{i_2}) \geq \ldots \geq w_{i_\tau}(x_{i_\tau})$. Consider any index $t \in [\tau]$ such that $w_{i_t}(x_{i_t}) \geq \gamma$. For any $\zeta \geq \gamma$, Lemma 13 implies that

$$t \leq \sum_{t=1}^{T} \mathbb{1}(Q_t = 1) \cdot \mathbb{1}(w_t(x_t) > \zeta) \leq \frac{17\beta}{\zeta^2} \cdot \mathfrak{e}_{f^\star}(\mathcal{F}, \zeta/2) \leq \frac{17\beta}{\zeta^2} \cdot \mathfrak{e}_{f^\star}(\mathcal{F}, \gamma/2). \qquad (15)$$

Taking $\zeta = w_{i_t}(x_{i_t})$ in Eq. (15) leads to the fact that

$$w_{i_t}(x_{i_t}) \leq \sqrt{\frac{17\beta \cdot \mathfrak{e}_{f^\star}(\mathcal{F}, \gamma/2)}{t}}.$$

Taking $\zeta = \gamma$ in Eq. (15) leads to the fact that

$$\tau \leq \frac{17\beta}{\gamma^2} \cdot \mathfrak{e}_{f^\star}(\mathcal{F}, \gamma/2).$$

We now have

$$\sum_{t=1}^{T} \mathbb{1}(Q_t = 1) \cdot \mathbb{1}(w_t > \gamma) \cdot 2w_t = \sum_{t=1}^{\tau} \mathbb{1}(w_{i_t} > \gamma) \cdot 2w_{i_t}(x_{i_t})$$

$$\leq 2 \sum_{t=1}^{\tau} \sqrt{\frac{17\beta \cdot \mathfrak{e}_{f^\star}(\mathcal{F}, \gamma/2)}{t}}$$

$$\leq \sqrt{34\beta \cdot \mathfrak{e}_{f^\star}(\mathcal{F}, \gamma/2) \cdot \tau}$$

$$\leq \frac{17\sqrt{2}\beta}{\gamma} \cdot \mathfrak{e}_{f^\star}(\mathcal{F}, \gamma/2).$$

$\square$

## H   Omitted details for Section 4.2

### H.1   Algorithm and main results

Algorithm 3 achieves the guarantees stated in Theorem 9. Theorem 9 is proved based on supporting lemmas derived in Appendix H.3. Note that, under the condition $\kappa \leq \varepsilon$, we still compete against the Bayes classifier $h^\star = h_{f^\star}$ in the analysis of Chow's excess error Eq. (2).

**Theorem 9.** *Suppose $\kappa \leq \varepsilon$. With probability at least $1 - 2\delta$, Algorithm 3 returns a classifier with Chow's excess error $O(\varepsilon \cdot \bar{\theta} \cdot \log(\frac{\mathrm{Pdim}(\mathcal{F})}{\varepsilon \gamma \delta}))$ and label complexity $O(\frac{\bar{\theta} \, \mathrm{Pdim}(\mathcal{F})}{\gamma^2} \cdot \log^2(\frac{\mathrm{Pdim}(\mathcal{F})}{\varepsilon \gamma}) \cdot \log(\frac{\mathrm{Pdim}(\mathcal{F})}{\varepsilon \gamma \delta}))$.*

*Proof.* We analyze under the good event $\mathcal{E}$ defined in Lemma 4, which holds with probability at least $1 - \delta$. Note that all supporting lemmas stated in Appendix H.3 hold true under this event.

We analyze the Chow's excess error of $\widehat{h}_m$, which is measurable with respect to $\mathfrak{F}_{\tau_{m-1}}$. For any $x \in \mathcal{X}$, if $g_m(x) = 0$, Lemma 20 implies that $\mathrm{excess}_\gamma(\widehat{h}_m; x) \leq 2\kappa$. If $g_m(x) = 1$, we know that $\widehat{h}_m(x) \neq \perp$ and $\frac{1}{2} \in (\mathrm{lcb}(x; \mathcal{F}_m), \mathrm{ucb}(x; \mathcal{F}_m))$. Since $\bar{f} \in \mathcal{F}_m$ by Lemma 17 and $\sup_{x \in \mathcal{X}} |\bar{f}(x) - f^\star(x)| \leq \kappa$ by assumption. The error incurred in this case is upper bounded by

$$\mathrm{excess}_\gamma(\widehat{h}_m; x) \leq 2|f^\star(x) - 1/2|$$

$$\leq 2\kappa + 2|\bar{f}(x) - 1/2|$$

$$\leq 2\kappa + 2w(x; \mathcal{F}_m).$$

**Algorithm 3** Efficient Active Learning with Abstention under Misspecification

---

**Input:** Accuracy level $\varepsilon > 0$, abstention parameter $\gamma \in (\varepsilon, 1/2)$ and confidence level $\delta \in (0, 1)$.

1: Define $T := \frac{\mathrm{Pdim}(\mathcal{F})}{\varepsilon\,\gamma}$, $M := \lceil \log_2 T \rceil$ and $C_\delta := O(\mathrm{Pdim}(\mathcal{F}) \cdot \log(T/\delta))$.

2: Define $\tau_m := 2^m$ for $m \geq 1$, $\tau_0 = 0$ and $\beta_m := (M - m + 1) \cdot \left(2\varepsilon^2 \tau_{M-1} + 2C_\delta\right)$.

3: **for** epoch $m = 1, 2, \ldots, M$ **do**

4:      Get $\widehat{f}_m := \arg\min_{f \in \mathcal{F}} \sum_{t=1}^{\tau_{m-1}} Q_t(f(x_t) - y_t)^2$.
                 // We use $Q_t \in \{0, 1\}$ to indicate whether the label of $x_t$ is queried.

5:      (Implicitly) Construct active set of regression function $\mathcal{F}_m \subseteq \mathcal{F}$ as

$$\mathcal{F}_m := \left\{ f \in \mathcal{F} : \sum_{t=1}^{\tau_{m-1}} Q_t(f(x_t) - y_t)^2 \leq \sum_{t=1}^{\tau_{m-1}} Q_t(\widehat{f}_m(x_t) - y_t)^2 + \beta_m \right\}.$$

6:      Construct classifier $\widehat{h}_m : \mathcal{X} \to \{+1, -1, \bot\}$ as

$$\widehat{h}_m(x) := \begin{cases} \bot, & \text{if } [\mathsf{lcb}(x; \mathcal{F}_m), \mathsf{ucb}(x; \mathcal{F}_m)] \subseteq \left[\frac{1}{2} - \gamma, \frac{1}{2} + \gamma\right]; \\ \mathrm{sign}(2\widehat{f}_m(x) - 1), & \text{o.w.} \end{cases}$$

     and query function $g_m : \mathcal{X} \to \{0, 1\}$ as

$$g_m(x) := \mathbb{1}\left(\frac{1}{2} \in (\mathsf{lcb}(x; \mathcal{F}_m), \mathsf{ucb}(x; \mathcal{F}_m))\right) \cdot \mathbb{1}(\widehat{h}_m(x) \neq \bot).$$

7:      **if** epoch $m = M$ **then**

8:          **Return** classifier $\widehat{h}_M$.

9:      **for** time $t = \tau_{m-1} + 1, \ldots, \tau_m$ **do**

10:        Observe $x_t \sim \mathcal{D}_{\mathcal{X}}$. Set $Q_t := g_m(x_t)$.

11:        **if** $Q_t = 1$ **then**

12:          Query the label $y_t$ of $x_t$.

---

Combining these two cases together, we have

$$\mathsf{excess}_\gamma(\widehat{h}_m) \leq 2\kappa + 2\mathbb{E}_{x \sim \mathcal{D}_{\mathcal{X}}}[\mathbb{1}(g_m(x) = 1) \cdot w(x; \mathcal{F}_m)].$$

Take $m = M$ and apply Lemma 19 leads to the following guarantee.

$$\begin{aligned} \mathsf{excess}_\gamma(\widehat{h}_M) &\leq 2\kappa + \frac{72\beta_M}{\tau_{M-1}\gamma} \cdot \theta_{\bar{f}}^{\mathrm{val}}\left(\mathcal{F}, \gamma/2, \sqrt{\beta_M/\tau_{M-1}}\right) \\ &\leq 2\kappa + O\left(\frac{\varepsilon^2}{\gamma} + \frac{\mathrm{Pdim}(\mathcal{F}) \cdot \log(T/\delta)}{T\,\gamma}\right) \cdot \theta_{\bar{f}}^{\mathrm{val}}\left(\mathcal{F}, \gamma/2, \sqrt{C_\delta/T}\right) \\ &= O\left(\varepsilon \cdot \bar{\theta} \cdot \log\left(\frac{\mathrm{Pdim}(\mathcal{F})}{\varepsilon\,\gamma\,\delta}\right)\right), \end{aligned}$$

where we take $\bar{\theta} := \sup_{\iota > 0} \theta_{\bar{f}}^{\mathrm{val}}(\mathcal{F}, \gamma/2, \iota)$ as an upper bound of $\theta_{\bar{f}}^{\mathrm{val}}(\mathcal{F}, \gamma/2, \sqrt{C_\delta/T})$, and use the fact that $T = \frac{\mathrm{Pdim}(\mathcal{F})}{\varepsilon\,\gamma}$ and the assumptions that $\kappa \leq \varepsilon < \gamma$.

We now analyze the label complexity (note that the sampling process of Algorithm 3 stops at time $t = \tau_{M-1}$). Note that $\mathbb{E}[\mathbb{1}(Q_t = 1) \mid \mathfrak{F}_{t-1}] = \mathbb{E}_{x \sim \mathcal{D}_{\mathcal{X}}}[\mathbb{1}(g_m(x) = 1)]$ for any epoch $m \geq 2$ and

time step $t$ within epoch $m$. Combine Lemma 2 with Lemma 18 leads to

$$\sum_{t=1}^{\tau_{M-1}} \mathbb{1}(Q_t = 1) \leq \frac{3}{2} \sum_{t=1}^{\tau_{M-1}} \mathbb{E}[\mathbb{1}(Q_t = 1) \mid \mathfrak{F}_{t-1}] + 4 \log \delta^{-1}$$

$$\leq 3 + \frac{3}{2} \sum_{m=2}^{M-1} \frac{(\tau_m - \tau_{m-1}) \cdot 36 \beta_m}{\tau_{m-1} \gamma^2} \cdot \theta_{\bar{f}}^{\mathrm{val}}\Big(\mathcal{F}, \gamma/2, \sqrt{\beta_m/\tau_{m-1}}\Big) + 4 \log \delta^{-1}$$

$$\leq 3 + 48 \sum_{m=2}^{M-1} \frac{\beta_m}{\gamma^2} \cdot \theta_{\bar{f}}^{\mathrm{val}}\Big(\mathcal{F}, \gamma/2, \sqrt{\beta_m/\tau_{m-1}}\Big) + 4 \log \delta^{-1}$$

$$\leq 3 + 4 \log \delta^{-1} + O\Big(\frac{M^2 \cdot \varepsilon^2 \cdot T}{\gamma^2} + \frac{M^2 \cdot C_\delta}{\gamma^2}\Big) \cdot \theta_{\bar{f}}^{\mathrm{val}}\Big(\mathcal{F}, \gamma/2, \sqrt{C_\delta/T}\Big)$$

$$= O\Big(\frac{\bar{\theta}\,\mathrm{Pdim}(\mathcal{F})}{\gamma^2} \cdot \Big(\log\Big(\frac{\mathrm{Pdim}(\mathcal{F})}{\varepsilon\,\gamma}\Big)\Big)^2 \cdot \log\Big(\frac{\mathrm{Pdim}(\mathcal{F})}{\varepsilon\,\gamma\,\delta}\Big)\Big)$$

with probability at least $1 - 2\delta$ (due to an additional application of Lemma 2); where we use the fact that $T = \frac{\mathrm{Pdim}(\mathcal{F})}{\varepsilon\,\gamma}$ and the assumptions that $\kappa \leq \varepsilon < \gamma$ as before. $\qquad \square$

**Theorem 12.** *Algorithm 3 can be efficiently implemented via the regression oracle and enjoys the same theoretical guarantees stated in Theorem 9. The number of oracle calls needed is $\widetilde{O}(\frac{\mathrm{Pdim}(\mathcal{F})}{\varepsilon\,\gamma^3})$ for a general set of regression functions $\mathcal{F}$, and $\widetilde{O}(\frac{\mathrm{Pdim}(\mathcal{F})}{\varepsilon\,\gamma})$ when $\mathcal{F}$ is convex and closed under pointwise convergence. The per-example inference time of the learned $\widehat{h}_M$ is $\widetilde{O}(\frac{1}{\gamma^2} \log^2(\frac{\mathrm{Pdim}(\mathcal{F})}{\varepsilon}))$ for general $\mathcal{F}$, and $\widetilde{O}(\log \frac{1}{\gamma})$ when $\mathcal{F}$ is convex and closed under pointwise convergence.*

*Proof.* Note that classifier $\widehat{h}_m$ and query function $q_m$ in Algorithm 3 are constructed in the way as the ones in Algorithm 1, Thus, Algorithm 3 can be efficiently implemented in the same way as discussed in Theorem 5, and enjoys the same per-round computational complexities. The total computational complexity is then achieved by multiplying the per-round computational complexity by $T = \frac{\mathrm{Pdim}(\mathcal{F})}{\varepsilon\,\gamma}$. $\qquad \square$

## H.2 Discussion on $\kappa \leq \varepsilon$

We provide guarantees (in Theorem 9) when $\kappa \leq \varepsilon$ since the learned classifier suffers from an additive $\kappa$ term in the excess error, as shown in the proof of Theorem 9. We next give preliminary discussions on this issue by relating active learning with to a (specific) regret minimization problem and connecting to existing lower bound in the literature. More specifically, we consider the perspective and notations discussed in Appendix G.1. Fix any epoch $m \geq 2$ and time step $t$ within epoch $m$. We have

$$\mathbf{Regret}_t = \mathbb{E}[\ell_t(a_t) - \ell_t(a_t^\star) \mid \mathfrak{F}_{t-1}] = \mathrm{err}_\gamma(\widehat{h}_m) - \mathrm{err}(h^\star) = \mathsf{excess}_\gamma(\widehat{h}_m) = \widetilde{O}\Big(\kappa + \frac{\bar{\theta}}{2^m\,\gamma}\Big),$$

where the bound comes from similar analysis as in the proof of Theorem 9. Summing the instantaneous regret over $T$ rounds, we have

$$\mathbf{Regret}(T) = \sum_{t=1}^{T} \mathbf{Regret}_t$$

$$\leq 2 + \sum_{m=2}^{M} (\tau_m - \tau_{m-1}) \cdot \mathsf{excess}_\gamma(\widehat{h}_m)$$

$$\leq \widetilde{O}\Big(\kappa \cdot T + \frac{\bar{\theta}}{\gamma}\Big).$$

The above bound indicates an additive regret term scales as $\kappa \cdot T$. On the other hand, it is known that an additive $\kappa \cdot T$ regret is in general unavoidable in linear bandits under model misspecification

(Lattimore et al., 2020). This connection partially explains/justifies why we only provide guarantee for Theorem 9 under $\kappa \leq \varepsilon$.

There are, however, many differences between the two learning problems. We list some distinctions below.

1. The regret minimization problem considered in Appendix G.1 only takes three actions $\mathcal{A} = \{+1, -1, \bot\}$, yet the lower bound in linear bandits is established with a large action set (Lattimore et al., 2020);

2. A standard contextual bandit problem will observe loss (with respect to the pulled action) at each step $t \in [T]$, however, the active learning problem will only observe (full) feedback at time steps when a query is issued, i.e., $\{t \in [T] : Q_t = 1\}$.

We leave a comprehensive study of the problem for feature work.

### H.3 Supporting lemmas

We use the same notations defined in Appendix E, except $\widehat{h}_m$, $g_m$ and $\beta_m$ are defined differently. We adapt the proofs Theorem 4 (in Appendix E) to deal with model misspecification.

Note that although we do not have $f^\star \in \mathcal{F}$ anymore, one can still define random variables of the form $M_t(f)$, and guarantees in Lemma 4 still hold. We use $\mathcal{E}$ to denote the good event considered in Lemma 4, we analyze under this event through out the rest of this section. We also only analyze under the assumption of Theorem 9, i.e., $\kappa^2 \leq \varepsilon$.

**Lemma 16.** *Fix any epoch $m \in [M]$. We have*

$$\widehat{R}_m(\bar{f}) \leq \widehat{R}_m(f^\star) + \frac{3}{2} \cdot \kappa^2 \tau_{m-1} + C_\delta,$$

*where $C_\delta := 8 \log\left(\frac{|\mathcal{F}| \cdot T^2}{\delta}\right)$.*

*Proof.* From Lemma 4 we know that

$$\widehat{R}_m(\bar{f}) - \widehat{R}_m(f^\star) \leq \sum_{t=1}^{\tau_{m-1}} \frac{3}{2} \cdot \mathbb{E}_t\left[Q_t\big(\bar{f}(x_t) - f^\star(x_t)\big)^2\right] + C_\delta$$

$$\leq \frac{3}{2} \cdot \kappa^2 \tau_{m-1} + C_\delta,$$

where we use the fact that $\mathbb{E}_t[y_t \mid x_t] = f^\star(x_t)$ (and thus $\mathbb{E}_t[M_t(\bar{f})] = \mathbb{E}_t[Q_t(\bar{f}(x_t) - f^\star(x_t))^2]$) on the first line; and use the fact $\sup_x |\bar{f}(x) - f^\star(x)| \leq \kappa$ on the second line. $\square$

**Lemma 17.** *The followings hold true:*

1. *$\bar{f} \in \mathcal{F}_m$ for any $m \in [M]$.*

2. *$\sum_{t=1}^{\tau_{m-1}} \mathbb{E}_t[M_t(f)] \leq 4\beta_m$ for any $f \in \mathcal{F}_m$.*

3. *$\sum_{t=1}^{\tau_{m-1}} \mathbb{E}[Q_t(x_t)(f(x_t) - \bar{f}(x_t))^2] \leq 9\beta_m$ for any $f \in \mathcal{F}_m$.*

4. *$\mathcal{F}_{m+1} \subseteq \mathcal{F}_m$ for any $m \in [M-1]$.*

*Proof.* 1. Fix any epoch $m \in [M]$. By Lemma 4, we have $\widehat{R}_m(f^\star) \leq \widehat{R}_m(f) + C_\delta/2$ for any $f \in \mathcal{F}$. Combining this with Lemma 16 leads to

$$\widehat{R}_m(\bar{f}) \leq \widehat{R}_m(f) + \frac{3}{2} \cdot \big(\kappa^2 \tau_{m-1} + C_\delta\big)$$

$$\leq \widehat{R}_m(f) + \beta_m,$$

for any $f \in \mathcal{F}$, where the second line comes from the definition of $\beta_m$ (recall that we have $\kappa \leq \varepsilon$ by assumption). We thus have $\bar{f} \in \mathcal{F}_m$ for any $m \in [M]$.

2. Fix any $f \in \mathcal{F}_m$. With Lemma 4, we have

$$
\begin{aligned}
\sum_{t=1}^{\tau_{m-1}} \mathbb{E}_t[M_t(f)] &\leq 2 \sum_{t=1}^{\tau_{m-1}} M_t(f) + C_\delta \\
&= 2\widehat{R}_m(f) - 2\widehat{R}_m(f^\star) + C_\delta \\
&\leq 2\widehat{R}_m(f) - 2\widehat{R}_m(\bar{f}) + 3\kappa^2 \tau_{m-1} + 3C_\delta \\
&\leq 2\widehat{R}_m(f) - 2\widehat{R}_m(\widehat{f}_m) + 3\kappa^2 \tau_{m-1} + 3C_\delta \\
&\leq 2\beta_m + 3\kappa^2 \tau_{m-1} + 3C_\delta \\
&\leq 4\beta_m,
\end{aligned}
$$

where the third line comes from Lemma 16; the fourth line comes from the fact that $\widehat{f}_m$ is the minimizer of $\widehat{R}_m(\cdot)$; and the fifth line comes from the fact that $f \in \mathcal{F}_m$.

3. Fix any $f \in \mathcal{F}_m$. With Lemma 4, we have

$$
\begin{aligned}
\sum_{t=1}^{\tau_{m-1}} \mathbb{E}_t[Q_t(x_t)(f(x_t) - \bar{f}(x_t))^2] &= \sum_{t=1}^{\tau_{m-1}} \mathbb{E}_t[Q_t(x_t)((f(x_t) - f^\star(x_t)) + (f^\star(x_t) - \bar{f}(x_t)))^2] \\
&\leq 2 \sum_{t=1}^{\tau_{m-1}} \mathbb{E}_t[Q_t(x_t)(f(x_t) - f^\star(x_t))^2] + 2\tau_{m-1}\kappa^2 \\
&= 2 \sum_{t=1}^{\tau_{m-1}} \mathbb{E}_t[M_t(f)] + 2\tau_{m-1}\kappa^2 \\
&\leq 8\beta_m + 2\tau_{m-1}\kappa^2 \\
&\leq 9\beta_m,
\end{aligned}
$$

where we use $(a + b)^2 \leq a^2 + b^2$ on the second line; and use statement 2 on the fourth line.

4. Fix any $f \in \mathcal{F}_{m+1}$. We have

$$
\begin{aligned}
\widehat{R}_m(f) - \widehat{R}_m(\widehat{f}_m) &\leq \widehat{R}_m(f) - \widehat{R}_m(f^\star) + \frac{C_\delta}{2} \\
&= \widehat{R}_{m+1}(f) - \widehat{R}_{m+1}(f^\star) - \sum_{t=\tau_{m-1}+1}^{\tau_m} M_t(f) + \frac{C_\delta}{2} \\
&\leq \widehat{R}_{m+1}(f) - \widehat{R}_{m+1}(\bar{f}) + \frac{3}{2}\kappa^2 \tau_m + C_\delta - \sum_{t=\tau_{m-1}+1}^{\tau_m} \mathbb{E}_t[M_t(f)]/2 + C_\delta \\
&\leq \widehat{R}_{m+1}(f) - \widehat{R}_{m+1}(\widehat{f}_{m+1}) + \frac{3}{2}\kappa^2 \tau_m + 2C_\delta \\
&\leq \beta_{m+1} + \frac{3}{2}\kappa^2 \tau_m + 2C_\delta \\
&\leq \beta_m,
\end{aligned}
$$

where the first line comes from Lemma 4; the third line comes from Lemma 16 and Lemma 4; the fourth line comes from the fact that $\widehat{f}_{m+1}$ is the minimizer with respect to $\widehat{R}_{m+1}$ and Lemma 4; the last line comes from the definition of $\beta_m$.

$\square$

Since the classifier $\widehat{h}_m$ and query function $g_m$ are defined in the same way as in Algorithm 1, Lemma 6 holds true for Algorithm 3 as well. As a result of that, Lemma 7 and Lemma 8 hold true with minor modifications. We present the modified versions below, whose proofs follow similar steps as in Lemma 7 and Lemma 8 but replace $f^\star$ with $\widehat{f}$ (and thus using concentration results derived in Lemma 17).

**Lemma 18.** *Fix any epoch $m \geq 2$. We have*

$$\mathbb{E}_{x \sim \mathcal{D}_{\mathcal{X}}}[\mathbb{1}(g_m(x) = 1)] \leq \frac{36\beta_m}{\tau_{m-1}\,\gamma^2} \cdot \theta_{\bar{f}}^{\mathrm{val}}\Big(\mathcal{F}, \gamma/2, \sqrt{\beta_m/\tau_{m-1}}\Big).$$

**Lemma 19.** *Fix any epoch $m \geq 2$. We have*

$$\mathbb{E}_{x \sim \mathcal{D}_{\mathcal{X}}}[\mathbb{1}(g_m(x) = 1) \cdot w(x; \mathcal{F}_m)] \leq \frac{36\beta_m}{\tau_{m-1}\gamma} \cdot \theta_{\bar{f}}^{\mathrm{val}}\Big(\mathcal{F}, \gamma/2, \sqrt{\beta_m/\tau_{m-1}}\Big).$$

**Lemma 20.** *Fix any $m \in [M]$. We have $\mathsf{excess}_\gamma(\widehat{h}_m; x) \leq 2\kappa$ if $g_m(x) = 0$.*

*Proof.* Recall that

$$\mathsf{excess}_\gamma(\widehat{h}; x) = \mathbb{1}\big(\widehat{h}(x) \neq \bot\big) \cdot \big(\mathbb{P}_{y|x}\big(y \neq \mathrm{sign}(\widehat{h}(x))\big) - \mathbb{P}_{y|x}\big(y \neq \mathrm{sign}(h^\star(x))\big)\big)$$
$$+ \mathbb{1}\big(\widehat{h}(x) = \bot\big) \cdot \big((1/2 - \gamma) - \mathbb{P}_{y|x}\big(y \neq \mathrm{sign}(h^\star(x))\big)\big).$$

We now analyze the event $\{g_m(x) = 0\}$ in two cases.

**Case 1: $\widehat{h}_m(x) = \bot$.**

Since $\bar{f}(x) \in [\mathsf{lcb}(x; \mathcal{F}_m), \mathsf{ucb}(x; \mathcal{F}_m)]$ by [Lemma 17](), we know that $\eta(x) = f^\star(x) \in [\frac{1}{2} - \gamma - \kappa, \frac{1}{2} + \gamma + \kappa]$ and thus $\mathbb{P}_y\big(y \neq \mathrm{sign}(h^\star(x))\big) \geq \frac{1}{2} - \gamma - \kappa$. As a result, we have $\mathsf{excess}_\gamma(\widehat{h}_m; x) \leq \kappa$.

**Case 2: $\widehat{h}_m(x) \neq \bot$ but $\frac{1}{2} \notin (\mathsf{lcb}(x; \mathcal{F}_m), \mathsf{ucb}(x; \mathcal{F}_m))$.**

We clearly have $\mathsf{excess}_\gamma(\widehat{h}_m; x) \leq 0$ if $\mathrm{sign}(\widehat{h}_m(x)) = \mathrm{sign}(h^\star(x))$. Now consider the case when $\mathrm{sign}(\widehat{h}_m(x)) \neq \mathrm{sign}(h^\star(x))$. Since $\bar{f}(x) \in [\mathsf{lcb}(x; \mathcal{F}_m), \mathsf{ucb}(x; \mathcal{F}_m)]$ and $|\bar{f}(x) - f^\star(x)| \leq \kappa$, we must have $|f^\star(x) - 1/2| \leq \kappa$ in that case, which leads to $\mathsf{excess}_\gamma(\widehat{h}_m; x) \leq 2|f^\star(x) - 1/2| \leq 2\kappa$. $\qquad\square$