# OpenReview forum: "Efficient Active Learning with Abstention"
_NeurIPS.cc/2022/Conference — NeurIPS 2022 Accept_

### Official Review · Reviewer_5zzs · 2022-07-05

**Rating:** 6
**Confidence:** 3
**Soundness:** 2 fair
**Presentation:** 3 good
**Contribution:** 3 good

**Summary:**

The paper studies active learning of general concept classes. Lower bound is known in this regime to rule out savings in label complexity over passive learning. However, [PT21] showed that with the additional action of abstention, active learning does provide exponential savings in terms of the error rate. This work follows the research line, and the main contribution falls into a computationally efficient algorithm that archives label complexity comparable to [PT21]. The main algorithm relies on efficient implementation of regression oracles, which has been developed in prior works.


**Questions:**


It is true that [PT21] runs with minimizing an empirical 0/1 loss which is NP-hard. Can you give more intuition on why the 0/1 loss is vital for their analysis, and why the regression oracle approach in the paper works as well?

**Limitations:**

  Yes.

**Strengths And Weaknesses:**

Strengths:
+ Active learning is a very useful tool to reduce labeling cost, and this paper studies an interesting and practical extension.
+ The core contribution on efficient learning paradigm is important.
+ The paper is well written and easy to follow, with right amount of reminders and pointers.

Weakness:
- The computational efficiency is phased in terms of number of calls to an oracle, yet leaving the runtime of that oracle unsettled. Please provide concrete computational cost analysis to justify the main contribution.
- It is true that [PT21] runs with minimizing an empirical 0/1 loss which is NP-hard. Can you give more intuition on why the 0/1 loss is vital for their analysis, and why the regression oracle approach in the paper works as well?

---

> ### Author Response · Authors · 2022-08-02
> **Response to Reviewer 5zzs**
>
> Thank you for your positive review. We hope our responses below can help resolve your concerns.
>
>
> 1. > The computational efficiency is phased in terms of number of calls to an oracle, yet leaving the runtime of that oracle unsettled. Please provide concrete computational cost analysis to justify the main contribution.
>
> **Response:** The implementation of the regression oracle should be viewed as an efficient operation since the regression oracle solves a convex optimization problem with respect to $f$, which even admits closed-form solutions for certain function classes. A concrete example is when $f$ is a linear function in $\mathbb{R}^{d}$: With a set of $N$ data points, we can implement the regression oracle with runtime $O((N+d)\cdot d^{2})$ using the standard least squares method.
>
> 2. > It is true that [PT21] runs with minimizing an empirical 0/1 loss which is NP-hard. Can you give more intuition on why the 0/1 loss is vital for their analysis, and why the regression oracle approach in the paper works as well?
>
> **Response:** The analysis of Puchkin and Zhivotovskiy (2021) follows the standard active learning analysis (as their authors agree, see Appendix B of their paper), which relies on connecting the empirical error on the labeled dataset and the whole dataset (e.g., Step 0 in their Appendix B). Such connection is established using the 0/1 loss. Besides, 0/1 loss is an unbiased estimator of the true error, which can be easily analyzed using standard concentration results.
>
> To get computational efficiency, we need to consider regression over the square loss (since ERM on 0/1 loss is known to be NP-hard). Our analysis (e.g., proof of Theorem 4 on page 18 of the supplementary material) shows that the classifier induced from a regression function with small square loss enjoys low error; such guarantee is developed with concentration results regarding the square loss (Lemmas 1-4 in Appendix D; pages 16-17 of supplementary material) and properties of the uneliminated set of regression functions (Lemmas 5-9 in Appendix E.1; pages 20-22 of supplementary material).

---

### Official Review · Reviewer_y5fo · 2022-07-09

**Rating:** 7
**Confidence:** 3
**Soundness:** 3 good
**Presentation:** 3 good
**Contribution:** 3 good

**Summary:**

This paper studies the pool-based active learning problem. The main contribution is to propose a computationally efficient algorithm to train a rejection model. Under the realizable case, the model enjoys $\epsilon$ chow's excess risk with $\widetilde{O}(\mathrm{polylog}(1/\epsilon))$ label complexity. The guarantee is achieved without any low noise assumption commonly used to achieve the exponential savings label complexity in literature. Although a similar rate (for learning with abstentions) has already appeared in the literature, the proposed method is more efficient (or practical) than the previous one. Besides the main result, the authors also show that (a slight modification of) the proposed method enjoys minimax optimal label complexity for the standard excess risk with the low noise assumption. Furthermore, this paper has shown a constant label complexity in a special case (with a finite hypothesis set) and presented the guarantees with model misspecification.

**Questions:**

Q1: can the proposed method achieves similar Chow's excess risk without the realizable assumption when comparing with the best model in the hypothesis space $\mathcal{F}$.

Q2: how to compute the parameter $\theta$ efficiently (please refer to the second point of the weakness for more details)

Q3: what is the computational cost of ucb and lcb for a hypothesis space $\mathcal{F}$ containing $f_\star$

**Limitations:**

This paper has discussed its limitation on the realizable assumption in Section 4.2. It has shown that the same exponential saving label complexity is achieved with misspecified model space as long as $\epsilon$ is less than the approximation error $\kappa$. The results partially address the limitation on the assumption, but I think the paper would become even strong if the author could show a similar convergence rate for Chow's excess risk compared with the best model in the hypothesis when $f_*$ is not in $\mathcal{F}.

**Strengths And Weaknesses:**

### Strength:
Overall, I think this is a nice paper with fruitful results. Specifically, the strengths of this paper are listed as follows,
+ novelty& significance: although the algorithm framework shares a similar spirit as the previous work [Krishnamurthy et al. 2017] in standard active learning with abstention, the new criterion for label querying is interesting to me. A similar rate for active learning with abstention has been achieved by [Puchkin and Zhivotovskiy, 2021], but a computationally efficient algorithm is always what we desire.

+ clarity: this paper is well written and clearly structured for the most part. Although there are fruitful results regarding Chow's excess risk and the standard excess risk (under different conditions), the authors have clearly organized them to make the results easy to follow.

### Weakness:
In general, I like the results of the paper, but I have still some reservations about the assumption and the computational cost as follows,
- about the realizable assumption: although the realizable has frequently appeared in the active learning literature,  the most related work on active learning with abstention seems not to require such an assumption (Theorem 1.1 of [Puchkin and Zhivotovskiy, 2021]). It seems to me that the realizable assumption is the price for the efficient algorithm since Algorithm 1 requires to approximate $\eta(y=+1|\mathbf{x})$ with the function $f(\mathbf{x})$ (by ucb and lcb). So, I think it would be necessary to make a more clear comparison with the previous work.

- about the efficient algorithm:
	- issue on the parameter setting: the algorithm takes the disagreement coefficient $\theta$ as the input. I am not sure whether such a coefficient can be calculated efficiently in general? (maybe an upper bound for $\theta$ is enough in special cases, but the realizable assumption could be violated.)
	- issue on the estimation of lcb and ucb: although the authors have referred to [Krishnamurthy et al. 2017] for the calculation of the lcb and ucb, I think it would be nice to discuss their computational costs since the efficiency is one of the main contributions of this paper. (for example how hard it is to compute lcb or ucb for a $\mathcal{F}$ containing the $f_\star$?).

---

> ### Author Response · Authors · 2022-08-02
> **Response to Reviewer y5fo**
>
> Thank you for your thorough review and positive comments. Please see our responses to specific comments below.
>
> ### **Responses to Weaknesses:**
>
> 1. > about the realizable assumption: although the realizable has frequently appeared in the active learning literature, the most related work on active learning with abstention seems not to require such an assumption (Theorem 1.1 of [Puchkin and Zhivotovskiy, 2021]). It seems to me that the realizable assumption is the price for the efficient algorithm since Algorithm 1 requires to approximate $\eta(y = +1 \vert x)$ with the function $f(x)$ (by ucb and lcb). So, I think it would be necessary to make a more clear comparison with the previous work.
>
> **Response:** Yes, the work of Puchkin and Zhivotovskiy (2021) doesn't require realizability (but instead requires the NP-hard ERM oracle). We tend to believe that (approximate) realizability is the price for computational efficiency. We are happy to add clarifications in the revision.
>
> 2. > issue on the parameter setting: the algorithm takes the disagreement coefficient $\theta$ as the input. I am not sure whether such a coefficient can be calculated efficiently in general? (maybe an upper bound for $\theta$ is enough in special cases, but the realizable assumption could be violated.)
>
> **Response:** This is a great question. We answer your question from the following two aspects.
>
> (1) Our algorithm only requires an upper bound on the disagreement coefficient $\theta$. We know that $\theta \leq d$ for linear functions and  $\theta \leq C_{\textsf{link}}\cdot d$ for generalized linear functions ( $C_{\textsf{link}}$ is a constant depending on the link function). More generally, we can upper bound $\theta$ by the eluder dimension or the (squared) star number, which are well-known complexity measures that have been previously studied in the literature (see page 15 of supplementary material for detailed discussion). That being said, once the eluder dimension/star number of a function class has been theoretically analyzed (there is no need to compute it empirically), we can use it as an upper bound for $\theta$.
>
> (2) We don't necessarily need to take $\theta$ as an input to the algorithm. Instead, we can simply run a modified version of Algorithm 1 with $T = \widetilde O (\frac{\textsf{Pdim}({\cal F})}{\epsilon \gamma})$ and achieve excess error $O(\epsilon \cdot \theta)$. In this case,  $\theta$ is only a theoretical quantity that our algorithm is agnostic of.
>
> 3. > issue on the estimation of lcb and ucb: although the authors have referred to [Krishnamurthy et al. 2017] for the calculation of the lcb and ucb, I think it would be nice to discuss their computational costs since the efficiency is one of the main contributions of this paper. (for example how hard it is to compute lcb or ucb for a ${\cal F}$ containing the $f^{\star}$?).
>
> **Response:** The computational cost is in terms of the number of calls to the regression oracle (defined above line 84), which solves a convex optimization problem wrt $f$ (and even admits a closed-form solution when $f$ is linear). Previous results (Proposition 8 on page 19 of supplementary material) show that one can achieve $\alpha$ approximation error with  $O(\frac{1}{\alpha^{2}} \log \frac{1}{\alpha})$ (or $O(\log \frac{1}{\alpha})$ when ${\cal F}$ is convex) calls to the regression oracle. Our analysis (Theorem 5, with proofs on page 19 of supplementary material) shows that it suffices to approximate lcb/ucb with approximation error $\alpha = O(\frac{\gamma}{\log T})$ to achieve the same theoretical guarantees shown in Theorem 4 up to changes in constant terms. This leads to the computational guarantees stated in Theorem 5.
>
> ---
>
> ### **Responses to Questions:**
>
> 1. > can the proposed method achieves similar Chow's excess risk without the realizable assumption when comparing with the best model in the hypothesis space ${\cal F}$.
>
> **Response:** We currently managed to provide guarantees when the approximation error $\kappa$ is small (see Theorem 9 in Section 4.2). We provide preliminary evidence (see Section 4.2 and Appendix H.2) on why such a requirement is needed in our analysis. We believe a comprehensive understanding of the problem without realizability is an important future direction.
>
> 2. > how to compute the parameter $\theta$ efficiently (please refer to the second point of the weakness for more details)
>
> **Response:** Please see our response to point 2 in the **Responses to Weaknesses** section above.
>
> 3. > what is the computational cost of ucb and lcb for a hypothesis space ${\cal F}$ containing $f^{\star}$
>
> **Response:** Please see our response to point 3 in the **Responses to Weaknesses** section above.

---

> > ### Comment · Reviewer_y5fo · 2022-08-08
> > **Response to the authors**
> >
> > Thank you for the detailed response. The rebuttal has addressed most parts of my concerns. But I am still a little bit confused by the boundedness of $\theta$.
> >
> > It seems that the main theorems of this paper rely on the crucial realizability assumption, where the $\mathcal{F}$ should contain a regressor characterizing the true conditional probability. But it is unclear to me whether the listed examples (such as linear function and generalized linear function) meet the condition in general. Does there exists a function class containing all possible true conditional probability but has bounded $\theta$. I believe a clearer discussion on this issue would make this paper more competitive.

---

> > > ### Author Response · Authors · 2022-08-09
> > > **Response to follow-up questions**
> > >
> > > Thank you for your reply.
> > >
> > > We believe it's hard to find *a single function class* that contains all possible true conditional probability (under *arbitrary* ${\cal D_{XY}}$) and has a non-trivial disagreement coefficient (we can trivially bound $\theta \leq \gamma/\epsilon$ for any ${\cal F}$; see Definition 1 and footnote 3).
> > > Our guarantees hold under realizability and bounded disagreement coefficient; we discuss example function classes in the paper (which strictly generalizes previous results focusing on linear models).
> > > We see two directions to further strengthen our results:
> > >
> > >  - Relax the realizability assumption. We develop preliminary results when the assumption is only approximately achieved (see Section 4.2). We believe a comprehensive understanding of the problem without realizability is an important future direction.
> > >
> > >  - Bound disagreement coefficient (or eluder dimension/star number) for richer function classes. Since our algorithms are designed for general function classes, any future developments on these complexity measures directly lead to broader applications of our algorithms.
> > >
> > > We are happy to add more discussions on this issue in the revision.

---

### Official Review · Reviewer_9njm · 2022-07-11

**Rating:** 8
**Confidence:** 3
**Soundness:** 4 excellent
**Presentation:** 3 good
**Contribution:** 4 excellent

**Summary:**

The paper proposes an active learning algorithm that can avoid sampling from regions of the input space with high label noise. The algorithm satisfies two important properties: 1) it achieves exponential improvements compared to passive learning with respect to an evaluation metric that penalizes abstentions (Chow’s excess error); and 2) the algorithm is computationally tractable for finite pseudo dimension function classes.


**Questions:**

- How does the analysis of Algorithm 1 change if we assume a finite unlabeled set? Can the bound of Theorem 4 be changed to factor in the size of the unlabeled set?

- Steps 5-6 of Algorithm 1 approximate the lcb/ucb over the set of functions $F_m$. For what function classes $F$ is this step tractable? How can the approximation error of the lcb/ucb influence the result of Theorem 4? For what function classes is the approximation of the lcb/ucb “reasonable”?

- Algorithm 1 enjoys strong guarantees at arbitrary noise levels. This is captured in the bounds via $\gamma$ which needs to be chosen as a function of the noise level (indeed this is explicit in the proof of Theorem 6) and $\theta$ (which captures the disagreement compared to $f^\star$ at a fixed level $\gamma$). How do $\theta$ and the bound on sample complexity change when $\gamma$ is chosen inappropriately in Theorem 6, without knowledge of the noise level?

- Can the noise-seeking noise condition be relaxed for Proposition 3? The negative result for uncertainty-based AL is similar in spirit to the one in “On the relationship between data efficiency and error for uncertainty sampling” Mussman & Liang, 2018. While the analysis here is significantly different, the result in Mussman et al only requires non-vanishing Bayes error to reveal the failure of uncertainty sampling.

- Minor remark: “Noise-seeking Massart/Tsybakov noise” can be a bit confusing. Perhaps something like “Random flip-allowing Massart noise” could make it easier to grasp (although it’s a bit of a mouthful)?


**Limitations:**

The paper generally addresses some of the poignant limitations of the analysis (e.g. focus on finite pseudo dimensions function classes, realizable vs agnostic case etc). See the Questions section for other limitations that could also be discussed in the paper.


**Strengths And Weaknesses:**

Strengths:

- The paper employs in a creative way techniques from contextual bandit literature to extend the idea of Puchkin et al and propose a computationally tractable algorithm. Moreover, the result is particularly remarkable since it does not require a condition constraining the amount of label noise, but rather captures it in the bound.

- The analysis for deriving the results is non-trivial and some of the connections to quantities from the contextual bandit literature (e.g. eluder dimension, disagreement coefficient) may be of independent interest for the active learning community.

- The paper contains several results that help to position the proposed algorithm in the broader active learning literature. For instance, the analysis of Section 3 confirms that the algorithm is minimax optimal (albeit not substantially better than passive learning) with respect to the standard excess risk.

Weaknesses:

- While the paper is generally easy to follow, certain details regarding the algorithm or the analysis could be discussed in more detail in the main text (see the Questions section)

- Minor remarks: There are a few typos in the paper (e.g. lines 259, 266, 274 etc). Also the pseudocode of Algorithm 1 can be made a bit more precise and easier to follow (perhaps add a notation for the labeled set; Q_t, x_t, y_t are not defined when they appears first in step 4; unspecified how $\hat{f}_1$ is selected etc).

---

> ### Author Response · Authors · 2022-08-02
> **Response to Reviewer 9njm (1/2)**
>
> Thank you for your thorough review and positive comments. Please see our responses to specific comments below.
>
> ### **Responses to Weaknesses:**
>
> 1. > While the paper is generally easy to follow, certain details regarding the algorithm or the analysis could be discussed in more detail in the main text (see the Questions section)
>
> **Response:** Thank you for your suggestion. We are quite space-limited in the submission, but we are happy to add more details to the main content in the revision.
>
> 2. > Minor remarks: There are a few typos in the paper (e.g. lines 259, 266, 274 etc). Also the pseudocode of Algorithm 1 can be made a bit more precise and easier to follow (perhaps add a notation for the labeled set; $Q_t, x_t, y_t$ are not defined when they appears first in step 4; unspecified how $\widehat f_1$ is selected etc).
>
> **Response:** Thank you for catching these typos; we'll correct them in the revision. We are also happy to adjust the pseudocode of Algorithm 1. Regarding your concerns: $Q_t, x_t, y_t$ are defined at lines 10 - 12; and  $\widehat f_1$ can be selected arbitrarily since the first epoch is of length  $2$ (and thus, the total number of labels queried in the first epoch is never larger than $2$).
>
> ---
>
> ### **Responses to Questions:**
>
> 1. > How does the analysis of Algorithm 1 change if we assume a finite unlabeled set? Can the bound of Theorem 4 be changed to factor in the size of the unlabeled set?
>
> **Response:** Our analysis works as long as one can randomly sample $(x,y)$ from the underlying distribution ${\cal D_{XY}} = {\cal D_{X}} \times {\cal D_{{Y} \vert {X}}}$ ($y$ is observed only after label query). When ${\cal X}$ is finite, we can take ${\cal D_{\cal X}}$ as the uniform distribution over ${\cal X}$, and ${\cal D_{{\cal Y} \vert {\cal X}}}$ as the labeling distribution. If ${\cal D_{{\cal Y} \vert {\cal X}}}$ is stochastic (with fresh randomness each time), our analyses/guarantees are the same as before. If ${\cal D_{{\cal Y} \vert {\cal X}}}$ is deterministic, the sample complexity is trivially upper bounded by the cardinality of ${\cal X}$: There is no need to query the label of a previously queried data point. Obtaining better dependence on the cardinality of ${\cal X}$ may require additional structural assumptions.
>
> 2. > Steps 5-6 of Algorithm 1 approximate the lcb/ucb over the set of functions ${\cal F_{\text{m}}}$. For what function classes ${\cal F}$ is this step tractable? How can the approximation error of the lcb/ucb influence the result of Theorem 4? For what function classes is the approximation of the lcb/ucb “reasonable”?
>
> **Response:** We can efficiently approximate lcb/ucb as long as one has access to a regression oracle that solves the (weighted) square loss optimization problem (defined above line 84) with respect to the function class ${\cal F}$. The optimization problem is reasonably easy to solve since it is convex with respect to the regression function $f \in {\cal F}$. It even admits closed-form solutions in many cases (e.g., it is reduced to standard least square when $f$ is linear).
>
> We carefully dealt with the approximation error so that the guarantees in Theorem 4 still hold (up to changes in constant terms). Previous results (Proposition 8 on page 19 of supplementary material) show that one can achieve $\alpha$ approximation error with  $O(\frac{1}{\alpha^{2}} \log \frac{1}{\alpha})$ (or $O(\log \frac{1}{\alpha})$ when ${\cal F}$ is convex) calls to the regression oracle. Our analysis (Theorem 5, with proofs shown on page 19 of supplementary material) shows that it suffices to approximate lcb/ucb with approximation error $\alpha = O(\frac{\gamma}{\log T})$ to achieve the same theoretical guarantees shown in Theorem 4 up to changes only in constant terms.
>
> **Continued in the next response.**

---

> > ### Author Response · Authors · 2022-08-02
> > **Response to Reviewer 9njm (2/2)**
> >
> > ### **Responses to Questions (Cont'd)**
> >
> > 3. > Algorithm 1 enjoys strong guarantees at arbitrary noise levels. This is captured in the bounds via $\gamma$ which needs to be chosen as a function of the noise level (indeed this is explicit in the proof of Theorem 6) and $\theta$ (which captures the disagreement compared to $f^{\star}$ at a fixed level $\gamma$). How do $\theta$ and the bound on sample complexity change when $\gamma$ is chosen inappropriately in Theorem 6, without knowledge of the noise level?
> >
> > **Response:** We first clarify two points: (1) Under Chow's excess error (defined at lines 87-97), $\gamma$ is a given parameter that is not chosen based on other quantities (it is associated with the definition of Chow's excess error). (2) Under standard excess error, $\gamma$ is chosen based on the noise level but not  $\theta$. Instead, $\theta$ (or an upper bound of $\theta$) is a function of $\gamma$, and its value is not affected even if $\gamma$ is chosen inappropriately regarding the noise level.
> >
> > We now discuss the effect of the unknown noise level. Our current analysis of Theorem 6 requires the knowledge of noise level (also previously assumed in active learning literature, e.g., Balcan et al. (2007), Hanneke (2014)). If $\gamma$ is chosen inappropriately (when the noise level is unknown), we can still upper bound the standard excess error using Eq. (4) (below line 247). However, the excess error bound may no longer match the minimax lower bound. Designing minimax optimal algorithms that can automatically adapt to the noise level is left as an interesting future direction.
> >
> > 4. > Can the noise-seeking noise condition be relaxed for Proposition 3? The negative result for uncertainty-based AL is similar in spirit to the one in “On the relationship between data efficiency and error for uncertainty sampling” Mussman \& Liang, 2018. While the analysis here is significantly different, the result in Mussman et al only requires non-vanishing Bayes error to reveal the failure of uncertainty sampling.
> >
> > **Response:** Thank you for pointing out the paper by Mussmann and Liang (2018). We answer this question from the following three aspects.
> >
> > (1) Proposition 3 states, ''there exists a learning problem satisfies Definition 5/Definition 6 (the noise-seeking conditions) such that any 'uncertainty-based' active learner doesn't perform well''. This is essentially a lower bound statement, and the fact of satisfying noise-seeking conditions only makes the lower bound stronger: It automatically ensures that ''there exists a learning problem such that any 'uncertainty-based' active learner doesn't perform well''.
> >
> > (2) To strengthen Proposition 3, one needs to change ''there exists a learning problem'' to ''for any learning problem''. However, we don't believe such a statement is true: There certainly exist easy learning problems where ''uncertainty-based'' active learners perform well.
> >
> > (3) Nevertheless, there is no conflict with the statements proved in Mussmann and Liang (2018): They analyze a special two-stage uncertainty sampling algorithm (see Section 4.1 of their paper), and our ''uncertainty-based'' active learner is formally defined in Definition 10 (page 24 of supplementary material). Also, they study and relative performance of uncertainty sampling with respect to random sampling, yet we consider the absolute performance of ''uncertainty-based'' active learners.
> >
> > 5. > Minor remark: “Noise-seeking Massart/Tsybakov noise” can be a bit confusing. Perhaps something like “Random flip-allowing Massart noise” could make it easier to grasp (although it’s a bit of a mouthful)?
> >
> > **Response:** Thank you for your suggestion. We will take your suggestion into consideration in the revision.
> > We currently use the term ''noise-seeking'' to refer to the noise-seeking behavior (i.e., over-sampling from the high-noise regions) of standard ''uncertainty-based'' algorithms under the proposed noise assumptions.

---

> > > ### Comment · Reviewer_9njm · 2022-08-05
> > > **Response to rebuttal**
> > >
> > > I would like to thank the authors for the detailed responses provided in the rebuttal. While the manuscript has not been updated yet on openreview, I hope that the authors will incorporate the suggestions from reviewers for the camera ready version of the paper. All things considered, I believe this paper would be a good addition to the conference, and hence, I keep my score and recommend acceptance.

---

> > > > ### Author Response · Authors · 2022-08-05
> > > > **Thanks for your reply**
> > > >
> > > > Thank you for your prompt response. We'll definitely incorporate suggestions from reviewers into the revision of the paper.

---

### Official Review · Reviewer_8Wbo · 2022-07-12

**Rating:** 3
**Confidence:** 1
**Soundness:** 2 fair
**Presentation:** 2 fair
**Contribution:** 2 fair

**Summary:**

Paper proposes an active learning algorithm in which labels are not acquired when the model chooses to abstain from predicting.

**Questions:**

N/A

**Strengths And Weaknesses:**

[note confidence score of 1 (@authors + @AC). this is largely outside my comfort zone with many proofs, of which I did not study in details. I would recommend discarding my opinion.]

I think lines 35-38 already starts this paper off on the wrong foot. The objective of active learning is to the learn the decision boundary with minimal labels; what the model does with points close to the decision boundary or with high uncertainty once this is learnt could be considered another problem. If the model can the learn the decision boundary with high accuracy given more queries around the decision boundary then this is justified.

Strengths
- The introduction makes clear what the paper is trying to achieve.

Weaknesses
- The paper is not easy to read.
- Scenario proposed does not reasonable.
- No experimental results whatsoever.
- Lots of propositions and theorems are stated in the paper, but all the proofs in the appendix. I have but skimmed these.

Originality:
Certainly seems new, but a more directed related work would be appreciated.

Quality:
I find the theoretical exposition somewhat lacking in places-- e.g. for proposition 3 and the statement follows of the algorithms superiority over any uncertainty-based AL method. No experimental results.

Clarity:
Neither well written nor well organised. Language is imprecise and unclear. Parts of the paper make strong statements without backing.

Significance:
Difficult to assess the impact of the method. The impact of the paper however will be small, given the problems above.

---

> ### Author Response · Authors · 2022-08-02
> **Response to Reviewer 8Wbo**
>
> We thank Reviewer 8Wbo for spending their time reviewing our paper. We first notice that Reviewer 8Wbo has clearly stated the following in their review:
>
> > [note confidence score of 1 (@authors + @AC). this is largely outside my comfort zone with many proofs, of which I did not study in details. I would recommend discarding my opinion.]
>
> We next respond to Reviewer 8Wbo's comments to help the reviewer resolve their concerns.
>
>
>
> 1. > Reviewer 8Wbo wrote ''I think lines 35-38 already starts this paper off on the wrong foot. The objective of active learning is to the learn the decision boundary with minimal labels; what the model does with points close to the decision boundary or with high uncertainty once this is learnt could be considered another problem. If the model can the learn the decision boundary with high accuracy given more queries around the decision boundary then this is justified.'', and ''Scenario proposed does not reasonable.''
>
> **Response:** Active learning aims to learn good classifiers with low label complexity (i.e., at least better than passive learning). **However, such a goal cannot be achieved without additional low noise assumptions due to fundamental lower bounds established in active learning (Kaariainen, 2006)**. As a result, to apply active learning in real-world scenarios (i.e., cases without low noise assumptions), it's necessary to consider a refinement of the label complexity goal (as suggested by Kaariainen (2006)). We consider one such refinement with abstention and Chow's excess error and provide the first computationally efficient active learning algorithm that achieves exponential label savings without any low noise assumptions. We believe our problem setup is reasonable, and our results are important to the active learning community (see Section 1.3 for a summary of other main contributions of our paper).
>
> 2. > Reviewer 8Wbo wrote ''Certainly seems new, but a more directed related work would be appreciated.'', ''I find the theoretical exposition somewhat lacking in places-- e.g. for proposition 3 and the statement follows of the algorithms superiority over any uncertainty-based AL method.'', and ''Parts of the paper make strong statements without backing.''
>
> **Response:** As clearly stated in line 163, we discuss additional related work and provide complete proofs in the Appendix (see the supplementary material) due to lack of space.
>
> 3. > Reviewer 8Wbo wrote ''No experimental results whatsoever.''
>
> **Response:** We agree that empirically examining the proposed algorithms is an interesting future direction. However, as it stands, our contributions are theoretical, and we would like them to be viewed as such.
>
> 4. > Reviewer 8Wbo wrote ''The paper is not easy to read.'', and ''Neither well written nor well organised. Language is imprecise and unclear.''
>
> **Response:** We hope Reviewer 8Wbo can explicitly point out where they find the paper hard to follow. We can further polish/re-organize the paper to make it more readable.

---

### Meta-Review · Area_Chair_N2ui · 2022-08-23

**Recommendation:** Accept
**Confidence:** Certain

**Metareview:**

In this paper, the authors develop the first computationally efficient active learning algorithm with abstention, while maintaining the exponential savings in terms of label complexity. Furthermore, the proposed algorithm enjoys other nice properties, such as recovering minimax rates in the standard setting. The algorithm is based on novel applications of techniques from contextual bandits, and the analysis is nontrivial.

On the other hand, the authors should improve their paper by addressing the concerns of reviewers, especially the realizable assumption.


**Award:**

No

---

### Decision · Program_Chairs · 2022-09-14

Accept